# LaSeR: Reinforcement Learning with Last-Token Self-Rewarding

**Wenkai Yang**[1][*]**, Weijie Liu**[2]**, Ruobing Xie**[2]**, Yiju Guo**[1]**,**
**Lulu Wu**[2]**, Saiyong Yang**[2]**, Yankai Lin**[1][†]
[1]Gaoling School of Artificial Intelligence, Renmin University of China
[2]LLM Department, Tencent
{wenkaiyang, yankailin}@ruc.edu.cn

## Abstract

Reinforcement Learning with Verifiable Rewards (RLVR) has recently emerged as a core paradigm for enhancing the reasoning capabilities of Large Language Models (LLMs). To address the lack of verification signals at test time after RLVR, prior studies incorporate the training of model's self-verification capabilities into the standard RLVR process, thereby unifying reasoning and verification capabilities within a single LLM. However, previous practice requires the LLM to sequentially generate solutions and self-verifications using two separate prompt templates, which doubles the inference cost per sample and significantly reduces efficiency. In this work, we theoretically reveal that the closed-form solution to the RL objective of self-verification training can be approximately reduced to a remarkably simple form: **the true reasoning reward of a solution is equal to its *last-token self-rewarding score***, which is computed as the difference between the policy model's next-token log-probability assigned to any pre-specified token at the solution's last token and a pre-calculated constant, scaled by the KL coefficient. Based on this insight, we propose **LaSeR** (Reinforcement Learning with Last-Token Self-Rewarding), an algorithm that simply augments the original RLVR loss with a Mean Squared Error (MSE) loss that aligns the last-token self-rewarding scores with the verifier-based reasoning rewards, and jointly optimizes the reasoning and self-rewarding capabilities of LLMs. The optimized self-rewarding scores serve as auxiliary reward signals in both training and testing to enhance model performance. Notably, our algorithm derives these scores from the predicted next-token probability distribution of the last solution token immediately after solution generation, thereby incurring only the minimal extra cost of at most one additional token inference. Experimental results show that our method not only improves the reasoning performance of the model also equips it with remarkable self-rewarding capability, thereby further boosting its inference-time scaling performance.[1]

## 1 Introduction

In the past few years, Large Language Models (LLMs) (Achiam et al., 2023; MetaAI, 2024a; Qwen Team, 2024; Liu et al., 2024a) have advanced significantly, excelling in various domains (Li et al., 2023; Wang et al., 2024b). However, they still face limitations in complex reasoning tasks (AI-MO, 2024a; OpenCompass, 2025; Rein et al., 2024; Jain et al., 2025). Recently, Reinforcement Learning with Verifiable Rewards (RLVR) has shown great promise in enhancing the complex reasoning abilities of LLMs, as demonstrated by OpenAI o1 (Jaech et al., 2024) and DeepSeek-R1 (Guo et al., 2025). By rewarding reasoning paths based on the consistency between final outcomes and ground-truth answers through a deterministic verifier, RLVR incentivizes LLMs to produce more deliberate reasoning chains while effectively mitigating the risk of reward hacking (Gao et al., 2023).

---

[*]Work done during an internship at Tencent.
[†]Corresponding author.
[1]Code and models are available at `https://github.com/RUCBM/LaSeR`.

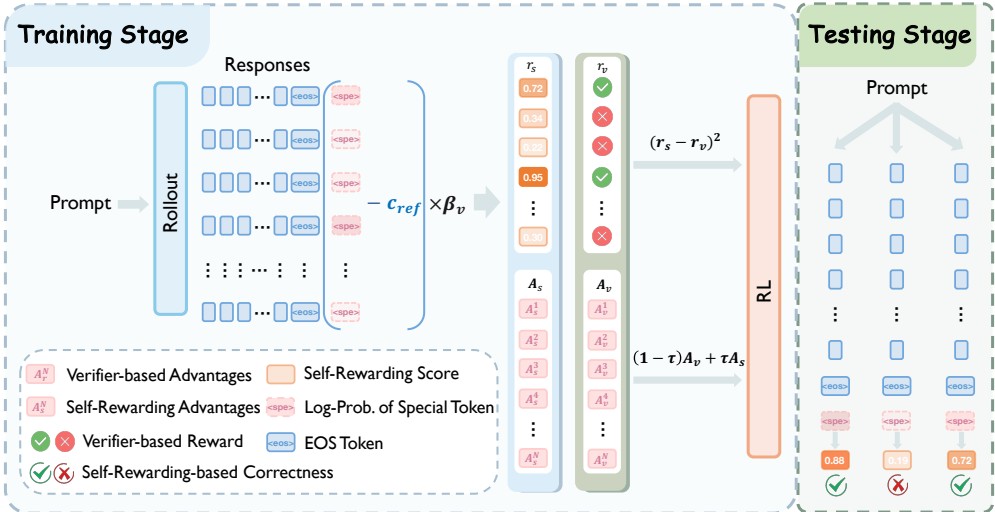

Figure 1: The full illustration of our method **LaSeR**. During training, our approach augments the standard RLVR process with an additional MSE loss between the verifier-based rewards ($r_v$) and the last-token self-rewarding scores ($r_s$), where $r_s$ is the difference between the policy model's next-token log-probabilities of a pre-specified special token at the final response token and a pre-calculated constant $c_{ref}$, scaled by the KL coefficient $\beta_v$. The optimized self-rewarding scores can serve as auxiliary reward signals in both training and testing to enhance model performance.

Despite its effectiveness, a limitation of standard RLVR is its inability to continue providing verification signals for model outputs in scenarios where ground truth answers are unavailable, such as during test-time inference (Zuo et al., 2025). To address this, the standard approach has been to train an external verifier (Lightman et al., 2023; Snell et al., 2024; Zhang et al., 2024; Gao et al., 2024; Yang et al., 2025b) to evaluate candidate solutions. More recently, in pursuit of endowing models with the ability for autonomous and continual self-improvement (Zuo et al., 2025), several works (Sareen et al., 2025; Liu et al., 2025a; Zha et al., 2025; Jiang et al., 2025) have instead explored jointly optimizing the reasoning and self-verification capabilities of a single policy model within the RLVR framework. However, we argue that **these methods have a major issue of inefficiency**: the external verifier requires additional training on a separate LLM during or after reinforcement learning (RL); while joint optimization involves generating both solutions and self-verifications sequentially under two separate prompt templates, which doubles the per-sample inference cost and reduces generation efficiency.

In this work, we propose **LaSeR** (Reinforcement Learning with Last-Token Self-Rewarding), a lightweight and highly effective algorithm that achieves this goal, jointly optimizing reasoning and self-verification capabilities at nearly zero additional cost. Our core insight is that a model's assessment in its own solution can be captured in the last token's predicted probability distribution. We first show theoretically that the RL objective of self-verification has a closed-form solution, where the true reasoning reward from the verifier is equal to the next-token log-probability ratio between the policy and reference models for a pre-specified special token (an unused token like "<|vision_start|>" that serves as the pre-defined ground truth for verifications on correct candidate solutions) at the last response token, scaled by the KL coefficient. We refer to this scaled log-probability ratio as the *last-token self-rewarding score*. Furthermore, we observe that for a randomly chosen special token, its predicted log-probability under the reference model is practically a constant, small value across all problems and solutions (see Figure 7 and Figure 8). This enables us to simplify the self-rewarding score into a remarkably simple form that depends only on the policy model's outputs and a pre-calculated constant, making it exceptionally efficient to compute.

Building on above analysis, we replace the explicit RL optimization for self-verification with a simple Mean Squared Error (MSE) loss. As illustrated in Figure 1, we train the model to align its last-token self-rewarding score with the true reasoning reward from the verifier. In specific, after the policy model generates the solution for each problem, we calculate the last-token self-rewarding score based on its last token's next-token log-probability for the pre-specified special token, and construct the corresponding MSE loss. This MSE objective is added directly to the standard RLVR loss, allowing

for seamless joint optimization for both the reasoning and self-rewarding capabilities of the policy model. At both training and testing time, our method generates each candidate solution and computes the self-rewarding score in a single forward pass, incurring the cost of at most one additional token inference with no extra generation required. This is significantly more efficient than prior approaches, which require a separate inference step. The optimized self-rewarding scores can not only complement the original reasoning rewards during RLVR to further enhance training performance, but also be used at test time to rank and weight solutions for more accurate answer aggregation.

We conduct experiments on both LLaMA3.2 (MetaAI, 2024b) and Qwen2.5 (Qwen Team, 2024) architectures, including pre-trained, mid-trained and reinforced variants, to demonstrate the effectiveness of our method in broader math reasoning tasks. Experimental results show that our methods not only effectively improve the reasoning performance of the policy model, but also allows its self-rewarding accuracy to reach a high level, thereby equipping the model with better confidence calibration of its own outputs and improving its inference-time scaling performance.

## 2 RELATED WORK

**RLVR for LLM Reasoning** Reinforcement Learning with Verifiable Rewards (RLVR), which sorely calculates binary rewards based on the final answers, has been shown to be highly effective in enhancing the reasoning capabilities of LLMs (Jaech et al., 2024; Guo et al., 2025; Team et al., 2025b). Current studies can be categorized into several directions, including but not limited to (1) designing more efficient and effective RLVR algorithms (Schulman et al., 2017; Shao et al., 2024; Yu et al., 2025a; Yue et al., 2025b; Liu et al., 2025b; Zheng et al., 2025; Dai et al., 2026), (2) extending RLVR to general reasoning domain (Ma et al., 2025; Zhou et al., 2025; Yu et al., 2025b) and agent scenarios (Wang et al., 2025b; Team et al., 2025a; Dong et al., 2025), (3) collecting diverse verifiable datasets (Hu et al., 2025; He et al., 2025; Liu & Zhang, 2025; Ma et al., 2025; Fan et al., 2025), and (4) analyzing the mechanisms of RLVR (Mukherjee et al., 2025; Yue et al., 2025a; Wen et al., 2025; Huan et al., 2025).

**External Verifiers for LLM Reasoning** Training external verifiers to identify the correctness of the LLM-generated solutions is an effective way to enhance the reasoning performance of LLMs in the inference time. External verifiers usually fall into two categories: (1) **Scalar Reward Models**: Outcome-supervised Reward Models (ORMs) (Cobbe et al., 2021; Yang et al., 2024) and Process-supervised Reward Models (PRMs) (Lightman et al., 2023; Wang et al., 2024a; Skywork-o1, 2024; Yuan et al., 2024) are two representative approaches. ORMs provide supervision by evaluating the final answer, while PRMs offer more fine-grained feedback by assessing the intermediate reasoning steps. (2) **Generative Verifiers**: Recent studies have explored the potential of training LLMs to perform natural language critiques of reasoning solutions generated by the LLM generators, and then to judge their final outcomes (Zhang et al., 2024; Gao et al., 2024; Yang et al., 2025b; Zhao et al., 2025). This paradigm has demonstrated stronger verification performance than scalar reward models, as it enables the LLM verifier to conduct deliberate chain-of-thought reasoning before arriving at the final judgment.

**Self-Verification for LLM Reasoning** Several recent studies (Sareen et al., 2025; Liu et al., 2025a; Zha et al., 2025; Jiang et al., 2025; Lu et al., 2025) aim to unify the roles of generator and verifier by equipping a single policy model with self-verification capability. The trained self-verification capability can be used in both the RL training and inference-time scaling stages to enhance the model performance. However, these approaches require generating solutions and self-verifications sequentially during training and inference. In contrast, our method derives the self-rewarding signal directly from the next-token probability distribution of the final token of the generated sequence, achieving a more efficient and effective unification of generation and self-verification. We note that a recent study (Lee et al., 2025) also aims to obtain the self-verification result after producing the solution. As discussed in Section 3.4 and Appendix C, our framework is more general and accommodates it as a special case.

## 3 METHODOLOGY

### 3.1 PRELIMINARIES

**RL Objective** We denote $\pi_{\boldsymbol{\theta}}$ as the target policy model to be optimized, and $\pi_{\text{ref}}$ as the reference model from which $\pi_{\boldsymbol{\theta}}$ is initialized. $D$ is the query set, $\boldsymbol{x}$ is an input and $\boldsymbol{y}$ is the generated response

to $\boldsymbol{x}$. The standard optimization objective of RL is formalized as

$$\mathcal{O}_{\pi_{\boldsymbol{\theta}}} = \max_{\pi_{\boldsymbol{\theta}}} \mathbb{E}_{\boldsymbol{x}\sim D, \boldsymbol{y}\sim\pi_{\boldsymbol{\theta}}(\cdot|x)} \left[ r(\boldsymbol{x}, \boldsymbol{y}) - \beta \mathcal{D}_{\mathrm{KL}}(\pi_{\boldsymbol{\theta}}\|\pi_{\mathrm{ref}}) \right], \tag{1}$$

where $r(\boldsymbol{x}, \boldsymbol{y})$ is a reward function, $\mathcal{D}_{\mathrm{KL}}$ is the Kullback–Leibler (KL) divergence loss.

**RLVR** Recently, RLVR (Guo et al., 2025; Hu et al., 2025) has emerged as an effective paradigm for enhancing the reasoning capabilities of LLMs. In RLVR, the reward function $r$ is typically chosen as a deterministic verifier $r_v$, such as a rule-based verifier, to evaluate whether the final extracted answer $\boldsymbol{a} \subset \boldsymbol{y}$ matches the ground-truth answer $\boldsymbol{a}^*$, and to produce binary feedback (e.g., $\{0,1\}$). That is,

$$r_v(\boldsymbol{x}, \boldsymbol{y}) = \mathbb{1}_{\{\boldsymbol{a}\equiv\boldsymbol{a}^*\}} = \begin{cases} 1 & \text{if } \boldsymbol{a} \text{ is semantically equivalent to } \boldsymbol{a}^*, \\ 0 & \text{otherwise.} \end{cases} \tag{2}$$

**Policy Gradient Method** Policy Gradient (Sutton et al., 1998) is a widely adopted algorithm to optimize the objective of Eq. (1), which updates the policy model via the estimated gradient

$$\nabla_{\boldsymbol{\theta}} \mathcal{O}_{\pi_{\boldsymbol{\theta}}} = \mathbb{E}_{\boldsymbol{x}\sim D, \boldsymbol{y}\sim\pi_{\boldsymbol{\theta}}(\cdot|x)} \left[ \sum_{t=1}^{T} A_t \nabla_{\boldsymbol{\theta}} \log \pi_{\boldsymbol{\theta}}(y_t|\boldsymbol{x}, \boldsymbol{y}_{<t}) \right], \tag{3}$$

where $A_t$ is the *advantage function* measuring the relative value of the action $a_t$ (i.e., token $y_t$) compared to the baseline value under state $s_t$ (i.e., sequence $(\boldsymbol{x}, \boldsymbol{y}_{<t})$). In practice, $A_t$ can be estimated in various ways (Schulman et al., 2017; Ahmadian et al., 2024). For example, Group Relative Policy Optimization (GRPO) (Shao et al., 2024) estimates the baseline value as the average reward within a sampled group $\{\boldsymbol{y}^1, \cdots, \boldsymbol{y}^K\}$ for the same problem, and computes the relative advantage for each token $y_t^i$ in sequence $\boldsymbol{y}^i$ as

$$A_t^i = (r_v^i - \mathrm{mean}(r_v^1, \cdots, r_v^K))/\mathrm{std}(r_v^1, \cdots, r_v^K), \quad r_v^i = r_v(\boldsymbol{x}, \boldsymbol{y}^i). \tag{4}$$

**Implicit Reward** Previous studies (Rafailov et al., 2023; Peters & Schaal, 2007) have identified that the optimal solution to the objective Eq. (1) satisfies that

$$r_v(\boldsymbol{x}, \boldsymbol{y}) = \beta \log[\pi_{\boldsymbol{\theta}}(\boldsymbol{y}|\boldsymbol{x})/\pi_{\mathrm{ref}}(\boldsymbol{y}|\boldsymbol{x})] + \beta \log Z(\boldsymbol{x}), \tag{5}$$

where $Z(\boldsymbol{x}) = \sum_{\boldsymbol{y}} \pi_{\mathrm{ref}}(\boldsymbol{y}|\boldsymbol{x}) \exp(\frac{1}{\beta} r_v(\boldsymbol{x}, \boldsymbol{y}))$ is a partition function. $\beta \log \frac{\pi_{\boldsymbol{\theta}}(\boldsymbol{y}|\boldsymbol{x})}{\pi_{\mathrm{ref}}(\boldsymbol{y}|\boldsymbol{x})}$ is termed as the *implicit reward*, which has been used in prior works (Mitchell et al., 2024; Liu et al., 2024b) to analyze the behavioral shift induced by the alignment process.

## 3.2 LaSeR: Reinforcement Learning with Last-Token Self-Rewarding

### 3.2.1 Formal Formulation

In training, ground-truth answers can be reliably used to determine the correctness of solutions. At test time, however, when ground-truth answers are unavailable, the use of verifiers becomes crucial for evaluating solution quality and providing feedback signals. To address this problem, in this work, we explore the promising paradigm of jointly optimizing the self-verification and reasoning capabilities of LLMs within the RLVR framework, thereby enabling them not only to produce high-quality reasoning paths but also to evaluate their own outputs at test time.

According to Eq. (5), as $Z(\boldsymbol{x})$ remains the same for all $\boldsymbol{y}$, a straight-forward idea is to utilize the implicit reward $\beta \log \frac{\pi_{\boldsymbol{\theta}}(\boldsymbol{y}|\boldsymbol{x})}{\pi_{\mathrm{ref}}(\boldsymbol{y}|\boldsymbol{x})}$ as the indicator to rank different generations at test time. However, this approach has a critical drawback: the absolute value of the implicit reward is **length-biased**, since the absolute value of $\beta \log \frac{\pi_{\boldsymbol{\theta}}(\boldsymbol{y}|\boldsymbol{x})}{\pi_{\mathrm{ref}}(\boldsymbol{y}|\boldsymbol{x})} = \beta \sum_i \log \frac{\pi_{\boldsymbol{\theta}}(y_i|\boldsymbol{x}, \boldsymbol{y}_{<i})}{\pi_{\mathrm{ref}}(y_i|\boldsymbol{x}, \boldsymbol{y}_{<i})}$ increases proportionally with the response length. In reasoning tasks, the incorrect solutions are usually longer than the correct solutions (Hassid et al., 2025), making the implicit reward unreliable in evaluating solution correctness (see Appendix D). Furthermore, disregarding $Z(\boldsymbol{x})$ and directly aligning the implicit reward with the true reasoning reward during training degrades the policy model's generation ability (Cui et al., 2025), since a fundamental gap (i.e., $\beta \log Z(\boldsymbol{x})$) exists between the solution to RLVR and that to reward modeling.

In this work, we begin by formulating our approach from the RL objective of verification. Given a problem $\boldsymbol{x}$, and a candidate solution $\boldsymbol{y}$, the model is required to produce a verification $\boldsymbol{z}$ to identify the correctness of the solution: $\boldsymbol{z} \sim \pi_{\boldsymbol{\theta}}(\cdot|\boldsymbol{x}, \boldsymbol{y})$. Thus, the RL objective of verification can be written as

$$\mathcal{V}_{\pi_{\boldsymbol{\theta}}} = \max_{\pi_{\boldsymbol{\theta}}} \mathbb{E}_{\boldsymbol{x}\sim D, \boldsymbol{y}\sim\pi_g(\cdot|x), \boldsymbol{z}\sim\pi_{\boldsymbol{\theta}}(\cdot|\boldsymbol{x},\boldsymbol{y})} \left[ \hat{r}(\boldsymbol{x}, \boldsymbol{y}, \boldsymbol{z}) - \beta_v \mathcal{D}_{\mathrm{KL}}(\pi_{\boldsymbol{\theta}}\|\pi_{\mathrm{ref}}) \right], \tag{6}$$

where $\pi_g$ is the generator to solve the problem (can also be the target model $\pi_{\boldsymbol{\theta}}$ itself in the self-verification setting), $\hat{r}(\boldsymbol{x}, \boldsymbol{y}, \boldsymbol{z})$ is the verification reward that measures the consistency between the true correctness of $\boldsymbol{y}$ and the verification result of $\boldsymbol{z}$. In practice, $\boldsymbol{z}$ can be either a single token—for instance, "Yes" or "No" to directly indicate whether the solution is verified as correct or incorrect—or a sequence that includes both a chain of thought and the final judgment. In this work, we focus on the former setting and simplify the ground-truth label space to two single tokens $z_c$ (e.g., "Yes") and $z_i$ (e.g., "No"). That is, the verification reward can be formulated as[2]

$$\hat{r}(\boldsymbol{x}, \boldsymbol{y}, \boldsymbol{z}) = \begin{cases} 1 & (\boldsymbol{z} = z_c \text{ and } r_v(\boldsymbol{x}, \boldsymbol{y}) = 1) \text{ or } (\boldsymbol{z} = z_i \text{ and } r_v(\boldsymbol{x}, \boldsymbol{y}) = 0) \\ 0 & \text{otherwise.} \end{cases} \quad (7)$$

Similarly, following from Eq. (5), the close-form solution to Eq. (6) can be written as

$$\hat{r}(\boldsymbol{x}, \boldsymbol{y}, \boldsymbol{z}) = \beta_v \log \frac{\pi_{\boldsymbol{\theta}}(\boldsymbol{z}|\boldsymbol{x}, \boldsymbol{y})}{\pi_{\text{ref}}(\boldsymbol{z}|\boldsymbol{x}, \boldsymbol{y})} + \beta_v \log Z(\boldsymbol{x}, \boldsymbol{y}), \quad Z(\boldsymbol{x}, \boldsymbol{y}) = \sum_{\boldsymbol{z}} \pi_{\text{ref}}(\boldsymbol{z}|\boldsymbol{x}, \boldsymbol{y}) \exp(\frac{1}{\beta_v} \hat{r}(\boldsymbol{x}, \boldsymbol{y}, \boldsymbol{z})). \quad (8)$$

Now, let's take a closer look at $Z(\boldsymbol{x}, \boldsymbol{y})$. First, **for $\boldsymbol{z} \in \{z_c, z_i\}$, $\pi_{\text{ref}}(\boldsymbol{z}|\boldsymbol{x}, \boldsymbol{y})$ is a extremely small positive value for any problem-solution pair $(\boldsymbol{x}, \boldsymbol{y})$, i.e., $\pi_{\text{ref}}(\boldsymbol{z}|\boldsymbol{x}, \boldsymbol{y}) \approx 0$, for $\boldsymbol{z} \in \{z_c, z_i\}$.** The reason is that the model is not specifically optimized for predicting the next token once it completes the generation and produces the final token (typically the "<EOS>" token). We present a numerical analysis to validate this claim in Figure 7 and Figure 8, and we can see **the value of $\pi_{\text{ref}}(z|\boldsymbol{x}, \boldsymbol{y})$ is less than $e^{-9}$ for common tokens and even less than $e^{-20}$ for unused special tokens**. Then, under an appropriate choice of $\beta_v$ we can get

$$\begin{aligned} Z(\boldsymbol{x}, \boldsymbol{y}) &= \sum_{\boldsymbol{z}} \pi_{\text{ref}}(\boldsymbol{z}|\boldsymbol{x}, \boldsymbol{y}) \exp(\frac{1}{\beta_v} \hat{r}(\boldsymbol{x}, \boldsymbol{y}, \boldsymbol{z})) = \sum_{\boldsymbol{z} \notin \{z_c, z_i\}} \pi_{\text{ref}}(\boldsymbol{z}|\boldsymbol{x}, \boldsymbol{y}) \exp(\frac{1}{\beta_v} \hat{r}(\boldsymbol{x}, \boldsymbol{y}, \boldsymbol{z})) \\ &+ \pi_{\text{ref}}(z_c|\boldsymbol{x}, \boldsymbol{y}) \exp(\frac{1}{\beta_v} \mathbb{I}_{r_v(\boldsymbol{x}, \boldsymbol{y})=1}) + \pi_{\text{ref}}(z_i|\boldsymbol{x}, \boldsymbol{y}) \exp(\frac{1}{\beta_v}(1 - \mathbb{I}_{r_v(\boldsymbol{x}, \boldsymbol{y})=1})) \\ &\approx (1 - 0 - 0) \exp(0) + 0 + 0 = 1 \implies \log Z(\boldsymbol{x}, \boldsymbol{y}) \approx 0. \end{aligned} \quad (9)$$

The above analysis reveals that, under our formulation, the partition function can be naturally discarded. Consequently, the optimal solution to Eq. (6) can be **approximately** reduced to:

$$\hat{r}(\boldsymbol{x}, \boldsymbol{y}, \boldsymbol{z}) = \beta_v \log[\pi_{\boldsymbol{\theta}}(\boldsymbol{z}|\boldsymbol{x}, \boldsymbol{y})/\pi_{\text{ref}}(\boldsymbol{z}|\boldsymbol{x}, \boldsymbol{y})]. \quad (10)$$

In particular, the true verification reward when the model verifies a solution as correct is:

$$\hat{r}(\boldsymbol{x}, \boldsymbol{y}, z_c) = r_v(\boldsymbol{x}, \boldsymbol{y}) = \beta_v \log[\pi_{\boldsymbol{\theta}}(z_c|\boldsymbol{x}, \boldsymbol{y})/\pi_{\text{ref}}(z_c|\boldsymbol{x}, \boldsymbol{y})]. \quad (11)$$

The first equation is derived from the definition in Eq. (7). The second equation reveals that **the true reasoning reward is equal to log-probability ratio of the policy model to the reference model at $z_c$, scaled by the KL coefficient**. Thus, to optimize the model's verification capability, we do not need to explicitly perform a RLVR procedure. Instead, we can directly optimize the following MSE loss:

$$L = \mathbb{E}_{\boldsymbol{x} \sim D, \boldsymbol{y} \sim \pi_g(\cdot|x)} \left( \beta_v \log[\pi_{\boldsymbol{\theta}}(z_c|\boldsymbol{x}, \boldsymbol{y})/\pi_{\text{ref}}(z_c|\boldsymbol{x}, \boldsymbol{y})] - r_v(\boldsymbol{x}, \boldsymbol{y}) \right)^2. \quad (12)$$

Thus, in the self-verification setting where $\pi_g = \pi_{\boldsymbol{\theta}}$, we can directly adds the above loss into the original RLVR loss to jointly optimize the reasoning and self-verification capabilities of the policy model:

$$\mathcal{S}_{\pi_{\boldsymbol{\theta}}} = \max_{\pi_{\boldsymbol{\theta}}} \mathbb{E}_{\boldsymbol{x} \sim D, \boldsymbol{y} \sim \pi_{\boldsymbol{\theta}}(\cdot|x)} \left\{ r_v(\boldsymbol{x}, \boldsymbol{y}) - \beta \mathcal{D}_{\text{KL}}(\pi_{\boldsymbol{\theta}} \| \pi_{\text{ref}}) - \alpha \left[ \beta_v \log \frac{\pi_{\boldsymbol{\theta}}(z_c|\boldsymbol{x}, \boldsymbol{y})}{\pi_{\text{ref}}(z_c|\boldsymbol{x}, \boldsymbol{y})} - r_v(\boldsymbol{x}, \boldsymbol{y}) \right]^2 \right\}, \quad (13)$$

where $\alpha$ is a loss balancing coefficient. We refer the term $r_s = \beta_v \log \frac{\pi_{\boldsymbol{\theta}}(z_c|\boldsymbol{x}, \boldsymbol{y})}{\pi_{\text{ref}}(z_c|\boldsymbol{x}, \boldsymbol{y})}$ to the **last-token self-rewarding score**, since it depends on the log-probability distributions of the last token in $\boldsymbol{y}$.

## 3.3 OTHER TECHNIQUES

Here, we discuss several practical techniques to further simplify and improve the efficiency and effectiveness of the self-rewarding MSE loss introduced above.

---

[2]In Appendix C, we further provide the derivations to demonstrate the general form of our framework.

**Simplification of the Log-Probability in the Reference Model** As illustrated in Figure 7 and Figure 8, the quantity $\log \pi_{\text{ref}}(z_c|\boldsymbol{x}, \boldsymbol{y})$ remains almost constant and stable during policy model's training. Therefore, we can regard it as a pre-calculated constant $c_{\text{ref}}$ in calculating the last-token self-rewarding score during both training and inference. This eliminates the need for forwarding $\boldsymbol{y}$ through the reference model and thus further enhances efficiency. In specific, $c_{\text{ref}}$ is the mean value of $\log \pi_{\text{ref}}(z_c|\boldsymbol{x}, \boldsymbol{y})$ on a small set of pre-generated set of $(\boldsymbol{x}, \boldsymbol{y})$. Furthermore, based on the findings in Figure 7, we select an unused special token as $z_c$ to make $\pi_{\text{ref}}(z_c|\boldsymbol{x}, \boldsymbol{y})$ closer to 0 and to further minimize its impact on the approximation of $Z(\boldsymbol{x}, \boldsymbol{y}) = 1$ and the stability of training.

**Self-Rewarding Loss Re-Weighting** During training, the numbers of correct and incorrect solutions are imbalanced, and their ratio dynamically changes. To prevent the last-token self-rewarding score from being biased toward the class with more samples, we apply a class-level loss re-weighting strategy within each optimization step. In each step, we calculate the total numbers of correct and incorrect solutions (identified by the verifier) for all problems in the current batch as $N_c$ and $N_i$. Then, we apply the loss re-weighting as

$$l = \frac{1}{N_c + N_i} \sum_{\boldsymbol{x}} \sum_{\boldsymbol{y}} \left[ w_c \mathbb{1}_{\{r_v(\boldsymbol{x}, \boldsymbol{y})=1\}} + w_i \mathbb{1}_{\{r_v(\boldsymbol{x}, \boldsymbol{y})=0\}} \right] \left[ \beta_v \log \pi_{\boldsymbol{\theta}}(z_c|\boldsymbol{x}, \boldsymbol{y}) - \beta_v c_{\text{ref}} - r_v(\boldsymbol{x}, \boldsymbol{y}) \right]^2, \quad (14)$$

where $w_c = \frac{N_c + N_i}{2 \times N_c}$ and $w_i = \frac{N_c + N_i}{2 \times N_i}$ are re-weighting factors. This practice achieves a more balanced self-verification capability. We provide empirical validations on this in Appendix J.

**Integration of Verifier-based and Self-Rewarding-based Advantages** The last-token self-rewarding scores can not only be used at test time, but also facilitate the training process through the integration of verifier-based and self-rewarding-based advantages. We believe such practice can help mitigate the issue of misjudgments by rule-based verifiers, which often occur when the format of ground-truth answer is overly complex, and produce more fine-grained rewards. For example, in GRPO, the final advantage can be calculated as:

$$\hat{A}_t^i = (1 - \tau) \frac{r_v^i - \text{mean}(r_v^1, \cdots, r_v^K)}{\text{std}(r_v^1, \cdots, r_v^K)} + \tau \frac{r_s^i - \text{mean}(r_s^1, \cdots, r_s^K)}{\text{std}(r_s^1, \cdots, r_s^K)},$$
$$\text{where } r_v^i = r_v(\boldsymbol{x}, \boldsymbol{y}^i) \text{ and } r_s^i = \beta_v \log \pi_{\boldsymbol{\theta}}(z_c|\boldsymbol{x}, \boldsymbol{y}^i) - \beta_v c_{\text{ref}}. \quad (15)$$

To stabilize training, we adopt a filtering strategy that sets $\tau = 0$ for any group whenever the standard deviation $\text{std}(r_s^1, \cdots, r_s^K)$ within this group falls below a threshold $T$, which is set to 0.1.

**Separate Warm-Up of Reasoning and Self-Rewarding Capabilities** During the initial phase of training, we optimize only the last-token self-rewarding score, without integrating self-rewarding-based advantages into the learning process. After a certain steps when the last-token self-rewarding loss is sufficiently small, we proceed to integrate verifier-based and self-rewarding-based advantages. Moreover, when training from base (i.e., pre-trained) models, we first perform standard RLVR without incorporating the last-token self-rewarding loss in order to warm up the model's reasoning capability, followed by a warm-up phase for the self-rewarding capability before the advantage integration.

By combining all the aforementioned techniques, our full algorithm **Reinforcement Learning with Last-Token Self-Rewarding** (LaSeR), is summarized in Algorithm 1 and illustrated in Figure 1. During the testing phase, once the model generates a solution, we compute the last-token self-rewarding score based on $r_s = \beta_v \log \pi_{\boldsymbol{\theta}}(z_c|\boldsymbol{x}, \boldsymbol{y}) - \beta_v c_{\text{ref}}$. We then clip it into the interval $[0, 1]$. Notably, we point out that this clipping is optional: after optimization, the self-rewarding scores naturally fall within $[0, 1]$ (refer to Table 7), further validating the boundedness of our self-scoring mechanism. The comparison between this score and 0.5 determines the self-verification outcome of the solution, or the score itself can be further used to perform weighted majority voting.

### 3.4 Brief Discussion

**Comparison Between LaSeR and Prior Approaches** Compared with previous methods (Sareen et al., 2025; Liu et al., 2025a; Zha et al., 2025) that requires the policy model to perform separate generations for solutions and verifications, our method directly derives the self-rewarding result from the next-token log-probability of the final solution token. In the RL process, the computation of token log-probabilities is typically carried out after all the generations are completed (Sheng et al., 2024). Therefore, we can directly replace the token id of the first padding token with the token id of the pre-specified token before computing the log-probabilities of the sequences, thereby **incurring no**

**additional computation cost during training**. **During inference, our method requires only one more token inference after the solution is completed**, which substantially reduces the computational cost compared to previous methods.

**Difference Between Last-Token Self-Rewarding Loss and Supervised Fine-Tuning Loss** An alternative to train the self-verification capability is to optimize the following SFT/BCE loss by maximizing the next-token probability of the token $z_c$ or $z_i$ based on the context $(\boldsymbol{x}, \boldsymbol{y})$:

$$L_{SFT} = -\mathbb{E}_{\boldsymbol{x} \sim D, \boldsymbol{y} \sim \pi_g(\cdot|x)} \left[ r_v(\boldsymbol{x}, \boldsymbol{y}) \cdot \log \pi_{\boldsymbol{\theta}}(z_c | \boldsymbol{x}, \boldsymbol{y}) + (1 - r_v(\boldsymbol{x}, \boldsymbol{y})) \cdot \log \pi_{\boldsymbol{\theta}}(z_i | \boldsymbol{x}, \boldsymbol{y}) \right]. \tag{16}$$

The major difference between SFT loss and our last-token self-rewarding loss in Eq. (12) is that the SFT loss drives $\pi_{\boldsymbol{\theta}}(z_c | \boldsymbol{x}, \boldsymbol{y})$ to fit 1 when $r_v(\boldsymbol{x}, \boldsymbol{y}) = 1$, which may lead to strong interference with the optimization of reasoning capability. In contrast, our loss drives $\pi_{\boldsymbol{\theta}}(z_c | \boldsymbol{x}, \boldsymbol{y})$ toward $\exp(1/\beta_v) \cdot \pi_{\text{ref}}(z_c | \boldsymbol{x}, \boldsymbol{y})$ for $r_v(\boldsymbol{x}, \boldsymbol{y}) = 1.0$. When $\beta_v$ is relatively large, $\pi_{\boldsymbol{\theta}}(z_c | \boldsymbol{x}, \boldsymbol{y})$ remains still very small, thereby exerting only a negligible influence on the original RLVR optimization (e.g., $\pi_{\boldsymbol{\theta}}(z_c | \boldsymbol{x}, \boldsymbol{y}) = e^{-13}$ when $\pi_{\text{ref}}(z_c | \boldsymbol{x}, \boldsymbol{y}) = e^{-23}$ and $\beta_v = 0.1$). We provide further discussions in Appendix C and put the empirical comparison in Appendix K.

# 4 EXPERIMENTS

## 4.1 EXPERIMENTAL SETTINGS

**Base Models and Baselines** We conduct empirical validations on both LLaMA3.2 MetaAI (2024b) and Qwen2.5 (Qwen Team, 2024) architectures, including three base models: OctoThinker-3B-Short-Base (Wang et al., 2025a) (mid-trained version of LLaMA3.2-3B-Base), Qwen2.5-7B-Base (Qwen Team, 2024) (pre-trained model) and Open-Reasoner-Zero-7B (Hu et al., 2025) (reinforced version of Qwen2.5-7B-Base). In principle, our method can be seamlessly integrated into any RLVR framework, as it only introduces an additional MSE loss term. In the main experiments, we adopt the widely used GRPO (He et al., 2025) as the base algorithm and primarily investigate the effectiveness of applying our method within GRPO. We also conduct experiments with PPO (Schulman et al., 2017) to demonstrate the generalizability of our method, and present the corresponding results in Appendix Q.

**Training and Evaluation Datasets** We adopt DeepMath-103K (He et al., 2025), a large-scale and high-quality mathematical reasoning dataset, for our RL training data. In testing, we evaluate both the reasoning and self-verification performance of each model on five typical math reasoning benchmarks: MATH500 (Hendrycks et al., 2021), AMC23 (AI-MO, 2024b), AIME24 (AI-MO, 2024a), AIME25 (OpenCompass, 2025), and OlympiadBench (He et al., 2024). Additionally, we also perform experiments in the general reasoning domain to validate the effectiveness of our method in general reasoning tasks beyond math reasoning. The results and analysis are in Appendix R.

**Training Settings** The detailed training hyper-parameters of GRPO are put in Appendix G. The prompt template for each model is in Appendix T. When applying our method, we set the hyper-parameters $(\beta_v, \alpha, \tau) = (0.1, 0.1, 0.1)$ (refer to Appendix I). $z_c$ is selected as "`<vision_start>`" for Qwen2.5-7B-Base and Open-Reasoner-Zero-7B, and "`<reserved_special_token_0>`" for OctoThinker-3B-Short-Base. The simplified constant of the reference log-probability, $c_{\text{ref}}$, is $-23.0$ for Qwen2.5-7B-Base and Open-Reasoner-Zero-7B, and $-25.0$ for OctoThinker-3B-Short-Base, as estimated from the results in Figure 7. More details are in Appendix G.

**Evaluation Settings** During generation, we set both the `temperature` and `top_p` to $1.0$ for all models. The `max_generation_len` is 8192. On MATH500 and OlympiadBench, we sample 2 solutions for each problem; whereas on AMC23, AIME24, and AIME25, we sample 32 solutions per problem. We then report the average Pass@1 accuracy of each model on each benchmark. We also evaluate the self-verification performance of each model by computing the self-verification F1 score, defined as the harmonic mean of self-verification accuracy on self-generated correct and incorrect solutions. Baselines perform self-verification based on the prompt in Appendix T. Any solution without a final answer is automatically treated as incorrect and excluded from the verification accuracy calculation. Detailed self-verification accuracy results are provided in Appendix N.

## 4.2 MAIN RESULTS AND ANALYSIS

We put the main results in Table 1. The key conclusion is that, **across different model variants, our method not only yields better reasoning performance for the policy model compared with**

Table 1: Reasoning and self-verification performance of each model on five mathematical reasoning benchmarks. We do not report the results of OctoThinker-based models on AIME24-25, as the number of correct solutions is quite insufficient for a reliable evaluation.

| Method | Reasoning Accuracy | | | | | | Self-Verification F1 Score | | | | | |
|---|---|---|---|---|---|---|---|---|---|---|---|---|
| | MATH-500 | AMC-23 | AIME-24 | AIME-25 | Olym.-Bench | Avg. | MATH-500 | AMC-23 | AIME-24 | AIME-25 | Olym.-Bench | Avg. |
| *OctoThinker-3B-Short-Base* | | | | | | | | | | | | |
| Base | 3.7 | 1.3 | - | - | 1.0 | 2.0 | 22.3 | 11.2 | - | - | 13.7 | 15.7 |
| GRPO | 49.8 | 25.3 | - | - | 17.3 | 30.8 | 56.9 | 47.3 | - | - | 48.8 | 51.0 |
| LaSeR | **53.1** | **27.0** | - | - | **18.2** | **32.8** | 73.6 | 70.2 | - | - | **73.6** | 72.5 |
| *- SWA* | 52.9 | 26.1 | - | - | **18.2** | 32.4 | 80.4 | 70.9 | - | - | 66.0 | 72.4 |
| *Qwen2.5-7B-Base* | | | | | | | | | | | | |
| Base | 35.8 | 20.6 | 3.5 | 1.6 | 12.3 | 14.8 | 36.4 | 30.8 | 27.6 | 32.9 | 36.9 | 32.9 |
| GRPO | 79.9 | 55.9 | **16.2** | 13.8 | 43.3 | 41.8 | 54.6 | 59.7 | 36.6 | 41.5 | 53.5 | 49.2 |
| LaSeR | **80.2** | 58.1 | 15.4 | **15.7** | **44.1** | **42.7** | 83.2 | 82.5 | 79.6 | 74.3 | 78.3 | 79.6 |
| *- SWA* | 78.0 | **58.3** | 15.4 | 12.3 | 41.7 | 41.1 | 79.7 | 80.2 | **81.3** | **74.9** | **83.3** | **79.9** |
| *Open-Reasoner-Zero-7B* | | | | | | | | | | | | |
| Base | 81.9 | 60.3 | 15.6 | **15.1** | 46.9 | 44.0 | 26.7 | 51.3 | 45.9 | 55.2 | 37.5 | 43.3 |
| GRPO | 83.1 | 61.9 | 18.1 | 15.0 | 47.1 | 45.0 | 57.1 | 44.8 | 14.6 | 28.1 | 49.5 | 38.8 |
| LaSeR | 82.8 | **62.7** | **19.1** | **15.1** | **47.8** | **45.5** | 87.2 | **79.7** | **64.6** | 77.7 | **78.7** | **77.6** |
| *- SWA* | **83.2** | 62.6 | 19.0 | 14.5 | 47.6 | 45.4 | **87.5** | 77.7 | 63.3 | 77.3 | 77.9 | 76.7 |

Table 2: Comparison of verification F1 scores between LaSeR (self-rewarding) and external reward models on responses generated by the same policy model Open-Reasoner-Zero-7B-LaSeR. **Bold** indicates the best result, and underline indicates the second-best.

| Method | MATH500 | AMC23 | AIME24 | AIME25 | Olym. | Avg. |
|---|---|---|---|---|---|---|
| Qwen2.5-Math-7B-PRM800K (7B RM) | 56.3 | 42.5 | 51.4 | 50.8 | 38.5 | 47.9 |
| Qwen2.5-Math-PRM-7B (7B RM) | 86.0 | 79.6 | 70.8 | 67.3 | 76.0 | 75.9 |
| Qwen2.5-Math-RM-72B (72B RM) | 86.8 | 79.4 | 71.0 | 71.4 | 75.5 | 76.8 |
| Open-Reasoner-Zero-7B-RM (7B RM) | 85.9 | 78.1 | **73.8** | **79.2** | 77.3 | **78.9** |
| LaSeR (7B Self-Rewarding) | **87.2** | **79.7** | 64.6 | 77.7 | **78.7** | 77.6 |

**the baseline, but also substantially enhances its self-verification capability by enabling the self-rewarding scores to achieve remarkably high F1 scores**.

Regarding reasoning performance, applying our algorithm leads to higher accuracy in most settings and consistently yields higher average accuracy on the three base models. We think there are two main reasons for this improvement: (1) First, our method encourages the model to encode its assessment of the overall solution in the final response token, which leads to better confidence calibration. Improved calibration itself can have a positive impact on the model's learning. (2) Second, by integrating self-rewarding-based advantages into verifier-based advantages, our approach enables more fine-grained advantage estimation, which in turn facilitates more effective learning. For comparison, we report the results without self-rewarding-based advantages (*-SWA*) in Table 1. Regarding self-verification performance, applying a simple last-token self-rewarding MSE loss substantially enhances the self-rewarding capability of the models, achieving around 80% F1 scores. This demonstrates strong self-verification accuracy on both correct and incorrect solutions. To further highlight the self-rewarding capabilities, we display the comparison results of verification F1 scores between LaSeR and several advanced external reward models (Qwen2.5-Math-7B-PRM800K (Zhang et al., 2025), Qwen2.5-Math-PRM-7B (Zhang et al., 2025), and Qwen2.5-Math-RM-72B (Yang et al., 2024)) on evaluating the solutions generated by the different reinforced models by LaSeR. The full results are in Table 6, while here we only display the results on Open-Reasoner-Zero-7B-LaSeR in Table 2. Moreover, to ensure a fairer comparison, we additionally train an ORM using the same backbone (Open-Reasoner-Zero-7B) and on the same data distribution (a total of 500K responses generated by Open-Reasoner-Zero-7B-LaSeR on DeepMath-103K), resulting in Open-Reasoner-Zero-7B-RM. The comparison results demonstrate the great effectiveness of self-rewarding. Moreover, it is worth noting that training and employing external reward models introduces additional computational

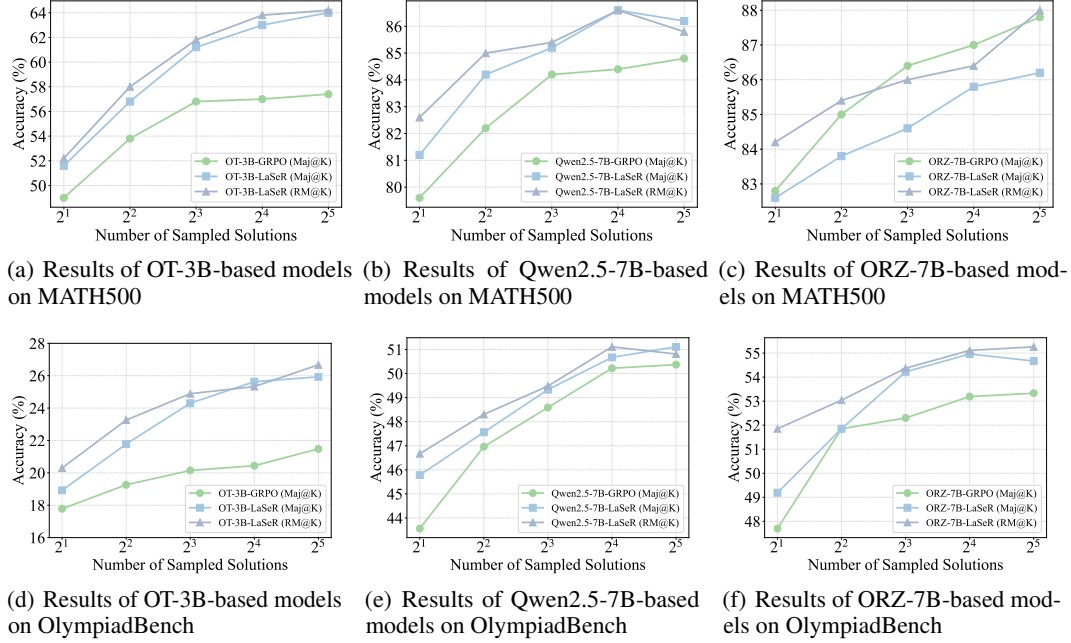

(a) Results of OT-3B-based models on MATH500

(b) Results of Qwen2.5-7B-based models on MATH500

(c) Results of ORZ-7B-based models on MATH500

(d) Results of OT-3B-based models on OlympiadBench

(e) Results of Qwen2.5-7B-based models on OlympiadBench

(f) Results of ORZ-7B-based models on OlympiadBench

Figure 2: The majority voting (Maj@K) and weighted majority voting (RM@K) results.

overhead during both training and inference. In contrast, our method jointly optimizes the reasoning and self-rewarding capabilities within the RLVR framework and obtains the verification outcomes directly after solution generation, incurring no extra computational cost at either training or test time. Furthermore, in Appendix P, we show that the model optimized through LaSeR can even score the outputs of other models, demonstrating **strong generalization in cross-model verification**.

### 4.3 INFERENCE-TIME SCALING RESULTS

Here, we explore the effectiveness of self-rewarding in the inference-time scaling via weighted majority voting. We compare majority voting results with (RM@K) and without (Maj@K) weighting by the last-token self-rewarding scores, on MATH500 and OlympiadBench. The results are shown in Figure 2. We denote the three base models by "OT-3B", "Qwen2.5-7B", and "ORZ-7B". The suffixes "-GRPO" and "-LaSeR" indicate the variants trained with GRPO and our method LaSeR, respectively. The results show that **the optimized self-rewarding capability of the model is highly effective on improving its own inference-time scaling performance**.

## 5 ANALYSIS

### 5.1 MORE CALIBRATION ANALYSIS OF SELF-REWARDING SCORES

In this section, we conduct additional calibration analysis to better understand the properties of our self-rewarding scores. We take Open-Reasoner-Zero-7B-LaSeR as the experimental model. To ensure reliability, we sample 32 times for each query in the following analysis.

We first visualize the distribution of optimized self-rewarding scores on OlympiadBench in Figure 3. Furthermore, we analyze the average **AUROC** score of the self-rewarding scores with respect to correctness. The AUROC is defined as the probability that, within the same query group (solutions generated for the same query), a correct solution receives a higher self-rewarding score than an incorrect one. The full results are in Table 9. These results re-validate the high discriminative power of the self-rewarding score between correct and incorrect solutions.

Then, we calculate and report the **Expected Calibration Error** (**ECE**) of all self-rewarding scores on each test set. ECE quantifies the discrepancy between a self-rewarding score and its empirical

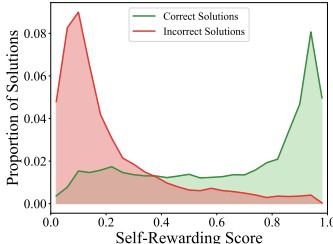 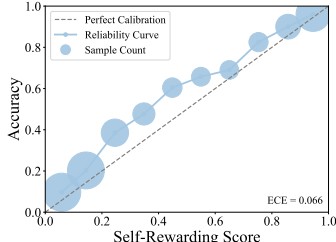 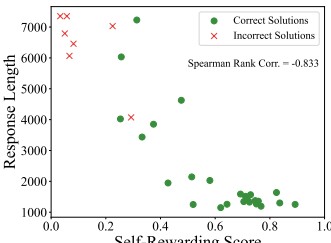

Figure 3: Self-rewarding score distribution on OlympiadBench.

Figure 4: Reliability diagram of self-rewarding scores on OlympiadBench.

Figure 5: Relationship between response lengths and self-rewarding scores.

Table 3: Comparison of reasoning and self-verification performance with and without reference log-probability simplification in our method. Base model is Open-Reasoner-Zero-7B.

| Method | Reasoning Accuracy | | | | | | Self-Verification F1 Score | | | | | |
|---|---|---|---|---|---|---|---|---|---|---|---|---|
| | MATH-500 | AMC-23 | AIME-24 | AIME-25 | Olym.-Bench | Avg. | MATH-500 | AMC-23 | AIME-24 | AIME-25 | Olym.-Bench | Avg. |
| w/ Simpl. | 82.5 | 61.6 | 18.8 | 16.2 | 46.5 | 45.1 | 82.3 | 79.3 | 77.9 | 79.2 | 78.4 | 79.4 |
| w/o Simpl. | 81.0 | 61.2 | 17.3 | 17.3 | 48.3 | 45.0 | 81.8 | 79.2 | 79.0 | 78.9 | 77.5 | 79.3 |

accuracy. The results on OlympiadBench are in Figure 4, while we put the full results in Table 10. As shown, our method demonstrates strong confidence calibration results.

Finally, we analyze the effect of response lengths. We visualize the relationship between response lengths and self-rewarding scores within a single chosen query group from OlympiadBench in Figure 5 as a preliminary analysis. Then, we calculate and report the average **Spearman Rank Correlation** between self-rewarding scores and response lengths over all query groups on each test set in Table 11. These results reveal that shorter responses generally tend to receive higher self-rewarding scores, which is consistent with recent findings (Hassid et al., 2025; Yang et al., 2025c) that shorter CoTs on the same query are generally more preferable.

## 5.2 THE IMPACT OF SIMPLIFYING THE REFERENCE LOG-PROBABILITIES TO A CONSTANT

As discussed in Section 3.3, we approximate the log-probability of the pre-specified token under the reference model, $\log \pi_{\text{ref}}(z_c|\boldsymbol{x}, \boldsymbol{y})$, by its mean computed over a small sample set when calculating the last-token self-rewarding scores. Here, we empirically validate this practice by conducting comparison experiments on Open-Reasoner-Zero-7B, with and without reference log-probability simplification in our method. We evaluate the checkpoint after 200 optimization steps in each setting (corresponding to the last checkpoint before advantage integration). The results are reported in Table 3. As shown, **the simplification does not affect the optimization of reasoning and self-rewarding capabilities**, since the performance under the two settings remains comparable. However, it effectively reduces the computational cost of calculating the last-token self-rewarding value by half.

## 6 CONCLUSION

In this work, we propose **LaSeR**, a lightweight and effective algorithm that jointly optimizes the reasoning and self-rewarding capabilities of LLMs. By deriving the closed-form solution to the RL objective of verification, we uncover a concise yet intriguing formula: the true reasoning reward provided by the verifier is equal to the last-token self-rewarding score produced by the model. This self-rewarding score depends on the model's next-token log-probability for a pre-specified token at the final response token, a pre-calculated constant, and the KL coefficient. Based on this insight, our method simply adds a MSE loss between the verifier-based reasoning rewards and the corresponding last-token self-rewarding scores into the standard RLVR process. The optimized self-rewarding scores can not only be incorporated back into the RL process to further enhance training, but also achieve high verification accuracy at test time, thereby improving solution ranking and selection.

ETHICS STATEMENT

Our work aims to jointly optimize the reasoning and self-rewarding capabilities of LLMs. Our method enables the LLMs to self-verify the correctness of its solutions with minimal extra computational cost and high verification accuracy, improving both performance and efficiency. Our method has the positive impact on the field of LLM reasoning by enabling LLMs to produce not only higher-quality and more interpretable solutions, but also accurate self-verifications. All datasets and models used in this study are publicly available.

REPRODUCIBILITY STATEMENT

We open-source the code and release the model checkpoints. We provide the full experimental settings in Section 4.1 and Appendix G to facilitate reproducibility. We discuss all the techniques used in our method in Section 5.2, Appendix E and Appendix J. We provide the ablation studies on hyper-parameters in Appendix I. All prompt templates used in our experiments are presented in Appendix T.

ACKNOWLEDGMENTS

We sincerely thank all the anonymous reviewers and (S)ACs for their careful reviews and constructive comments. This work was supported by The National Natural Science Foundation of China (No. 62376273 and No. U2436209), Beijing Nova Program (No. 20240484568) and Beijing Natural Science Foundation (L253001).

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

# A    THE STATEMENT ON THE USE OF LARGE LANGUAGE MODELS

In this work, we only use LLMs to correct grammatical issues and polish the writing. We do not use LLMs for research ideation or full paper writing.

# B    ADDITIONAL RELATED WORK

**RLHF for LLMs** Reinforcement Learning with Human Feedback (RLHF) (Ouyang et al., 2022) has been a key technique for aligning a model's capabilities and behaviors with human preferences. The standard pipeline typically consists of three stages: supervised fine-tuning on high-quality instructional data, learning a reward model from pairwise human preference data, and optimizing the policy model via reinforcement learning to maximize the learned reward. Prior work in RLHF spans a broad range of directions, including but not limited to: (1) Developing more effective or efficient RLHF algorithms and frameworks (Schulman et al., 2017; Ahmadian et al., 2024; Rafailov et al., 2023); (2) Constructing diverse, large-scale, and high-quality preference datasets (Bai et al., 2022; Cui et al., 2024); (3) Training stronger or domain-specific reward models to improve RLHF performance (Dong et al., 2024; Cai et al., 2024; Yuan et al., 2025); (4) Exploring multi-objective alignment (Guo et al., 2024; Li et al., 2025).

# C    GENERAL FORM OF SELF-REWARDING MSE LOSS

In the main content, we derive the self-rewarding MSE loss in Eq. (12) by instantiating the verification reward in Eq. (7) under the binary 0/1 setting (though this is common choice). Here, we make further derivations to obtain the general form of our self-rewarding MSE loss by allowing the rewards for correct and incorrect verifications to take arbitrary real values, denoted as $\hat{r}_c$ and $\hat{r}_i$, respectively. Then, we clarify why our method adopts the MSE loss instead of the BCE loss, and highlight that the solution to the BCE loss in Eq. (16) is in fact a special case of all available solutions in our framework.

Under the general definition, following the same derivation as in Eq. (9), we obtain $Z(\boldsymbol{x}, \boldsymbol{y}) = \exp(\frac{\hat{r}_i}{\beta_v})$, which indicates $\log Z(\boldsymbol{x}, \boldsymbol{y}) = \frac{\hat{r}_i}{\beta_v}$ (still a constant but not necessarily to be 0). Therefore, $\log Z(\boldsymbol{x}, \boldsymbol{y}) = 0$ in Eq. (9) is a special case when we define $\hat{r}_i = 0$. Then, the optimal solution to the original RL target can be reduced to a general form

$$\hat{r}(\boldsymbol{x}, \boldsymbol{y}, \boldsymbol{z}) = \beta_v \log[\pi_{\boldsymbol{\theta}}(\boldsymbol{z}|\boldsymbol{x}, \boldsymbol{y})/\pi_{\text{ref}}(\boldsymbol{z}|\boldsymbol{x}, \boldsymbol{y})] + \hat{r}_i.$$

That is,

$$\hat{r}(\boldsymbol{x}, \boldsymbol{y}, \boldsymbol{z}) - \hat{r}_i = \beta_v \log[\pi_{\boldsymbol{\theta}}(\boldsymbol{z}|\boldsymbol{x}, \boldsymbol{y})/\pi_{\text{ref}}(\boldsymbol{z}|\boldsymbol{x}, \boldsymbol{y})].$$

Now, **we can see that the left-hand side of the equation is no longer a 0/1 binary reward but rather two arbitrary scalar values. Therefore, it is natural to model this objective using an MSE loss rather than a BCE loss.**

Following above, the true verification reward when the model verifies a solution as correct is

$$\hat{r}(\boldsymbol{x}, \boldsymbol{y}, z_c) = \hat{r}_c \mathbb{I}_{r_v(\boldsymbol{x}, \boldsymbol{y})=1} + \hat{r}_i \mathbb{I}_{r_v(\boldsymbol{x}, \boldsymbol{y})=0} = \beta_v \log[\pi_{\boldsymbol{\theta}}(z_c|\boldsymbol{x}, \boldsymbol{y})/\pi_{\text{ref}}(z_c|\boldsymbol{x}, \boldsymbol{y})] + \hat{r}_i.$$

Note that the reasoning rewards $r_v(\boldsymbol{x}, \boldsymbol{y})$ defined in Eq. (2) can also be two arbitrary values, but we simplified them to $\{0, 1\}$ and this will not affect our derivation. This formulation indicates that the probability $\pi_{\boldsymbol{\theta}}(z_c|\boldsymbol{x}, \boldsymbol{y})$ alone is sufficient to model the reward score. In this general framework, our self-rewarding MSE loss is

$$L = \mathbb{E}_{\boldsymbol{x} \sim D, \boldsymbol{y} \sim \pi_g(\cdot|x)} \left( \beta_v \log[\pi_{\boldsymbol{\theta}}(z_c|\boldsymbol{x}, \boldsymbol{y})/\pi_{\text{ref}}(z_c|\boldsymbol{x}, \boldsymbol{y})] + \hat{r}_i - (\hat{r}_c \mathbb{I}_{r_v(\boldsymbol{x}, \boldsymbol{y})=1} + \hat{r}_i \mathbb{I}_{r_v(\boldsymbol{x}, \boldsymbol{y})=0}) \right)^2.$$

Now, we show that **the solution to the SFT/BCE loss in Eq. (16) is in fact a special case of all available solutions in our framework**. After optimization (i.e., the optimal solution to the MSE loss), we can see that when $r_v(\boldsymbol{x}, \boldsymbol{y}) = 0$ (i.e., the solution is incorrect based on the deterministic verifier), $\pi_{\boldsymbol{\theta}}(z_c|\boldsymbol{x}, \boldsymbol{y}) = \pi_{\text{ref}}(z_c|\boldsymbol{x}, \boldsymbol{y})$, which means the $\pi_{\boldsymbol{\theta}}(z_c|\boldsymbol{x}, \boldsymbol{y})$ remains unchanged (almost near zero, consistent with the solution to SFT loss when $r_v(\boldsymbol{x}, \boldsymbol{y}) = 0$). When $r_v(\boldsymbol{x}, \boldsymbol{y}) = 1$ (i.e., the solution is correct based on the deterministic verifier), our method drives $\pi_{\boldsymbol{\theta}}(z_c|\boldsymbol{x}, \boldsymbol{y})$ to fit

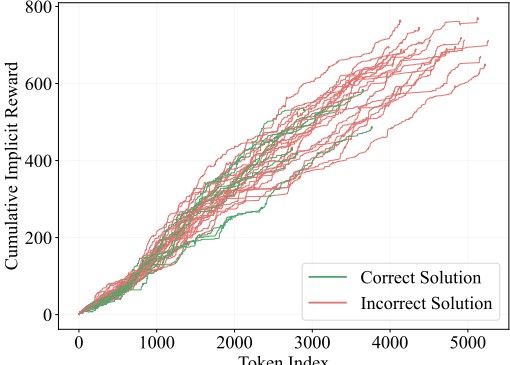
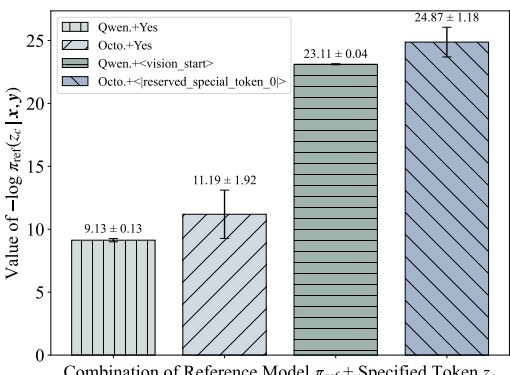

Figure 6: Cumulative implicit reward values across 32 reasoning trajectories sampled from Open-Reasoner-Zero-7B on an AIME2024 problem. Red lines correspond to wrong solutions and green lines correspond to correct solutions.

Figure 7: The mean and standard deviation of $-\log \pi_{\text{ref}}(z_c|\boldsymbol{x}, \boldsymbol{y})$ under different combinations of reference model $\pi_{\text{ref}}$ and pre-specified token $z_c$ over 300 input-output pairs.

$\pi_{\text{ref}}(z_c|\boldsymbol{x}, \boldsymbol{y}) \exp(\frac{\hat{r}_c - \hat{r}_i}{\beta_v})$. As we can see, when $\beta_v$ continues to decrease, the optimized probability $\pi_{\boldsymbol{\theta}}(z_c|\boldsymbol{x}, \boldsymbol{y})$ approaches 1, which is exactly the solution to the BCE loss. However, in our framework, **we can explicitly control the value of $\beta_v$, allowing $\pi_{\boldsymbol{\theta}}(z_c|\boldsymbol{x}, \boldsymbol{y})$ to converge to any desired target**. As shown in Appendix K, this controllability provides greater flexibility compared with directly forcing $\pi_{\boldsymbol{\theta}}(z_c|\boldsymbol{x}, \boldsymbol{y})$ toward 1, while also exerting much less interference on the optimization of the reasoning ability. Thus, our framework is a fundamentally different but more general method.

## D  THE LENGTH BIAS IN IMPLICIT REWARD

Here, we present the trend of the cumulative implicit reward values ($\log \frac{\pi_{\boldsymbol{\theta}}(\boldsymbol{y}_{<i}|\boldsymbol{x})}{\pi_{\text{ref}}(\boldsymbol{y}_{<i}|\boldsymbol{x})}$ where $\pi_{\text{ref}}$ is Qwen2.5-7B-Base) across 32 reasoning trajectories sampled from Open-Reasoner-Zero on an AIME2024 problem, showing how they vary with the increasing trajectory lengths. As illustrated in Figure 6, the curves of all samples exhibit a positive correlation between the implicit reward and the number of tokens, and longer trajectories tend to yield higher final implicit reward scores, indicating a strong length bias in implicit reward. Since incorrect solutions are generally longer than correct ones in reasoning tasks (Hassid et al., 2025), implicit reward is therefore not a reliable indicator of the relative quality of reasoning paths at test time.

## E  STATISTICS OF $\log \pi_{\text{REF}}(z_c|\boldsymbol{x}, \boldsymbol{y})$

We present the mean and standard deviation of $-\log \pi_{\text{ref}}(z_c|\boldsymbol{x}, \boldsymbol{y})$ computed over 300 input-output pairs. The reference model $\pi_{\text{ref}}$ is chosen as either Qwen2.5-7B-Base or OctoThinker-3B-Short-Base, and the evaluation is performed under two different choices of $z_c$ for each reference model (one common token and one unused special token): "Yes" and "`<vision_start>`" for Qwen2.5-7B-Base, "Yes" and "`<|reserved_special_token_0|>`" for OctoThinker-3B-Short-Base. The results in Figure 7 indicates that $-\log \pi_{\text{ref}}(z_c|\boldsymbol{x}, \boldsymbol{y})$ remains nearly constant and extremely small, with only a low standard deviation across different $\boldsymbol{x}$ and $\boldsymbol{y}$. Thus, we can consider $\log \pi_{\text{ref}}(z_c|\boldsymbol{x}, \boldsymbol{y})$ as a constant when calculating the last-token self-rewarding scores, which effectively reduces the computational cost by half.

Furthermore, in the term $\log \pi_{\text{ref}}(z_c|\boldsymbol{x}, \boldsymbol{y})$, the sequence $\boldsymbol{y}$ is generated by the policy model, whose parameters evolve during training. To ensure that our simplification continues to hold throughout the optimization process, we report the dynamics of both the mean and the standard deviation of $-\log \pi_{\text{ref}}(z_c|\boldsymbol{x}, \boldsymbol{y})$ over the course of training. Specifically, we calculate the mean and standard deviation of $-\log \pi_{\text{ref}}(z_c|\boldsymbol{x}, \boldsymbol{y})$ based on the 300 generated samples from the policy model at training steps 0, 100, 200, ..., and 1000. We conduct evaluations by taking Qwen2.5-7B-Base as the

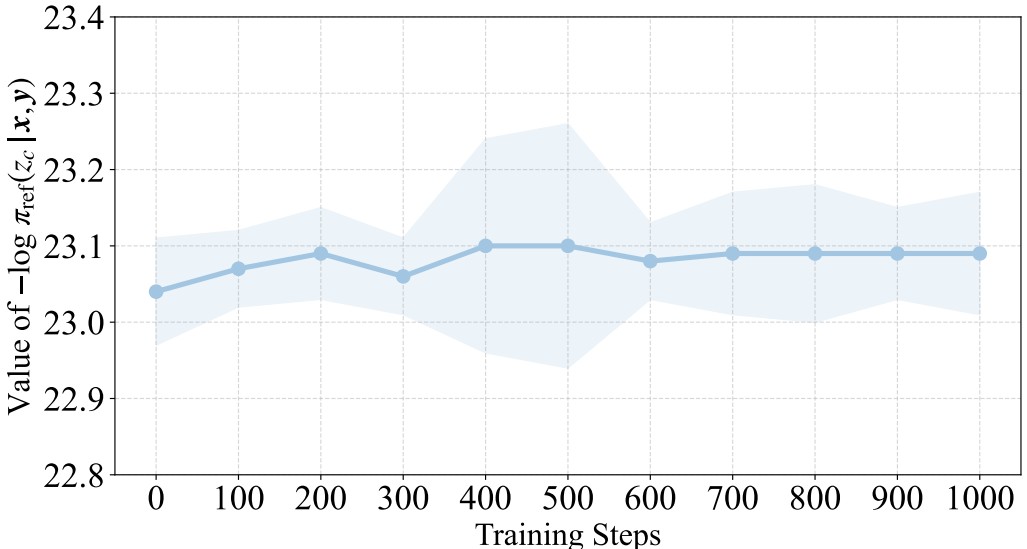

Figure 8: The dynamics of mean and standard deviation of $-\log \pi_{\text{ref}}(z_c|\boldsymbol{x}, \boldsymbol{y})$ during policy model's training.

---

**Algorithm 1: LaSeR**: Reinforcement Learning with Last-Token Self-Rewarding

---

**Input:** Initial policy model $\pi_{\boldsymbol{\theta}}$, prompts $D$, verifier $r_v$, warm-up hyper-parameters $w_r$ and $w_{sr}$,
coefficient $\beta_v$, pre-specified token $z_c$, pre-calculated $c_{\text{ref}} = \mathbb{E}_{(\boldsymbol{x}, \boldsymbol{y})}[\log \pi_{\text{ref}}(z_c|\boldsymbol{x}, \boldsymbol{y})]$

**for** *Step $s = 1, \cdots, S$* **do**

    1. Set $\pi_{old} \leftarrow \pi_{\boldsymbol{\theta}}$;

    2. Sample batch prompts $D_s$ from $D$;

    3. Generate solutions $\{\boldsymbol{y}^i\}_{i=1}^{K}$ for each $\boldsymbol{x} \in D_s$;

    4. Calculate verifier-based rewards and advantages (e.g., Eq. (4)), calculate RL loss;

    5. If $s \geq w_r$, calculate last-token self-rewarding loss based on Eq. (14) and add it to RL loss;

    6. If $s \geq w_{sr}$, calculate self-rewarding-based advantages and perform advantage integration
      based on Eq. (15);

    7. Update the policy model $\pi_{\boldsymbol{\theta}}$ using any RL algorithm with integrated loss and advantages;

**end**

**Output:** $\pi_{\boldsymbol{\theta}}$

---

base model and "`<vision_start>`" as the specified special token $z_c$. The dynamics are presented in Figure 8. As shown, **the mean of** $-\log \pi_{\text{ref}}(z_c|\boldsymbol{x}, \boldsymbol{y})$ **remains stable throughout the policy model updates**, which supports our motivation and confirms the feasibility of approximating it as a pre-computed constant in our method to improve efficiency.

## F    FULL ALGORITHM

We display the full procedure of our method in Algorithm 1.

## G    DETAILED TRAINING SETTINGS

We use verl (Sheng et al., 2024) as our RL training framework. The basic training hyper-parameters in both GRPO training and LaSeR training for each model are put in Table 4, and the newly introduced training hyper-parameters for LaSeR are put in Table 5. The number of optimization steps is 1000 for Qwen2.5-7B-Base and OctoThinker-3B-Short-Base, and 500 for Open-Reasoner-Zero-7B. In RL, a reasoning reward of 1.0 is given if the final answer and the answer format are both correct; otherwise, it is 0.0. In our method, the reasoning warm-up is performed for Qwen2.5-7B-Base and OctoThinker-

Table 4: Basic training hyper-parameters of both GRPO and LaSeR.

| Hyper-parameter | Value |
|---|---|
| Train Batch Size | 128 |
| Micro Batch Size | 128 |
| Rollout $n$ | 8 |
| Maximum Prompt Length | 2048 |
| Maximum Response Length | 8192 |
| Temperature | 1.0 |
| Top $p$ | 1.0 |
| LR | $1 \times 10^{-6}$ |
| KL Coefficient | 0.0 |

Table 5: Unique training hyper-parameters of LaSeR.

| Hyper-parameter | Value |
|---|---|
| Coefficient $\beta_v$ | 0.1 |
| Loss Weight $\alpha$ | 0.1 |
| Self-Rewarding Adv. Weight $\tau$ | 0.1 |
| Reasoning Warm-Up Steps | 200 |
| Self-Rewarding Warm-Up Steps | 200 |

Table 6: Comparison of verification F1 scores between LaSeR (self-rewarding) and external reward models (Qwen2.5-Math-7B-PRM800K, Qwen2.5-Math-PRM-7B, and Qwen2.5-Math-RM-72B) on responses generated by different policy models.

| Method | MATH500 | AMC23 | AIME24 | AIME25 | Olym. | Avg. |
|---|---|---|---|---|---|---|
| *Generator: OctoThinker-3B-Short-LaSeR* | | | | | | |
| Qwen2.5-Math-7B-PRM800K (7B RM) | 77.0 | 68.9 | - | - | 68.5 | 71.5 |
| Qwen2.5-Math-PRM-7B (7B RM) | 80.9 | 63.5 | - | - | 64.1 | 69.5 |
| Qwen2.5-Math-RM-72B (72B RM) | **89.2** | **71.7** | - | - | 72.9 | **77.9** |
| LaSeR (3B Self-Rewarding) | 73.6 | 70.2 | - | - | **73.6** | 72.5 |
| *Generator: Qwen2.5-7B-Laser* | | | | | | |
| Qwen2.5-Math-7B-PRM800K (7B RM) | 59.4 | 52.7 | 58.8 | 53.8 | 52.0 | 55.3 |
| Qwen2.5-Math-PRM-7B (7B RM) | 82.5 | 79.2 | 75.1 | 72.3 | 77.8 | 77.4 |
| Qwen2.5-Math-RM-72B (72B RM) | **87.8** | 80.7 | **81.3** | **74.8** | 75.4 | **80.0** |
| LaSeR (7B Self-Rewarding) | 83.2 | **82.5** | 79.6 | 74.3 | **78.3** | 79.6 |
| *Generator: Open-Reasoner-Zero-7B-LaSeR* | | | | | | |
| Qwen2.5-Math-7B-PRM800K (7B RM) | 56.3 | 42.5 | 51.4 | 50.8 | 38.5 | 47.9 |
| Qwen2.5-Math-PRM-7B (7B RM) | 86.0 | 79.6 | 70.8 | 67.3 | 76.0 | 75.9 |
| Qwen2.5-Math-RM-72B (72B RM) | 86.8 | 79.4 | **71.0** | 71.4 | 75.5 | 76.8 |
| LaSeR (7B Self-Rewarding) | **87.2** | 79.7 | 64.6 | **77.7** | **78.7** | **77.6** |

3B-Short-Base only, and the self-rewarding warm-up is perform for all models. The number of reasoning warm-up steps is set to 200 for both Qwen2.5-7B-Base and OctoThinker-3B-Short-Base, and the number of self-rewarding warm-up steps is 200 across all models.

# H COMPARISON OF VERIFICATION PERFORMANCE BETWEEN LASER AND ADVANCED EXTERNAL VERIFIERS

Here, we display the comparison results of verification F1 scores between LaSeR and three advanced external reward models (Qwen2.5-Math-7B-PRM800K (Zhang et al., 2025), Qwen2.5-Math-PRM-7B (Zhang et al., 2025), and Qwen2.5-Math-RM-72B (Yang et al., 2024)) on evaluating the solutions generated by the different reinforced models, i.e., OctoThinker-3B-Short-LaSeR, Qwen2.5-7B-LaSeR, and Open-Reasoner-Zero-7B-LaSeR. The full results in Table 6 show that LaSeR outperforms equally sized state-of-the-art external verifiers in assessing the model's own solutions, and even matches the verification performance of a 72B reward model, demonstrating its non-trivial effectiveness in enhancing self-rewarding capability.

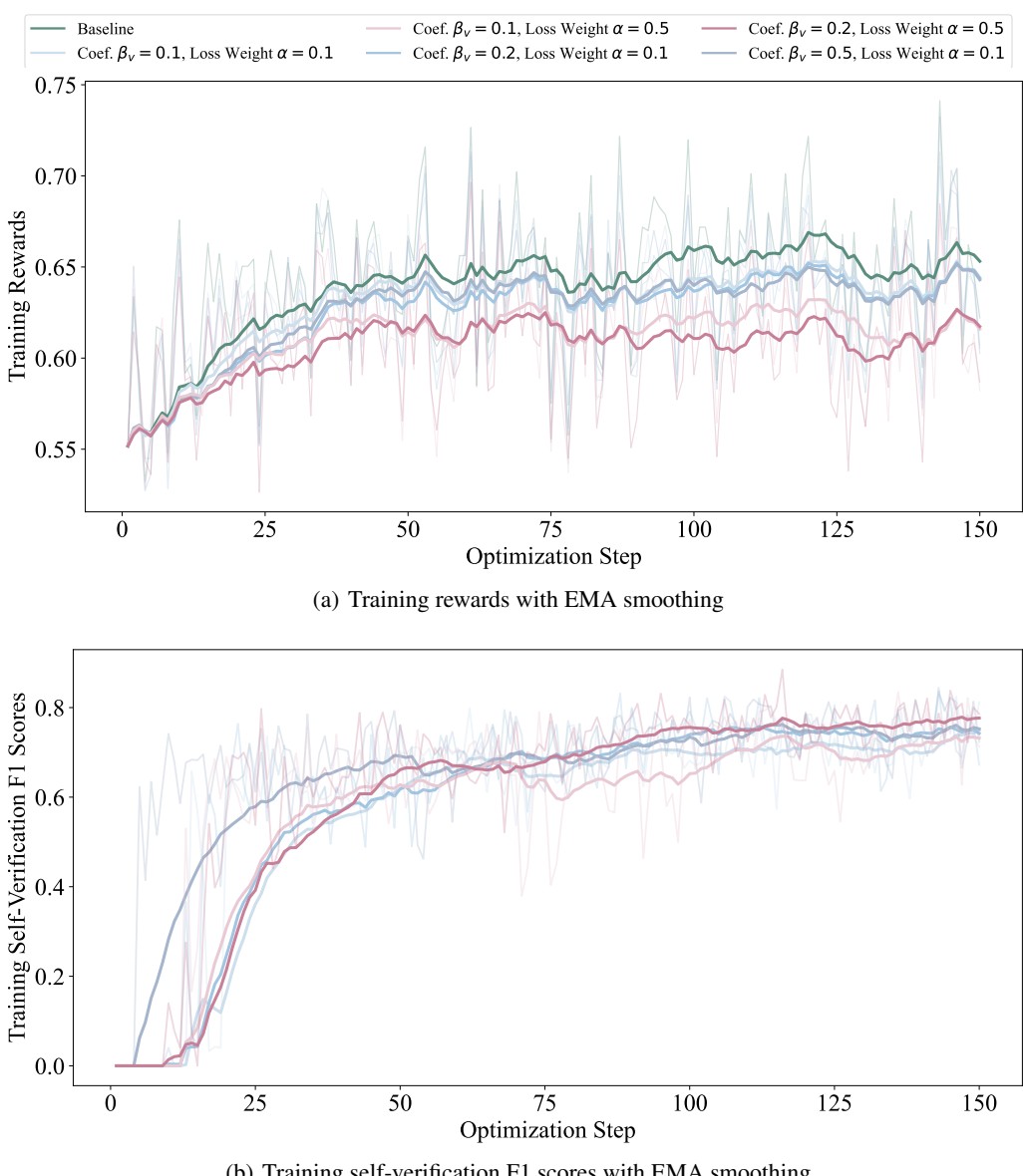

(a) Training rewards with EMA smoothing

(b) Training self-verification F1 scores with EMA smoothing

Figure 9: The curves of training rewards and training self-verification F1 scores under different combinations of hyper-parameters with EMA smoothing (EMA coef.=0.9).

# I  ABLATION STUDIES ON SELF-REWARDING HYPER-PARAMETERS

Here, we display the curves (with Exponential Moving Average (EMA) smoothing) of training rewards and training self-verification F1 scores of our method under different choices of coefficient $\beta_v$ and self-rewarding MSE loss weight $\alpha$. The experiments are conducted on Open-Reasoner-Zero-7B, which help to skip the reasoning warm-up phase compared with using Qwen2.5-7B-Base and OctoThinker-3B-Short-Base, while the results are similar in other two base models ater reasoning warm-up. The dynamics of training rewards and training self-verification F1 scores are displayed in Figure 9. As we can see, assigning a larger weight $\alpha$ to the last-token self-rewarding loss has a more detrimental impact on the model's reasoning capabilities. On the other hand, the coefficient $\beta_v$ has little impact on optimizing the self-rewarding scores, as long as it remains within a reasonable range (0.1 ~ 0.5). However, much smaller values of $\beta_v$ can impair the model's reasoning capability, as indicated by the analysis in the end of Section 3.4. For example, when $\beta_v = 0.05$, we should

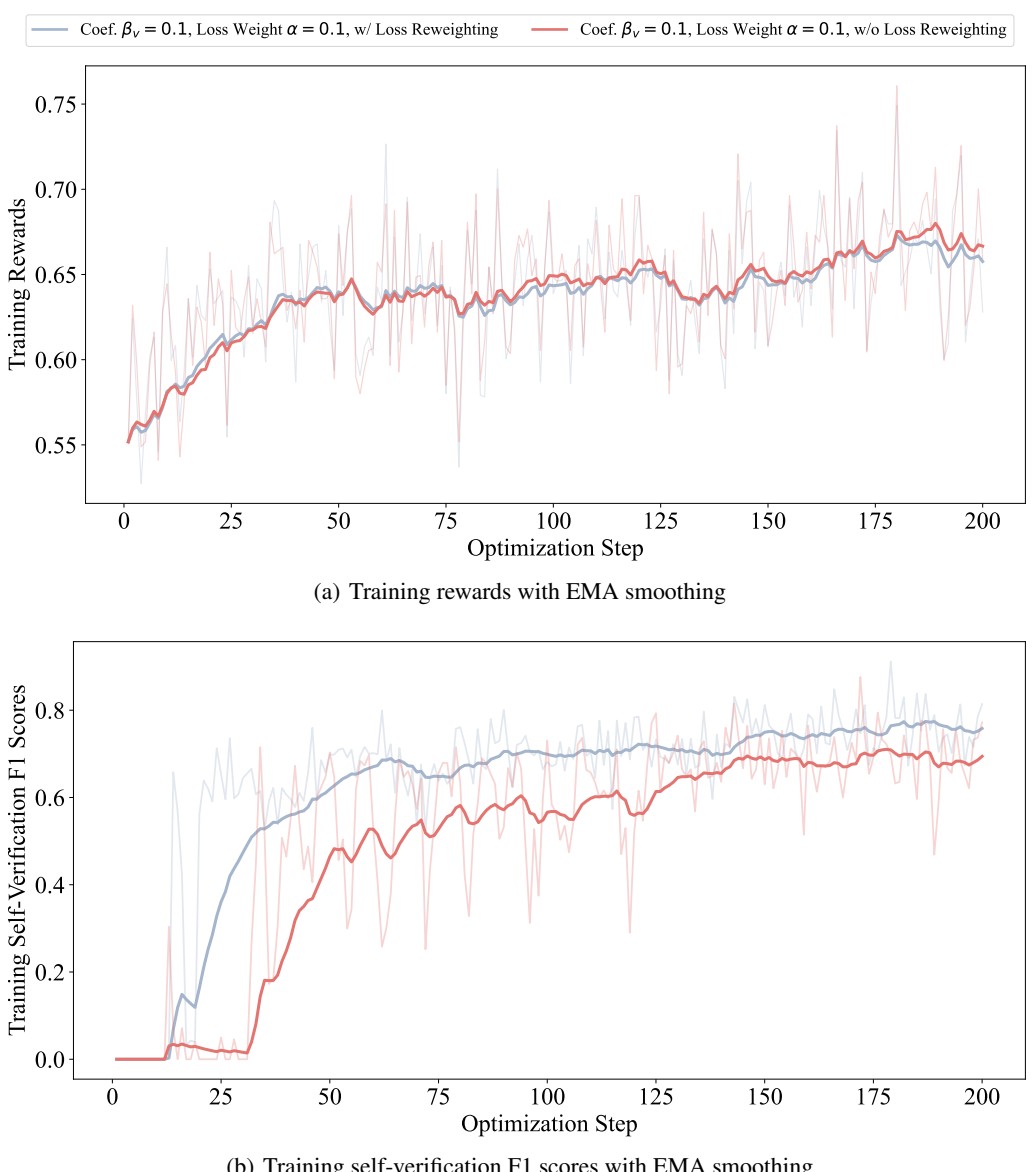

(a) Training rewards with EMA smoothing

(b) Training self-verification F1 scores with EMA smoothing

Figure 10: The curves of training rewards and training self-verification F1 scores of our method with and without class-level loss re-weighting practice (EMA coef.=0.9).

have $\pi_{\boldsymbol{\theta}}(z_c|\boldsymbol{x}, \boldsymbol{y}) = e^{-3} \approx 0.05$ under $\pi_{\text{ref}}(z_c|\boldsymbol{x}, \boldsymbol{y}) = e^{-23}$ and $r_v(\boldsymbol{x}, \boldsymbol{y}) = 1$, then the large value of $\pi_{\boldsymbol{\theta}}(z_c|\boldsymbol{x}, \boldsymbol{y})$ causes large interference with the optimization of reasoning capability. In our main experiments, we choose $(\beta_v, \alpha) = (0.1, 0.1)$.

## J   THE EFFECT OF CLASS-LEVEL RE-WEIGHTING ON THE BALANCED SELF-VERIFICATION CAPABILITY

We present the training dynamics of our method on Open-Reasoner-Zero-7B, with and without class-level loss re-weighting, in Figure 10 for comparison. As shown, applying loss re-weighting leads to a more balanced self-verification performance by mitigating the bias toward the majority class with larger sample size, while still maintaining high reasoning accuracy.

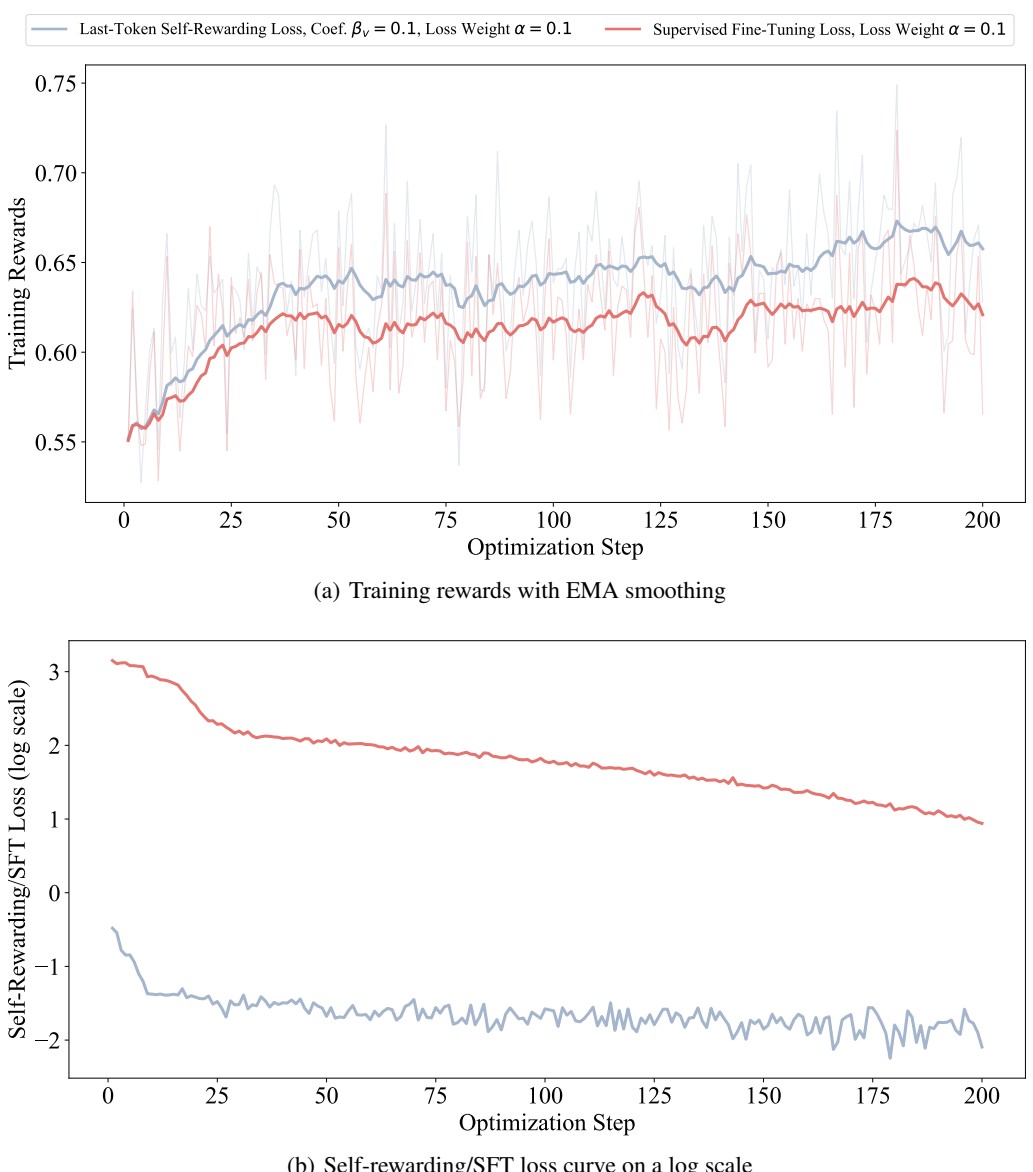

(a) Training rewards with EMA smoothing

(b) Self-rewarding/SFT loss curve on a log scale

Figure 11: The comparison of the training dynamics between the last-token self-rewarding loss and the SFT loss.

## K  COMPARISON BETWEEN LAST-TOKEN SELF-REWARDING LOSS AND SUPERVISED FINE-TUNING LOSS

Following the discussion in Section 3.4, we compare the training performance of our introduced last-token self-rewarding loss with the supervised fine-tuning (SFT) loss on Open-Reasoner-Zero-7B. The training dynamics are shown in Figure 11. As observed, applying the SFT loss to optimize the self-rewarding capability causes substantial interference with the optimization of reasoning capability, leading to a marked degradation in training rewards. Moreover, the SFT loss degrades extremely slowly, indicating that directly driving $\pi_{\boldsymbol{\theta}}(z_c|\boldsymbol{x}, \boldsymbol{y})$ from 0 to 1 for $r_v(\boldsymbol{x}, \boldsymbol{y}) = 1$ is inherently difficult. However, our method only requires fitting $\pi_{\boldsymbol{\theta}}(z_c|\boldsymbol{x}, \boldsymbol{y})$ to $\exp(1/\beta_v) \cdot \pi_{\mathrm{ref}}(z_c|\boldsymbol{x}, \boldsymbol{y})$ for $r_v(\boldsymbol{x}, \boldsymbol{y}) = 1$, which is considerably easier and introduces much less interference.

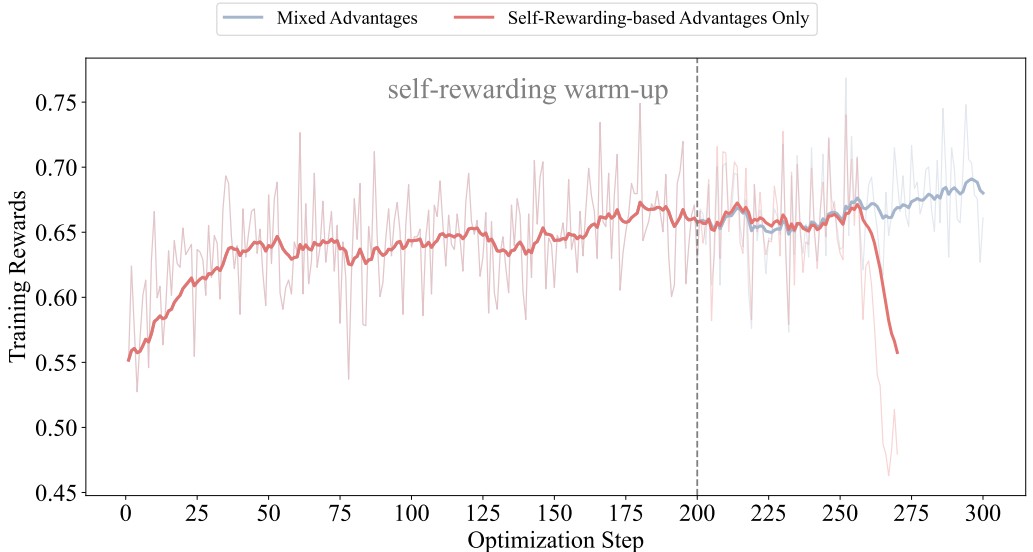

Figure 12: The ablation of only using self-rewarding scores for RL.

Table 7: The the range between the minimum and maximum self-rewarding scores over all solutions in each test set.

| Model | MATH500 | AMC23 | AIME24 | AIME25 | OlympiadBench |
|---|---|---|---|---|---|
| OT-3B-Short-LaSeR | [0.000, 1.010] | [-0.031, 1.001] | [-0.023, 0.858] | [-0.061, 0.722] | [-0.036, 1.013] |
| Qwen2.5-7B-LaSeR | [-0.021, 1.031] | [0.013, 1.020] | [-0.052, 1.010] | [-0.041, 0.986] | [-0.056, 1.029] |
| ORZ-7B-LaSeR | [-0.005, 1.013] | [-0.085, 1.005] | [-0.035, 0.998] | [-0.031, 0.963] | [-0.036, 1.016] |

## L   ABLATION OF ONLY USING SELF-REWARDING SCORES FOR RL

In our method, after self-rewarding warm-up, we we incorporate self-rewarding–based advantages into the verifier-based advantages to provide more fine-grained learning signals. Here, we explore the effect of only using the self-rewarding scores for RL. Specifically, taking Open-Reasoner-Zero-7B as the base model, we take the checkpoint after 200 training steps with our method as the starting point. We then continue RL training using only the self-rewarding score as the optimization signal for reasoning ability (while still using the rule-based rewards to optimize the self-rewarding scores). The dynamics of training rewards are shown in Figure 12. We observe that after an additional 60 steps, the training collapses. This indicates that using only self-rewarding scores for RL easily leads to training instability, and should be complemented with the rule-based rewards.

## M   THE VALUE RANGE OF THE OPTIMIZED SELF-REWARDING SCORE

In Table 7, we show the the range between the minimum and maximum self-rewarding scores over all solutions in each test set of each model. The results validate that after optimization, the self-rewarding score naturally falls within the target interval $[0, 1]$.

## N   DETAILED SELF-VERIFICATION RESULTS

We report the detailed self-verification results of each model on self-generated solutions across all benchmarks in Table 8, including both overall accuracy and F1 score. Our method consistently yields significant improvements in model's self-rewarding and self-verification capabilities, while incurring only minimal additional computational cost.

Table 8: Detailed self-verification results.

| Method | MATH500 | | AMC23 | | AIME24 | | AIME25 | | Olym. | |
|---|---|---|---|---|---|---|---|---|---|---|
| | Acc. | F1 | Acc. | F1 | Acc. | F1 | Acc. | F1 | Acc. | F1 |
| ***OctoThinker-3B-Short-Base*** | | | | | | | | | | |
| Base | 60.2 | 22.3 | 52.3 | 11.2 | - | - | - | - | 62.0 | 13.7 |
| GRPO | 58.2 | 56.9 | 66.7 | 47.3 | - | - | - | - | 66.4 | 48.8 |
| LaSeR | 77.0 | 73.6 | 77.3 | 70.2 | - | - | - | - | 80.3 | 73.6 |
| *- SWA* | 81.0 | 80.4 | 84.1 | 70.9 | - | - | - | - | 83.5 | 66.0 |
| ***Qwen2.5-7B-Base*** | | | | | | | | | | |
| Base | 45.0 | 36.4 | 30.7 | 30.8 | 24.5 | 27.6 | 28.2 | 32.9 | 33.8 | 36.9 |
| GRPO | 76.5 | 54.6 | 61.1 | 59.7 | 60.4 | 36.6 | 72.5 | 41.5 | 54.6 | 53.5 |
| LaSeR | 88.0 | 83.2 | 81.5 | 82.5 | 92.2 | 79.6 | 90.5 | 74.3 | 79.5 | 78.3 |
| *- SWA* | 87.8 | 79.7 | 79.6 | 80.2 | 94.3 | 81.3 | 92.2 | 74.9 | 83.9 | 83.3 |
| ***Open-Reasoner-Zero-7B*** | | | | | | | | | | |
| Base | 79.6 | 26.7 | 66.6 | 51.3 | 39.6 | 45.9 | 47.6 | 55.2 | 55.2 | 37.5 |
| GRPO | 52.9 | 57.1 | 50.9 | 44.8 | 66.9 | 14.6 | 78.9 | 28.1 | 54.7 | 49.5 |
| LaSeR | 90.1 | 87.2 | 77.7 | 79.7 | 87.2 | 64.6 | 92.8 | 77.7 | 80.1 | 78.7 |
| *- SWA* | 89.0 | 87.5 | 76.2 | 77.7 | 87.7 | 63.3 | 93.6 | 77.3 | 80.2 | 77.9 |

Table 9: The average AUROC of reward scores on each benchmark. In calculation, we discard query groups in which all solutions are either correct or incorrect, as such cases provide no meaningful signal for assessing the effectiveness of the self-rewarding scores. To ensure reliability, we sample 32 times for each query. The generator is Open-Reasoner-Zero-7B-LaSeR.

| Method | MATH500 | AMC23 | AIME24 | AIME25 | Olym. | Avg. |
|---|---|---|---|---|---|---|
| Qwen2.5-Math-RM-72B (72B RM) | **0.77** | **0.73** | 0.71 | 0.60 | 0.65 | 0.69 |
| LaSeR (7B Self-Rewarding) | 0.72 | 0.68 | **0.77** | **0.74** | **0.67** | **0.72** |

Table 10: ECE of reward scores on each benchmark (lower is better). To ensure reliability, we sample 32 times for each query. The generator is Open-Reasoner-Zero-7B-LaSeR.

| Method | MATH500 | AMC23 | AIME24 | AIME25 | Olym. | Avg. |
|---|---|---|---|---|---|---|
| Qwen2.5-Math-RM-72B (72B RM) | 0.178 | **0.152** | 0.346 | 0.384 | 0.194 | 0.251 |
| LaSeR (7B Self-Rewarding) | **0.077** | 0.162 | **0.072** | **0.071** | **0.066** | **0.090** |

Table 11: The Spearman Rank Correlation between self-rewarding scores and response lengths on Open-Reasoner-Zero-7B-LaSeR. To ensure reliability, we sample 32 times for each query.

| | MATH500 | AMC23 | AIME24 | AIME25 | Olym. | Avg. |
|---|---|---|---|---|---|---|
| Spearman Rank Corr. | -0.26 | -0.34 | -0.58 | -0.52 | -0.40 | -0.42 |

## O   FULL RESULTS OF THE CALIBRATION ANALYSIS OF SELF-REWARDING SCORES

Here, we display the full results of the calibration analysis of self-rewarding scores made in Section 5.1. We first display the results of average AUROC score (the probability that, within the same query group, a correct solution receives a higher self-rewarding score than an incorrect one) of the self-rewarding scores with respect to correctness in Table 9. The results demonstrate the high discriminative power of the self-rewarding score between correct and incorrect solutions. The comparison results of Expected Calibration Error (ECE, the discrepancy between a self-rewarding score and its empirical accuracy) in Table 10 demonstrate that our method achieves better confidence calibration. Finally, we show the average Spearman Rank Correlation (the direction and strength of the monotonic relationship between two variables by computing the correlation between their ranked values) between self-rewarding scores and response lengths in Table 11. The results reveal that shorter responses generally tend to

Table 12: Cross-model verification results. Each row corresponds to the average verification F1 score of a given verifier, evaluated on solution sets generated by different generators, as indexed by the columns.

| Verifier ↓ | Generator → | | |
|---|---|---|---|
| | OT-3B-Short-GRPO | Qwen2.5-7B-GRPO | ORZ-7B-LaSeR |
| OT-3B-Short-GRPO | 51.0 | 42.7 | 43.6 |
| Qwen2.5-7B-GRPO | 56.4 | 49.2 | 46.4 |
| ORZ-7B-LaSeR | **69.2** | **77.5** | **77.6** |

Table 13: Results of applying our method within PPO framework on Open-Reasoner-Zero-7B.

| Method | Reasoning Accuracy | | | | | | Self-Verification F1 Score | | | | | |
|---|---|---|---|---|---|---|---|---|---|---|---|---|
| | MATH-500 | AMC-23 | AIME-24 | AIME-25 | Olym.-Bench | Avg. | MATH-500 | AMC-23 | AIME-24 | AIME-25 | Olym.-Bench | Avg. |
| ***Open-Reasoner-Zero-7B*** | | | | | | | | | | | | |
| Base | 81.9 | 60.3 | 15.6 | 15.1 | 46.9 | 44.0 | 26.7 | 51.3 | 45.9 | 55.2 | 37.5 | 43.3 |
| PPO | 82.1 | 58.6 | **16.7** | 14.8 | 47.0 | 43.8 | 51.8 | 43.7 | 14.7 | 15.4 | 40.5 | 33.2 |
| LaSeR[PPO] | **82.9** | **61.2** | 15.4 | **15.7** | **47.9** | **44.6** | **85.6** | **80.0** | **76.1** | **81.6** | **78.3** | **80.3** |

receive higher self-rewarding scores, which is consistent with recent findings (Hassid et al., 2025; Yang et al., 2025c) that shorter CoTs on the same query are generally more preferable.

## P    CROSS-MODEL VERIFICATION RESULTS

Here, we explore the cross-model verification performance. In specific, we compare the verification F1 scores of three models—OctoThinker-3B-Short-GRPO, Open-Reasoner-Zero-7B-GRPO, and Open-Reasoner-Zero-7B-LaSeR—evaluated on each other's responses. The results are in Table 12. Each value in the table represents the average F1 score across all five benchmarks. Surprisingly, we find that our method not only enables effective self-rewarding, but also **achieves high accuracy and F1 when evaluating the CoTs generated by other models, demonstrating strong generalization ability**.

## Q    RESULTS OF APPLYING LaSeR WITHIN PPO

Here, we conduct additional experiments using PPO as the RL algorithm and explore the potential of applying our method within the PPO framework to validate the generality of our method. We conduct experiments on Open-Reasoner-Zero-7B. The overall reasoning and self-verification results are presented in Table 13, and the inference-time scaling results are in Figure 13. We use "LaSeR[PPO]" to denote our method based on the PPO framework. As shown, **our method also achieves strong effectiveness in enhancing the reasoning, self-verification, and inference-time scaling performance of the policy model within the PPO framework**.

## R    THE GENERALIZABILITY OF LaSeR TO GENERAL REASONING DOMAIN

We conduct additional experiments to explore the generalizability of our method to general reasoning domain. We use a filtered version (Yu et al., 2025b) of WebInstruct-verified dataset (Ma et al., 2025), and conduct RL experiments on Qwen3-4B-Base (Yang et al., 2025a). We use the `general-verifier-1.5B` model from Ma et al. (2025) as the model-based verifier and adopt GRPO as the RL algorithm. The basic training and testing hyper-parameters for experiments on WebInstruct-verified are the same as those in Table 4 and Table 5, while the number of optimization steps here is 800. The simplified constant of the reference log-probability $c_{ref}$ is $-23.0$. For our method, we do not perform the advantage integration strategy here. The reason is that we observe the self-verification F1 score of our method during training is relatively low in the general reasoning setting (only between

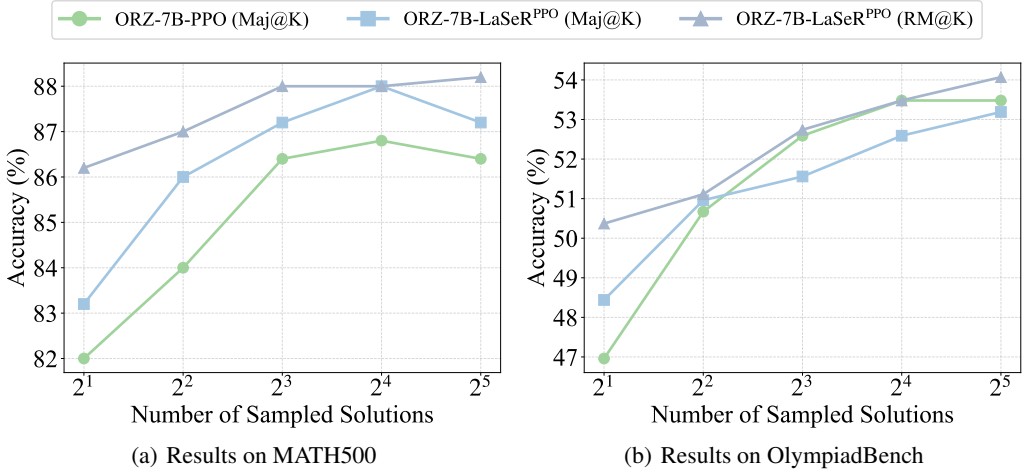

Figure 13: The majority voting (Maj@K) and weighted majority voting (RM@K) results of Open-Reasoner-Zero-7B-PPO (ORZ-7B-PPO) and Open-Reasoner-Zero-7B-LaSeR[PPO] (ORZ-7B-LaSeR[PPO]) on MATH500 and OlympiadBench.

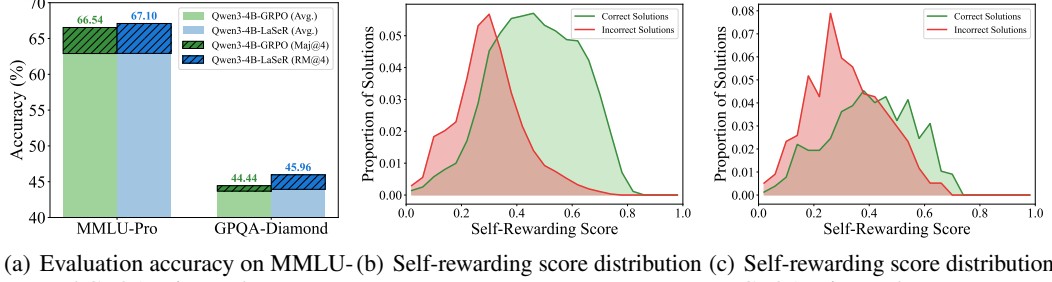

(a) Evaluation accuracy on MMLU-Pro and GPQA-Diamond    (b) Self-rewarding score distribution on MMLU-Pro    (c) Self-rewarding score distribution on GPQA-Diamond

Figure 14: The generalizability of LaSeR on general reasoning tasks.

65% and 70%, and the self-rewarding score distributions in the test sets shown in Figure 14(b) and Figure 14(c) also reveal this phenomenon). This leads to large noise in the self-rewarding-based advantage estimation, and consequently, the integration of self-rewarding-based advantages results in performance degradation. We conduct evaluations on two general reasoning benchmarks: MMLU-Pro (Wang et al., 2024b) and GPQA-Diamond (Rein et al., 2024). We sample 4 solutions per problem on each dataset for each model, and calculate both the average accuracy and the (weighted) majority voting accuracy.

We have several findings: (1) First, from Figure 14(a), we observe that jointly optimizing the self-rewarding capability does not impact the model's general reasoning ability, allowing the policy model to achieve comparable average reasoning accuracy to the baseline. (2) However, as mentioned above, the optimized self-rewarding score on general reasoning tasks does not achieve the high accuracy seen in math reasoning tasks. We can see that the self-rewarding score distributions for correct and incorrect solutions on MMLU-Pro exhibit certain overlap, and the distinction further diminishes on the more challenging benchmark GPQA-Diamond. We speculate that two factors may contribute to this: (a) The model's general reasoning ability is inherently weaker than its math reasoning ability, which limits the upper bound of its self-rewarding capabilities in the general reasoning domain. (b) The model-based verifier used in the experiment (general-verifier-1.5B) has limited verification ability, resulting in high noise in the reasoning rewards, which in turn affects the optimization of the self-rewarding capability. (3) Though not perfect, **the optimized self-rewarding scores can still provide useful signals during inference time, leading to better weighted majority voting results**. To examine this effect, we analyze the distribution of self-rewarding scores across solutions generated

Table 14: The average self-rewarding AUROC score of Qwen3-4B-LaSeR to assess how well the self-rewarding score separates correct from incorrect solutions. In calculation, we discard query groups in which all solutions are either correct or incorrect, as such cases provide no meaningful signal for assessing the effectiveness of the self-rewarding scores. To ensure reliability, here, we sample 32 times for each query. For computational efficiency, we evaluate on 1000 samples (Yu et al., 2025b) from MMLU-Pro.

|  | MMLU-Pro-1000 | GPQA-D |
|---|---|---|
| AUROC | 0.61 | 0.58 |

for the same query. We calculate and report the average AUROC score, which is the probability that a correct solution sampled from the same query group (solutions generated for the same query) receives a higher self-rewarding score than an incorrect one. The results are reported in Table 14. We find that the AUROC scores are around 0.6, indicating that **within the same query group, the self-rewarding scores assigned to correct solutions are generally higher than those assigned to incorrect ones**. This reveals that, though not perfect, the self-rewarding signal can still effectively support solution re-ranking, thereby improving the performance of weighted majority voting.

## S  FURTHER REDUCTION OR INCREASE OF SELF-REWARDING COST

In this section, we discuss two additional variants of LaSeR for future work. In the current method, we calculate the last-token self-rewarding score based on the next-token log-probability distribution of the "<EOS>" token, requiring one additional token inference compared with standard inference. One potential way to further reduce the inference cost of LaSeR is to derive the last-token self-rewarding score directly from the predicted log-probability of pre-specified token $z_c$ at the "<EOS>" token position. Specifically, let $y_T$ denote the "<EOS>" token in $\boldsymbol{y}$. Then, the reduced last-token self-rewarding score can be defined as $r_s = \beta_v \log \pi_{\boldsymbol{\theta}}(z_c|\boldsymbol{x}, \boldsymbol{y}_{<T}) - \beta_v c_{\text{ref}}$, as we have observed that $\pi_{\text{ref}}(z_c|\boldsymbol{x}, \boldsymbol{y}_{<T})$ remains nearly constant across $(\boldsymbol{x}, \boldsymbol{y})$ (e.g., approximately $e^{-28}$ for Qwen2.5-7B-Base). In this case, **we can achieve ideally zero additional inference cost for self-rewarding compared with standard generation** by directly calculating the self-rewarding score from the log-probability distribution at the "<EOS>" token position, without requiring any extra token inference. In theory, this works because setting a relatively large $\beta_v$ still yields a small value of $\pi_{\boldsymbol{\theta}}(z_c|\boldsymbol{x}, \boldsymbol{y}_{<T})$ (e.g., $\pi_{\boldsymbol{\theta}}(z_c|\boldsymbol{x}, \boldsymbol{y}_{<T}) = e^{-18}$ when $\beta_v = 0.1$ and $c_{\text{ref}} = -28$), thereby allowing $\pi_{\boldsymbol{\theta}}(\text{<EOS>}|\boldsymbol{x}, \boldsymbol{y}_{<T})$ to still dominate the probability mass. However, although the probability is very low, we observe that the generator may still select $z_c$ at the end of the sequence in few cases during training, which can adversely affect training stability as the generator continues to generate after $z_c$. One straight-forward solution may be to set the sampling hyper-parameter $top\_p$ to a value less than 1.0. Future work can further investigate advanced strategies to make the above adjustment more principled and robust.

While reducing the self-rewarding cost improves efficiency, an alternative is **to increase the inference cost in exchange for stronger self-rewarding capability**. That is, instead of computing the self-rewarding score based on the log-probability distribution of a single token only, we can increase the number of additional inference tokens by calculating it over $M$ tokens as $r_s = \beta_v \sum_{m=1}^{M} \log \pi_{\boldsymbol{\theta}}(z_c|\boldsymbol{x}, \boldsymbol{y}, \underbrace{z_c, \cdots, z_c}_{m-1 \text{ times}})) - M\beta_v c_{\text{ref}}$. It is a promising direction for future research to explore whether increasing the number of additional inference tokens can yield positive inference-time scaling effect for latent self-rewarding capability.

## T  PROMPT TEMPLATES

We show the training, evaluation and self-verification prompt templates used in our experiments in the end.

---

**Training and Evaluation Prompt Template for OctoThinker-3B-Short-Base**

---

<bos_token> A conversation between User and Assistant. The user asks a question, and the Assistant solves it. The assistant first thinks about the reasoning process in the mind and then provides the user with the answer.
User: You must put your answer inside \boxed{} and Your final answer will be extracted automatically by the \boxed{} tag.
{question}
Assistant:

---

**Training Prompt Template for Qwen2.5-7B-Base**

---

<bos_token> A conversation between User and Assistant. The User asks a question, and the Assistant solves it. The Assistant first thinks about the reasoning process in the mind and then provides the User with the answer. The reasoning process is enclosed within <think> </think> and answer is enclosed within <answer> </answer> tags, respectively, i.e., <think> reasoning process here </think> <answer> answer here </answer>.
User: You must put your answer inside <answer> </answer> tags, i.e., <answer> answer here </answer>. And your final answer will be extracted automatically by the \boxed{} tag.
This is the problem:
{question}
Assistant: <think>

---

**Zero-Shot Evaluation Prompt Template for Qwen2.5-7B-Base**

---

< |im start| >system
You are a helpful assistant.< |im end| >
< |im start| >user
{question}
Please reason step by step, and put your final answer within \boxed{}.< |im end| >
< |im start| >assistant

---

**Training and Evaluation Prompt Template for Open-Reasoner-Zero-7B**

---

A conversation between User and Assistant. The User asks a question, and the Assistant solves it. The Assistant first thinks about the reasoning process in the mind and then provides the User with the answer. The reasoning process is enclosed within <think> </think> and answer is enclosed within <answer> </answer> tags, respectively, i.e., <think> reasoning process here </think> <answer> answer here </answer>.
User: You must put your answer inside <answer> </answer> tags, i.e., <answer> answer here </answer>. And your final answer will be extracted automatically by the \boxed{} tag.
{question}
Assistant: <think>

---

**Training and Evaluation Prompt Template for Qwen3-4B-Base**

---

< |im_start| >user
{question}
Please reason step by step, and put your final answer within \boxed{}.< |im_end| >
< |im_start| >assistant

---

**Prompt Template for Self-Verification (Modified from Liu et al. (2025a))**

Below you are presented with a question and a tentative response. Your task is to evaluate the response and assign a rating to the response based on the following clear criteria:
Rating Criteria:
1. Missing final answer, or incorrect response with the wrong final answer: assign \boxed{0}.
2. Correct response with the correct final answer: assign \boxed{1}.
### Question Begin ###
{question}
### Question End ###
### Response Begin ###
{response}
### Response End ###
First provide your evaluation process, then clearly state your final rating value enclosed in \boxed{} at the end.

---

