# OpenReview forum: "LaSeR: Reinforcement Learning with Last-Token Self-Rewarding"
_ICLR.cc/2026/Conference — ICLR 2026 Poster_

### Official Review · Reviewer_Z5vT · 2025-10-29

**Soundness:** 3
**Presentation:** 2
**Contribution:** 3
**Rating:** 6
**Confidence:** 3

**Summary:**

This paper introduces a self-verification calculation method. It expands on the implicit reward and provides an expression through approximation to eliminate the expensive partition function evaluation, enabling the model to estimate the reward through a simple single token probability. The derived expression simplifies reward estimation and allows the model to self-evaluate its reasoning confidence. Empirical results show improvements on mathematical reasoning benchmarks, particularly in F1 scores for self-verification tasks.

**Strengths:**

1. Approximating implicit rewards through single-token probability is elegant and efficient. Theoretically it is novel. The derivation can be extended to future works for self-verification since it is simple enough to apply.

2. There are empirical performance improvements in self-verification F1 scores on math reasoning tasks. Table 1 F1 scores show clear improvements over the self-verification.

**Weaknesses:**

1. The key observation of negligible $log Z$ does not have a strict guarantee that it will be approximately 0 when the model weights are changed. It would be beneficial to add additional analysis like Figure 4 for the post training model.

2. It is unclear why the self-reward is added on top of the true reward instead of being used directly for optimization. This design choice should be justified. The claim could be stronger to have an ablation study with only self-verification as the reward signal, separating it out from GRPO advantage.

3. Equation 9 would benefit to have more explanations.

4. Appendix F appears to evaluate only on the model’s own outputs. It would be informative to test cross-model generalization, for example, using Qwen2.5-7B-Base to verify outputs from Open-Reasoner-Zero-7B.

**Questions:**

n/a

---

> ### Author Response · Authors · 2025-11-24
> **Author Response (Part 1)**
>
> We sincerely thank you for your positive feedback and helpful suggestions. We are glad that you think our method is novel, elegant and efficient. We are encouraged that you think our method achieves remarkable self-verification performance. To address your remaining questions, we make the following detailed response.
>
> -------
>
> **Q1:** Regarding the stability of $\log Z(\boldsymbol{x}, \boldsymbol{y})$ across model updates.
>
> **A1:** Thank you for your insightful question. Following your suggestion, we visualize the dynamics of both the mean and standard deviation of $-log \pi_{ref}(z_c|\boldsymbol{x}, \boldsymbol{y})$ (where $y$ is generated by the policy model) throughout the policy model's training process. Specifically, we calculate the mean and standard deviation of $-log \pi_{ref}(z_c|\boldsymbol{x}, \boldsymbol{y})$ based on the 300 generated samples from the policy model at training steps 0, 100, 200, …, and 1000. We conduct evaluations by taking ```Qwen2.5-7B-Base``` as the base model. The dynamics are presented in the following table. As shown, **the mean of $-log \pi_{ref}(z_c|\boldsymbol{x}, \boldsymbol{y})$ remains stable throughout the policy model updates**, indicating that $\pi_{ref}(z_c|\boldsymbol{x}, \boldsymbol{y})$ remains near zero during training and $\log Z(\boldsymbol{x}, \boldsymbol{y})$ can be reliably approximated to 0. Furthermore, the ablation results in Table 2 in the submission also helps to validate the feasibility of this practice.
>
>
> Table 1. The dynamics of mean and standard deviation of $-log \pi_{ref}(z_c|\boldsymbol{x}, \boldsymbol{y})$ during policy model's training.
>
> |Checkpoint|0th Step| 100th Step| 200th Step|300th Step| 400th Step| 500th Step| 600th Step|700th Step| 800th Step|900th Step| 1000th Step|
> |:---:|:---:|:---:|:---:|:---:|:---:|:---:|:---:|:---:|:---:|:---:|:---:|
> | mean $\pm$ std |23.04 $\pm$ 0.07 | 23.07 $\pm$ 0.05 |  23.09 $\pm$ 0.06  | 23.06  $\pm$  0.05 | 23.10  $\pm$ 0.14 | 23.10  $\pm$ 0.16 | 23.08 $\pm$ 0.05 | 23.09 $\pm$ 0.08 |  23.09 $\pm$  0.09 | 23.09 $\pm$ 0.06 |  23.09 $\pm$ 0.08 |
>
>
> ------
>
> **Q2:** Regarding the ablation that only uses the self-rewarding scores for RL.
>
> **A2:** We follow your suggestion to explore the effect of only using the self-rewarding scores for RL. Specifically, taking ```Open-Reasoner-Zero-7B``` as the base model, we take the checkpoint after 200 training steps with our method as the starting point. We then continue RL training using only the self-rewarding score as the optimization signal for reasoning ability (while still using the rule-based verifier's rewards to optimize the self-rewarding scores). We observe that **after an additional 60 steps, the training collapses**, evidenced by a sharp increase in training response length and a rapid drop in training rewards. This indicates that using only self-rewarding scores for RL easily leads to training instability, and should be complemented with the rule-based verifier's rewards. We will put the training dynamics in the revision for ablation studies.

---

> > ### Author Response · Authors · 2025-11-24
> > **Part 2**
> >
> > **Q3:** Regarding further explanations on Equation 9.
> >
> > **A3:** Thank you for your question. In the following, we make detailed derivations based on Eq. (9) to obtain the general form of our self-rewarding MSE loss.
> >
> > We note that in Eq. (7), the rewards for correct and incorrect verifications can be any two values, $\hat{r}_c$ and $\hat{r}_i$ (though 0/1 is the common choice)​. Under this general definition, following the same derivation as in Eq.(9), we obtain $Z(\boldsymbol{x},\boldsymbol{y})= exp(\frac{\hat{r}\_i}{\beta_v})$, which indicates $\log Z(\boldsymbol{x},\boldsymbol{y})=\frac{\hat{r}\_i}{\beta_v}$ (still a constant but not necessarily to be 0). Therefore, $\log Z(\boldsymbol{x},\boldsymbol{y}) = 0$ in Eq. (9) is a special case when we define $\hat{r}
> > _{i}=0$.
> >
> >
> > Following above, the optimal solution to the original RL target can be reduced to a general form
> > $$
> > \hat{r}(\boldsymbol{x},\boldsymbol{y},\boldsymbol{z}) = \beta_{v} \log [\pi_{\boldsymbol{\theta}}(\boldsymbol{z}|\boldsymbol{x},\boldsymbol{y}) / \pi_{ref}(\boldsymbol{z}|\boldsymbol{x},\boldsymbol{y})] + \hat{r}\_{i}.
> > $$
> > Now, the true verification reward when the model verifies a solution as correct is
> > $$
> > \hat{r}(\boldsymbol{x},\boldsymbol{y},z_{c}) = \hat{r}\_{c}I_{r_{v}(\boldsymbol{x},\boldsymbol{y})=1} + \hat{r}\_{i} I_{r_{v}(\boldsymbol{x},\boldsymbol{y})=0} =  \beta_{v} \log [\pi_{\boldsymbol{\theta}}(z_{c}|\boldsymbol{x},\boldsymbol{y}) / \pi_{ref}(z_{c}|\boldsymbol{x},\boldsymbol{y})] + \hat{r}\_{i}.
> > $$
> > Note that the reasoning rewards $r_{v}(\boldsymbol{x},\boldsymbol{y})$ defined in Eq. (2) can also be two arbitrary values, but we simplified it to $(1,0)$ and this will not affect our derivation. In this general framework, our self-rewarding MSE loss is
> > $$
> > L=E_{\boldsymbol{x}\sim D, \boldsymbol{y}\sim \pi_{g}(\cdot | x)}\left( \beta_{v} \log [ \pi_{\boldsymbol{\theta}}(z_{c}|\boldsymbol{x},\boldsymbol{y}) / \pi_{ref}(z_{c}|\boldsymbol{x},\boldsymbol{y})] +\hat{r}\_i - (\hat{r}\_{c} I_{r_{v}(\boldsymbol{x},\boldsymbol{y})=1} + \hat{r}\_{i} I_{r_{v}(\boldsymbol{x},\boldsymbol{y})=0}) \right)^{2}.
> > $$
> >
> >
> > We will incorporate the above derivation and discussion into the revision to strengthen our paper.
> >
> > ----
> >
> > **Q4:** Regarding the cross-model verification results.
> >
> > **A4:** We follow your suggestion to explore the cross-model verification performance. In specific, we compare the verification F1 scores of three models—```OctoThinker-3B-Short-GRPO```, ```Open-Reasoner-Zero-7B-GRPO```, and ```Open-Reasoner-Zero-7B-LaSeR```—evaluated on each other's responses. The results are in the table below. Each value in the table represents the average F1 score across all benchmarks. Surprisingly, we find that our method not only enables effective self-rewarding, but **our model also achieves high accuracy and F1 when evaluating the CoTs generated by other models, demonstrating strong generalization ability**.
> >
> >
> > Table 2. Cross-model verification results. Each column corresponds to the average verification F1 score of a given verifier, evaluated on solution sets generated by different generators, as indexed by the rows.
> >
> > ||OctoThinker-3B-Short-GRPO|Qwen2.5-7B-GRPO|Open-Reasoner-Zero-7B-LaSeR |
> > |---|:------:|:------:|:------:|
> > |Generator| |
> > |OctoThinker-3B-Short-GRPO|  51.0| 56.4 | **69.2** |
> > |Qwen2.5-7B-GRPO | 42.7  | 49.2 | **77.5**|
> > |Open-Reasoner-Zero-7B-LaSeR |  43.6  | 46.4 | **77.6**|
> >
> > -------
> >
> > We hope the above response addresses your questions. We are incorporating the new results and discussions into the revision, and will upload the revision once it is ready.

---

### Official Review · Reviewer_9vmN · 2025-10-30

**Soundness:** 2
**Presentation:** 3
**Contribution:** 3
**Rating:** 4
**Confidence:** 3

**Summary:**

LaSeR proposes a way to make LLMs “self-reward” by showing that, under its RL objective, the verifier reward can be approximated by a last-token self-rewarding score, the KL-scaled difference between the policy’s next-token log-probability on a pre-specified token at the final solution token and a precomputed constant. It augments RLVR with a simple MSE loss aligning this score to verifier rewards and then uses the learned score as an auxiliary signal at train and test time. The experiment shows this method can improve model reasoning ability and self-verification ability for test-time scaling.

**Strengths:**

- Detailed analysis of the proposed approach, combining both theoretical and empirical analysis.
- LaSeR is lightweight & efficient. Adding an MSE term aligning “last-token self-reward” to verifier rewards; computing the score costs at most one extra token.
- Weighting majority-vote by the learned self-reward scores improves solution selection at test time.

**Weaknesses:**

- Potential issue in the partition function expansion (Eq. 9)
  - The paper defines the verification reward as **one-hot** over the *ground-truth* label: reward = 1 **only** when the predicted token equals the correct label (e.g., ($z_+\in{z_c,z_i}$) depending on the task’s truth), and 0 otherwise (Eq. 7).
  - However, when expanding the partition function (Z(x,y)) in Eq. (8), Eq. (9), it adds both $(z_c)$ and $(z_i)$ terms with the factor $(\exp(1/\beta_v))$, treating *both* as rewarded. I think only the *true* label ($z_+$) should carry reward 1 while the other label $(z_-)$ gets reward 0. Then
 $  Z(x,y)=\sum_{z\notin{z_c,z_i}}\pi_{\text{ref}}(z|x,y) e^{0}+\pi_{\text{ref}}(z_+|x,y)e^{1/\beta_v}+\pi_{\text{ref}}(z_-|x,y) e^{0}
  =1+\pi_{\text{ref}}(z_+|x,y)\big(e^{1/\beta_v}-1\big)$
- 0.5 threshold on a non-probability score (L318). The score ($r_s=\beta_v\log\pi_\theta(z_c\mid x,y)-\beta_v c_{\text{ref}}$) is a scaled/shifted log-probability (unbounded, not a probability). Thresholding it at **0.5** is arbitrary—changing ($\beta_v$) or ($c_{\text{ref}}$) shifts the decision boundary. It also ignores the competing label ($z_i$) even though verification is binary.
- The experiment is limited. Only base model and GRPO compared, and some setup (Open-Reasoner-Zero-7B x AIME25) shows limited advantage or is just not better than baseline at all. As the margin is so small, the statistical interval should be included.

**Questions:**

- As the primary focus is to integrate with GRPO using verifiable reward, why MSE loss is used for Eq. 12 instead of BCE loss?
- How is the weighted majority voting for RM@K?

---

> ### Author Response · Authors · 2025-11-24
> **Author Response (Part 1)**
>
> We sincerely thank you for your careful reviews and helpful questions. We are glad that you think our work includes detailed theoretical and empirical analysis. We are encouraged that you think our method is lightweight and highly efficient, and archives great effectiveness on enhancing the inference-time scaling performance. We make the following response to address your questions.
>
> **Q1:** Regarding the minor derivation error in the partition function expansion.
>
> **A1:** We thank you for your careful review. We agree that there is a minor derivation error in the partition function expansion, as you pointed out; however, **this does not affect the final outcome**.
>
> Let's look back into Eq. (9), since we have validated that $\pi_{ref}(z_{c}|\boldsymbol{x},\boldsymbol{y}) \approx 0$ and $\pi_{ref}(z_{i}|\boldsymbol{x},\boldsymbol{y}) \approx 0$,
> $$
> Z(\boldsymbol{x},\boldsymbol{y})=\sum_{\boldsymbol{z}} \pi_{ref}(\boldsymbol{z}|\boldsymbol{x},\boldsymbol{y}) \exp (\frac{1}{\beta_{v}}\hat{r}(\boldsymbol{x},\boldsymbol{y},\boldsymbol{z}))=\sum_{\boldsymbol{z} \notin \{z_{c},z_{i} \}} \pi_{ref}(\boldsymbol{z}|\boldsymbol{x},\boldsymbol{y}) \exp (\frac{1}{\beta_{v}}\hat{r}(\boldsymbol{x},\boldsymbol{y},\boldsymbol{z}))\\ + \pi_{ref}(z_{c}|\boldsymbol{x},\boldsymbol{y}) \exp (\frac{1}{\beta_{v}}\hat{r}(\boldsymbol{x},\boldsymbol{y},z_{c}))  + \pi_{ref}(z_{i}|\boldsymbol{x},\boldsymbol{y}) \exp (\frac{1}{\beta_{v}}\hat{r}(\boldsymbol{x},\boldsymbol{y},z_{i})) \\ = (1 -  \pi_{ref}(z_{c}|\boldsymbol{x},\boldsymbol{y}) -\pi_{ref}(z_{i}|\boldsymbol{x},\boldsymbol{y})) \exp (0) +\pi_{ref}(z_{c}|\boldsymbol{x},\boldsymbol{y})
> \exp (\frac{1}{\beta_{v}}I_{r_{v}(\boldsymbol{x},\boldsymbol{y})=1})  + \pi_{ref}(z_{i}|\boldsymbol{x},\boldsymbol{y}) \exp (\frac{1}{\beta_{v}}(1-I_{r_{v}(\boldsymbol{x},\boldsymbol{y})=1}))
> \approx 1  + 0  + 0 =1
> \implies \log Z(\boldsymbol{x}, \boldsymbol{y}) \approx 0.
> $$
>
> We will fix this error in the revision. Furthermore, in **A5**, we derive a generalized form of our self-rewarding MSE loss by broadening the verification reward in Eq. (7) from the special 0/1 case to arbitrary scalar values. We show that, under this generalization, $\log Z(\boldsymbol{x}, \boldsymbol{y})$ still remains a constant (not necessarily equal to zero). Moreover, this generalized formulation naturally explains our choice of the MSE loss over the BCE loss, thereby addressing your question (**Q5**) on this point. Please refer to **A5** for the complete discussion.
>
> -----
>
>
> **Q2:** Regarding the interpretation of the self-rewarding score.
>
> **A2:** We believe there might be some misunderstandings on our self-rewarding socre, and we would like to clarify that **our self-rewarding score is a bounded and valid probabilistic measure, and that 0.5 serves as a meaningful decision threshold**.
>
> First, **the choices of $\beta_v$ and $c_{ref}$ are fixed and consistent across both training and testing**. That is, the same values used during training are also applied at test time, and we should not alter them arbitrarily, which prevents any scale shift in the self-rewarding scores. Since during training our MSE objective drives the self-rewarding score toward 0 or 1, a threshold of 0.5 naturally becomes a meaningful decision boundary, consistent with the very common practice. Then, we provide evidence in **Table 4** in our following response **A6** to demonstrate that **the optimized self-rewarding scores naturally fall within [0, 1], further validating both the boundedness and effectiveness of our self-scoring mechanism**.
>
>
> Second, we explain the reason why we only need the probability of $\pi_{ref}(z_{c}|\boldsymbol{x},\boldsymbol{y})$ for verification. The main point is that **our method differs from the standard binary classification setting trained with a BCE loss. Instead of mapping two labels to two tokens, our MSE loss requires the model to predict a single probability value, which only depends on the probability associated with one token**. The theoretical basis for this design lies in Eq. (11), from which we can see that the self-rewarding score derived from $\pi_{ref}(z_{c}|\boldsymbol{x},\boldsymbol{y})$ directly reflects the correctness of the verification. More discussion can be found in **A5**, where we address your question regarding the difference between our MSE loss and the BCE loss.

---

> > ### Author Response · Authors · 2025-11-24
> > **Part 2**
> >
> > **Q3:** Regarding the reasoning accuracy gains.
> >
> > **A3:** First, we would like to clarify that the primary scope and contribution of our work is to introduce a novel and highly efficient algorithm that **jointly optimizing the model's self-rewarding capability *at nearly zero additional cost*, while theoretically *not interfering with the optimization of the model's reasoning ability***. The latter follows directly from the discussion in the second paragraph of Section 3.4: our method induces only a negligible change to the output probability distribution of the final token ($\pi_{\boldsymbol{\theta}}(z_c|\boldsymbol{x},\boldsymbol{y})$ changes from $e^{-23}$ to $e^{-13}$), and therefore introduces virtually no conflict with the optimization of the model‘s reasoning ability.
> >
> > Then, we observe that introducing the self-rewarding-based advantages (SWA) yields performance gains in **10 out of 13** evaluation cases, and consistently improves the average reasoning performance of different policy models. Compared with sparse rule-based rewards, the self-rewarding scores provide a more fine-grained and informative training signal, which effectively guides optimization. This observation is also consistent with recent findings in the literature [1].
> >
> > Finally, follow the above clarification, we would like to point out that  **the primary advantage of our method over the baseline lies in the fact that well-calibrated self-rewarding scores enable stronger inference-time scaling performance through weighted majority voting**. Besides the inference-time scaling results on Open-Reasoner-Zero-7B in Figure 2 in the submission, here, we display the full results on 3 backbones to further **validate the effectiveness of our method on enhancing model's inference-time scaling performance**.
> >
> > Table 1. Full inference-time scaling results via (weighted) majority voting.
> >
> >
> > || ||MATH500 | | | | | OlympiadBench| |  |
> > |--------|:------:|:------:|:------:|:------:|:----:|:------:|:------:|:----:|:------:|:----:|
> > |K|2 |  4 | 8 | 16 | 32 | 2 | 4 | 8 | 16 | 32 |
> > |***Base Model: OctoThinker-3B-Short-Base***|
> > |OctoThinker-3B-Short-GRPO (Maj@K)| 49.0| 53.8| 56.8| 57.0| 57.4 |17.8| 19.3| 20.2| 20.4| 21.5|
> > |OctoThinker-3B-Short-LaSeR (Maj@K)| 51.6| 56.8| 61.2|63.0|64.0| 18.9| 21.8| 24.3| **25.6**| 25.9|
> > |OctoThinker-3B-Short-LaSeR (RM@K)| **52.2** | **58.0** | **61.8** | **63.8** | **64.2** | **20.3** | **23.3**| **24.9** | 25.3| **26.7**|
> > |***Base Model: Qwen2.5-7B-Base***|
> > |Qwen2.5-7B-GRPO (Maj@K)| 79.6| 82.2|84.2|84.4|84.8| 43.6| 47.0| 48.6 |50.2 | 50.4 |
> > |Qwen2.5-7B-LaSeR (Maj@K)|81.2| 84.2| 85.2| **86.6**| **86.2** | 45.8 | 47.6| 49.3 | 50.7 | **51.1**|
> > |Qwen2.5-7B-LaSeR (RM@K)| **82.6**| **85.0**| **85.4**| **86.6**| 85.8| **46.7** | **48.3** |**49.5** | **51.1** | 50.8 |
> > |***Base Model: Open-Reasoner-Zero-7B***|
> > |Open-Reasoner-Zero-7B-7B-GRPO (Maj@K)| 82.8|85.0|**86.4**| **87.0** |87.8 | 47.7 | 51.9 | 52.3| 53.2| 53.3 |
> > |Open-Reasoner-Zero-7B-LaSeR (Maj@K)| 82.6| 83.8|84.6|85.8|86.2| 49.2| 51.9 | 54.2 | 55.0 | 54.7 |
> > |Open-Reasoner-Zero-7B-LaSeR (RM@K)| **84.2** | **85.4**|86.0|86.4|**88.0**  | **51.9** | **53.0** | **54.4** | **55.1** | **55.3**|
> >
> > ---
> >
> > [1] Tao, Leitian, et al. "Hybrid Reinforcement: When Reward Is Sparse, It's Better to Be Dense." arxiv 2025

---

> ### Author Response · Authors · 2025-11-24
> **Part 3**
>
> **Q4:** Regarding the comment "Only base model and GRPO are compared".
>
> **A4:** We would like to clarify that **our method can be seamlessly integrated with any RL framework** (e.g., GRPO, PPO, DAPO, etc.), jointly optimizing both the reasoning and self-rewarding capabilities of the policy model. In our submission, we mainly conduct experiments under GRPO. Here, we conduct additional experiments using PPO as the RL algorithm and explore the potential of applying our method within the PPO framework to validate the generality of our method. We conduct experiments on Open-Reasoner-Zero-7B (ORZ-7B), and the results are presented below. As shown, **our method also achieves strong effectiveness in enhancing the reasoning, self-verification, and inference-time scaling performance of the policy model within the PPO framework**.
>
> Table 2. Results of applying our method within PPO framework on ORZ-7B.
>
> || ||Reasoning Accuracy | | | | ||Self-Verification F1 Score| | | |
> |--------|:------:|:------:|:------:|:------:|:----:|:------:|:------:|:----:|:------:|:----:|:------:|:----:|
> ||MATH500 | AMC23 |AIME24|AIME25|Olym. |Avg. | MATH500 | AMC23 |AIME24|AIME25|Olym.|Avg.|
> |Base|  81.9 | 60.3 |15.6 |15.1 |46.9 | 44.0 |26.7| 51.3 |45.9 |55.2 |37.5| 43.3|
> |PPO| 82.1| 58.6  | **16.7** | 14.8 | 47.0| 43.8 |51.8|43.7| 14.7|15.4|40.5|33.2|
> |LaSeR (Ours)| **82.9** |  **61.2** | 15.4 | **15.7** | **47.9** | **44.6** | **85.6** | **80.0** | **76.1** | **81.6** | **78.3** | **80.3** |
>
>
> Table 3. Inference-time scaling results of PPO experiments.
>
> || ||MATH500 | | | | | OlympiadBench| |  |
> |--------|:------:|:------:|:------:|:------:|:----:|:------:|:------:|:----:|:------:|:----:|
> |K|2 |  4 | 8 | 16 | 32 | 2 | 4 | 8 | 16 | 32 |
> |PPO (Maj@K)| 82.0 | 84.0| 86.4 | 86.8 | 86.4 | 47.0 | 50.7| 52.6 |**53.5** | 53.5 |
> |LaSeR (Maj@K)| 83.2 | 86.0 | 87.2 | 88.0 |87.2 |48.4 | 51.0 | 51.6 | 52.6 | 53.2|
> |LaSeR (RM@K)| **86.2** | **87.0** | **88.0** | **88.0** | **88.2** | **50.4** | **51.1** |  **52.7** | **53.5** | **54.1** |

---

> ### Author Response · Authors · 2025-11-24
> **Part 4**
>
> **Q5:** Regarding the comparison between our introduced self-rewarding MSE loss and BCE loss.
>
>
> **A5:**  In the following, we first make detailed derivations to obtain the general form of our self-rewarding MSE loss. Then, we clarify why our method adopts the MSE loss instead of the BCE loss, and highlight that **the solution to the above SFT loss is in fact a special case of all available solutions in our framework**.
>
> We note that in Eq. (7), the rewards for correct and incorrect verifications can be any two values, $\hat{r}\_c$ and $\hat{r}\_i$ (though 0/1 is the common choice)​. Under this general definition, following the same derivation as in Eq.(9), we obtain $Z(\boldsymbol{x},\boldsymbol{y})= exp(\frac{\hat{r}\_i}{\beta_v})$, which indicates $\log Z(\boldsymbol{x},\boldsymbol{y})=\frac{\hat{r}\_i}{\beta_v}$ (still a constant but not necessarily to be 0). Then, the optimal solution to the original RL target can be reduced to a general form
> $$
> \hat{r}(\boldsymbol{x},\boldsymbol{y},\boldsymbol{z}) = \beta_{v} \log [\pi_{\boldsymbol{\theta}}(\boldsymbol{z}|\boldsymbol{x},\boldsymbol{y}) / \pi_{ref}(\boldsymbol{z}|\boldsymbol{x},\boldsymbol{y})] + \hat{r}\_{i}.
> $$
> That is,
> $$
> \hat{r}(\boldsymbol{x},\boldsymbol{y},\boldsymbol{z}) -  \hat{r}\_{i} = \beta_{v} \log [\pi_{\boldsymbol{\theta}}(\boldsymbol{z}|\boldsymbol{x},\boldsymbol{y}) / \pi_{ref}(\boldsymbol{z}|\boldsymbol{x},\boldsymbol{y})] .
> $$
>
> Now, **we can see that the left-hand side of the equation is no longer a 0/1 binary reward but rather two arbitrary scalar values. Therefore, it is natural to model this objective using an MSE loss rather than a BCE loss.**
>
> Following above, the true verification reward when the model verifies a solution as correct is
> $$
> \hat{r}(\boldsymbol{x},\boldsymbol{y},z_{c}) = \hat{r}\_{c}I_{r_{v}(\boldsymbol{x},\boldsymbol{y})=1} + \hat{r}\_{i} I_{r_{v}(\boldsymbol{x},\boldsymbol{y})=0} =  \beta_{v} \log [\pi_{\boldsymbol{\theta}}(z_{c}|\boldsymbol{x},\boldsymbol{y}) / \pi_{ref}(z_{c}|\boldsymbol{x},\boldsymbol{y})] + \hat{r}\_{i}.
> $$
> Note that the reasoning rewards $r_{v}(\boldsymbol{x},\boldsymbol{y})$ defined in Eq. (2) can also be two arbitrary values, but we simplified it to $(1,0)$ and this will not affect our derivation.
> This formulation indicates that the probability $\pi_{\boldsymbol{\theta}}(z_{c}|\boldsymbol{x},\boldsymbol{y})$ alone is sufficient to model the reward score.
> In this general framework, our self-rewarding MSE loss is
> $$
> L=E_{\boldsymbol{x}\sim D, \boldsymbol{y}\sim \pi_{g}(\cdot | x)}\left( \beta_{v} \log [ \pi_{\boldsymbol{\theta}}(z_{c}|\boldsymbol{x},\boldsymbol{y}) / \pi_{ref}(z_{c}|\boldsymbol{x},\boldsymbol{y})] +\hat{r}\_i - (\hat{r}\_{c}I_{r_{v}(\boldsymbol{x},\boldsymbol{y})=1} + \hat{r}\_{i} I_{r_{v}(\boldsymbol{x},\boldsymbol{y})=0}) \right)^{2}.
> $$
>
>
> Now, we show that the solution to the above SFT loss is in fact a special case of all available solutions in our framework. After optimization (i.e., the optimal solution to the MSE loss), we can see that when $r_v(\boldsymbol{x},\boldsymbol{y})=0$ (i.e., the solution is incorrect based on the deterministic verifier), $\pi_{\boldsymbol{\theta}}(z_c|\boldsymbol{x},\boldsymbol{y}) = \pi_{ref}(z_c|\boldsymbol{x},\boldsymbol{y}) $, which means the $\pi_{\boldsymbol{\theta}}(z_c|\boldsymbol{x},\boldsymbol{y})$ remains unchanged (almost near zero, consistent with the solution to SFT loss when $r_v(\boldsymbol{x},\boldsymbol{y})=0$). When $r_v(\boldsymbol{x},\boldsymbol{y})=1$ (i.e., the solution is correct based on the deterministic verifier), our method drives $\pi_{\boldsymbol{\theta}}(z_c|\boldsymbol{x},\boldsymbol{y})$ to fit $\pi_{ref}(z_c|\boldsymbol{x},\boldsymbol{y}) exp(\frac{\hat{r}\_{c}-\hat{r}\_{i}}{\beta_v}) $. As we can see, when $\beta_v$ continues to decrease, the optimized probability $\pi_{\boldsymbol{\theta}}(z_c|\boldsymbol{x},\boldsymbol{y})$ approaches 1, which is exactly the solution to the SFT loss. However, in our framework, **we can explicitly control the value of $\beta_v$, allowing $\pi_{\boldsymbol{\theta}}(z_c |\boldsymbol{x}, \boldsymbol{y})$ to converge to any desired target**. As shown in Figure 9, this controllability provides greater flexibility compared with directly forcing $\pi_{\boldsymbol{\theta}}(z_c |\boldsymbol{x}, \boldsymbol{y})$ toward 1, while also exerting much less interference on the optimization of the reasoning ability. Thus, **our framework is a fundamentally different but more general method**.

---

> > ### Author Response · Authors · 2025-11-24
> > **Part 5**
> >
> > **Q6:** Regarding the implementation of RM@K.
> >
> > **A6:** As described in Lines 316–319, during testing, each sampled solution for the same query receives a self-rewarding score $ r_{s}=\beta_{v} \log \pi_{\boldsymbol{\theta}}(z_{c}|\boldsymbol{x},\boldsymbol{y}) - \beta_{v} c_{ref}$. We then clip the self-rewarding scores into the interval [0, 1]. Notably, we point out that this clipping is optional: after optimization, the self-rewarding scores naturally fall within [0, 1] (see the table below), further validating both the boundedness and effectiveness of our self-scoring mechanism. Finally, we select the final answer with the greatest total sum of self-rewarding scores to achieve weighted majority voting (RM@K).
> >
> > Table 4. The the range between the minimum and maximum self-rewarding scores over all solutions in each test set.
> >
> >
> > |Model|MATH500|AMC23|AIME24|AIME25|OlympiadBench|
> > |:---|:------:|:------:|:------:|:------:|:------:|
> > |OctoThinker-3B-Short-LaSeR| [0.000, 1.010]|[-0.031, 1.001] |[-0.023, 0.858] | [-0.061, 0.722] | [-0.036, 1.013]|
> > |Qwen2.5-7B-LaSeR | [-0.021, 1.031]| [0.013, 1.020] | [-0.052, 1.010] | [-0.041, 0.986] |  [-0.056, 1.029] |
> > |Open-Reasoner-Zero-7B-LaSeR | [-0.005, 1.013]|  [-0.085, 1.005] |  [-0.035, 0.998]|[-0.031, 0.963] | [-0.036, 1.016]|
> >
> >
> > -----
> >
> >
> > We hope the above response addresses your questions. We are incorporating the new results and discussions into the revision, and will upload the revision once it is ready.

---

### Official Review · Reviewer_xmH7 · 2025-10-31

**Soundness:** 3
**Presentation:** 2
**Contribution:** 3
**Rating:** 4
**Confidence:** 4

**Summary:**

The paper presents **LaSeR**, a reinforcement learning algorithm that augments the standard RLVR objective with an additional Mean Squared Error (MSE) loss aligning last-token self-reward scores with verifier-based rewards, enabling joint optimization of reasoning and self-verification. By approximating the reference log-probability as a constant, applying class-balanced re-weighting, and integrating the self-reward into advantage estimation, LaSeR provides an efficient improvement over vanilla GRPO with minimal computational overhead. Experiments on multiple mathematical reasoning benchmarks and three base models show consistent gains in both reasoning accuracy and self-verification performance, as well as the effectiveness of self-verification on test-time scaling and its generalization to other domains.

**Strengths:**

- **Theoretical grounding and originality:** The derivation of the self-reward term is well-grounded in established RL literature, with a clear and rigorous justification for approximating the partition term as zero using both theoretical analysis and empirical evidence. While related works exist, LaSeR presents a coherent and novel formulation of self-verification that links last-token modeling with verifier-based rewards.
- **Efficiency and practicality:** LaSeR introduces minimal training and inference overhead compared to prior self-verification or RLVR methods. The approximation of the reference log-probability as a constant further simplifies implementation without sacrificing performance, making the approach highly practical for large-scale use. Additional training techniques grounded in empirical observation further enhance the method’s engineering value.
- **Comprehensive experimentation:** The study conducts extensive experiments across three base models and five mathematical reasoning benchmarks, with an additional evaluation on a science dataset. Detailed ablations on hyperparameters and the constant reference log-probability assumption validate the design choices. Together, these experiments strengthen the reliability of the conclusions.

**Weaknesses:**

- **[Motivation]** The motivation could be more clearly articulated. The paper frames LaSeR mainly as an efficiency improvement over existing joint reasoning–verification training paradigms but does not sufficiently justify *why* integrating these two abilities within a single model is necessary, instead of using a separate generator–verifier system. A clearer discussion of the conceptual or practical advantages of joint training would strengthen the motivation and emphasize the broader significance of the approach.
- **[Significance]** The reasoning accuracy gains over the GRPO baseline are modest, and the performance difference between LaSeR and LaSeR w/ SWA is small. This limits the overall perceived significance and makes it difficult to assess the impact of the SWA module.
- **[Experimentation]** The experimental analysis is somewhat limited. For instance, the generalization evaluation on GPQA reports only inference-time scaling results, without presenting zero-shot reasoning or verification performance, making it difficult to fully assess out-of-domain robustness. The comparison between SFT and MSE reward curves also omits the final performance results, leaving the claimed advantage of MSE insufficiently supported. Furthermore, the paper does not investigate why reasoning ability improves under LaSeR or whether the auxiliary objective influences the model’s reasoning dynamics or solution style. A deeper analysis in these aspects would offer clearer insight into the mechanisms driving the observed improvements.
- **[Presentation]** Section 3 could be more condensed, as some of the detailed derivations and techniques may be moved to the appendix to improve readability and maintain focus on the key contributions.

**Questions:**

- How is the weighted majority voting implemented?
- How are the special tokens chosen for representing correctness (e.g., `<vision_start>`)? Are they selected randomly?
- Figure 6 shows that many correct solutions have self-reward scores below 0.5, even though the mean score for correct cases is higher than for incorrect ones. Does this suggest that the self-reward signal works better for re-ranking rather than binary verification? It would be helpful if the authors could clarify this behavior or provide more results regarding this point.

---

> ### Author Response · Authors · 2025-11-24
> **Author Response (Part 1)**
>
> We sincerely thank you for your helpful comments and questions. We are encouraged that you think our paper presents a clear and novel problem formulation, and our theoretical derivation is rigorous and well-grounded. We are glad that you believe that our method shows great efficiency and practicality with comprehensive experiments. To address your remaining concerns, we make the following detailed response.
>
> ---
>
> **Q1:** Regarding the clearer illustration of the motivation of joint reasoning-verification training.
>
> **A1:** Thank you for your insightful question. Optimizing both reasoning and self-verification within a single model offers two key advantages over a separate generator–verifier framework.
>
> (1) **Self-verification is a crucial capability for sustainable model evolution.**
> Equipping the model with internal self-verification enables it to autonomously evaluate, reflect, and iterate on its own outputs when interacting with testing environments or unlabeled data [1]. Such self-driven improvement is essential for achieving scalable and continual capability growth. Recent studies [2,3] also show that enhancing self-verification can positively influence reasoning quality: the model's capacity for self-reflection tends to generalize and contribute to more reliable reasoning trajectories.
>
> (2) **Jointly optimizing the two abilities significantly improves training and inference efficiency.**
> During training, the model no longer requires a separately trained verifier, which saves substantial computational overhead. During inference, the system does not need to deploy and run an additional verifier model, making the approach far more practical and resource-efficient.
>
> We will add the above illustration into the revision to make the motivation clearer.
>
> ---
>
> **Q2:** Regarding the reasoning accuracy gains compared with baseline and our method without self-rewarding-based advantage integration.
>
> **A2:** First, we would like to clarify that the primary scope and contribution of our work is to introduce **a novel and highly efficient algorithm that jointly optimizing the model's self-rewarding capability *at nearly zero additional cost*, while theoretically *not interfering with the optimization of the model's reasoning ability***. The latter follows directly from the discussion in the second paragraph of Section 3.4: our method induces only a negligible change to the output probability distribution of the final token ($\pi_{\boldsymbol{\theta}}(z_c|\boldsymbol{x},\boldsymbol{y})$ changes from $e^{-23}$ to $e^{-13}$), and therefore introduces virtually no conflict with the optimization of the model‘s reasoning ability.
>
> Then, we observe that introducing the self-rewarding-based advantages (SWA) yields performance gains in **10 out of 13** evaluation cases, and consistently improves the average reasoning performance of different policy models. Compared with sparse rule-based rewards, the self-rewarding scores provide a more fine-grained and informative training signal, which effectively guides optimization. This observation is also consistent with recent findings in the literature [4].
>
> Finally, follow the above clarification, we would like to point out that  **the primary advantage of our method over the baseline lies in the fact that well-calibrated self-rewarding scores enable stronger inference-time scaling performance through weighted majority voting**. Besides the inference-time scaling results on Open-Reasoner-Zero-7B in Figure 2 in the submission, here, we display the full results on 3 backbones to further **validate the effectiveness of our method on enhancing model's inference-time scaling performance**.
>
>
> ---
>
> [1] Zuo, Yuxin, et al. "Ttrl: Test-time reinforcement learning." arxiv 2025
>
> [2] Liu, Xiaoyuan, et al. "Trust, But Verify: A Self-Verification Approach to Reinforcement Learning with Verifiable Rewards." arxiv 2025
>
> [3] Zha, Kaiwen, et al. "RL Tango: Reinforcing Generator and Verifier Together for Language Reasoning." NeurIPS 2025
>
> [4] Tao, Leitian, et al. "Hybrid Reinforcement: When Reward Is Sparse, It's Better to Be Dense."  arxiv 2025

---

> > ### Author Response · Authors · 2025-11-24
> > **Part 2**
> >
> > Table 1. Full inference-time scaling results via (weighted majority voting).
> >
> >
> > || ||MATH500 | | | | | OlympiadBench| |  |
> > |-|:-:|:-:|:-:|:-:|:-:|:-:|:-:|:-:|:-:|:-:|
> > |K|2|4|8|16|32|2|4|8|16|32|
> > |***Base Model: OctoThinker-3B-Short-Base***|
> > |OctoThinker-3B-Short-GRPO (Maj@K)|49.0|53.8|56.8|57.0|57.4|17.8|19.3| 20.2|20.4|21.5|
> > |OctoThinker-3B-Short-LaSeR (Maj@K)|51.6|56.8|61.2|63.0|64.0|18.9|21.8|24.3|**25.6**|25.9|
> > |OctoThinker-3B-Short-LaSeR (RM@K)|**52.2**|**58.0**|**61.8**|**63.8**|**64.2**|**20.3**| **23.3**|**24.9** |25.3| **26.7**|
> > |***Base Model: Qwen2.5-7B-Base***|
> > |Qwen2.5-7B-GRPO (Maj@K)| 79.6| 82.2|84.2|84.4|84.8| 43.6| 47.0| 48.6 |50.2 | 50.4 |
> > |Qwen2.5-7B-LaSeR (Maj@K)|81.2| 84.2| 85.2| **86.6**| **86.2**|45.8|47.6| 49.3 | 50.7 | **51.1**|
> > |Qwen2.5-7B-LaSeR (RM@K)| **82.6**| **85.0** | **85.4**| **86.6**| 85.8| **46.7** | **48.3** |**49.5** | **51.1** | 50.8 |
> > |***Base Model: Open-Reasoner-Zero-7B***|
> > |Open-Reasoner-Zero-7B-7B-GRPO (Maj@K)|82.8|85.0|**86.4**|**87.0**|87.8|47.7 | 51.9 | 52.3| 53.2| 53.3 |
> > |Open-Reasoner-Zero-7B-LaSeR (Maj@K)|82.6| 83.8|84.6|85.8|86.2| 49.2| 51.9 | 54.2 | 55.0 | 54.7 |
> > |Open-Reasoner-Zero-7B-LaSeR (RM@K)| **84.2** | **85.4**|86.0|86.4|**88.0**  | **51.9** | **53.0** | **54.4** | **55.1** | **55.3**|
> >
> >
> > ---
> >
> > **Q3:** Regarding the generalizability of our method to other tasks.
> >
> > **A3:**  Thank you for your suggestion. In the original submission, we primarily evaluate the out-of-distribution (OOD) generalization of our method to general reasoning tasks after training solely on math data. Here, we conduct additional experiments by performing RLVR with our method on general-reasoning datasets, thereby assessing the broader generalizability of our approach within the general reasoning domain.
> >
> > In specific, we use a filtered version [5] of WebInstruct-verified dataset [6], and conduct RL
> > experiments on Qwen3-4B-Base. We use the ```general-verifier-1.5B``` model from [6] as the model-based verifier and adopt GRPO as the RL algorithm. For our method, we do not perform the advantage integration strategy here. The reason is that we observe the self-verification F1 score of our method during training is relatively low in the general reasoning setting (only between 65% and 70%, and the self-verification F1 scores in the test sets also reveal this phenomenon).
> > This leads to large noise in the self-rewarding-based advantage estimation, and consequently, the integration of self-rewarding-based advantages results in performance degradation. After training, we conduct evaluations on two general reasoning benchmarks: MMLU-Pro [7] and GPQA-Diamond [8]. We sample 4 solutions per problem on each dataset for each model, and calculate both the average accuracy and the (weighted) majority voting accuracy. The results are shown in the following.
> >
> > Table 2. Results on general reasoning tasks.
> >
> > ||Average Accuracy | |Self-Verification F1 Score| |(Weighted) Majority Voting Accuracy||
> > |--------|:------:|:------:|:------:|:------:|:------:|:------:|
> > ||MMLU-Pro | GPQA-D  | MMLU-Pro | GPQA-D  |  MMLU-Pro | GPQA-D  |
> > |Qwen3-4B-GRPO|62.90 |43.69| 22.92 | 18.50 | 66.54 | 44.44|
> > |Qwen3-4B-LaSeR (Ours)| **62.92** | **43.94** | **59.04** | **44.35** | **67.10**  | **45.96**|
> >
> >
> > -----
> >
> > [5] Yu, Tianyu, et al. "RLPR: Extrapolating RLVR to General Domains without Verifiers." arxiv 2025
> >
> > [6] Ma, Xueguang, et al. "General-reasoner: Advancing llm reasoning across all domains." NeurIPS 2025
> >
> > [7] Wang, Yubo, et al. "Mmlu-pro: A more robust and challenging multi-task language understanding benchmark."  NeurIPS 2024
> >
> > [8] Rein, David, et al. "Gpqa: A graduate-level google-proof q&a benchmark." COLM 2024

---

> ### Author Response · Authors · 2025-11-24
> **Part 3**
>
> (following above...) We have several findings: (1)  First, we observe that **jointly optimizing the self-rewarding capability by our method does not impact the model's general reasoning ability**, allowing the policy model to achieve
> comparable average reasoning accuracy to the baseline. (2) However, as mentioned above, the optimized self-rewarding score on general reasoning tasks does not achieve the high accuracy seen in math reasoning tasks. We make further visualizations to see that the self-rewarding score distributions for correct and incorrect solutions on MMLU-Pro exhibit certain overlap, and the distinction further diminishes on the more challenging benchmark GPQA-Diamond (we will put these visualizations into the revision). We speculate that two factors may contribute to this: (a) The model's general reasoning ability is inherently weaker than its math reasoning ability, which limits the upper bound of its self-rewarding capabilities in the general reasoning domain. (b) The model-based verifier used in the experiment (```general-verifier-1.5B```) has limited verification ability, resulting in high noise in the reasoning rewards, which in turn affects the optimization of the self-rewarding capability. (3) Though not perfect, **the optimized self-rewarding scores can still provide useful signals during inference time, leading to better weighted majority voting results**. To examine this effect, we analyze the distribution of self-rewarding scores across solutions generated for the same query. We calculate and report the average *AUROC* score, which is the probability that a correct solution sampled from the same query group (solutions generated for the same query) receives a higher self-rewarding score than an incorrect one. The results are reported in the following Table 3. We find that the AUROC scores are around 0.6, indicating that within the same query group, the self-rewarding scores assigned to correct solutions are generally higher than those assigned to incorrect ones.
> This reveals that, though not perfect, **the self-rewarding signal can still effectively support solution re-ranking, thereby improving the performance of weighted majority voting.**
> A promising direction for future work is to further explore and unlock the full potential of our method in the general reasoning domain.
>
> Table 3. The average self-rewarding AUROC score of ```Qwen3-4B-LaSeR``` to assess how well the self-rewarding score separates correct from incorrect solutions. In calculation, we discard query groups in which all solutions are either correct or incorrect, as such cases provide no meaningful signal for assessing the effectiveness of the self-rewarding scores. To ensure reliability, we sample 32 times for each query. For computational efficiency, we evaluate on 1000 samples [5] from MMLU-Pro.
>
> ||MMLU-Pro-1000 |GPQA-D |
> |----|:------:|:------:|
> |AUROC|0.61| 0.58 |
>
> --------
>
> **Q4:** Regarding the performance comparison between our introduced self-rewarding MSE loss and supervised fine-tuning loss.
>
> **A4:** We have diaplayed the training rewards and self-rewarding optimization losses of both our self-rewarding loss (MSE loss) and supervised fine-tuning loss (BCE loss) in Figure 9 in the submission, in order to demonstrate the advantage of our approach: **it effectively optimizes the model's self-rewarding capability without compromising the improvement of its reasoning ability**. Here, following your suggestion, we also report the final performance of checkpoints obtained after 200 training steps under both loss choices (the final checkpoint in Figure 9). The following table shows that **while the SFT loss optimizes the self-rewarding capability very slowly, it also introduces a negative impact on reasoning performance**. As discussed in Section 3.4, this is because the SFT loss forces $\pi_{\boldsymbol{\theta}}(z_c|\boldsymbol{x}, \boldsymbol{y}) $ toward 1, which can strongly interfere with the optimization of the reasoning ability and thereby lead to optimization difficulties.
>
> Table 4. The final performance of checkpoints obtained after 200 training steps under both self-rewarding MSE loss and SFT loss.
>
> || ||Reasoning Accuracy | | | | ||Self-Verification F1 Score| | | |
> |--------|------:|------:|------:|------:|----:|------:|------:|----:|------:|----:|------:|----:|
> ||MATH500 | AMC23 |AIME24|AIME25|Olym. |Avg. | MATH500 | AMC23 |AIME24|AIME25|Olym.|Avg.|
> |SFT Loss| 81.5| 59.5 | 17.6 | 15.4 | 45.9 | 44.0| 0.0 | 0.0  | 0.0 | 0.0 | 0.0 | 0.0|
> |MSE Loss (Ours)|  **82.5** |**61.6**| **18.8** | **16.2** |  **46.5** | **45.1** | **82.3**  | **79.3** | **77.9**  | **79.2** | **78.4** | **79.4** |
>
>
>
> ----
>
> [5] Yu, Tianyu, et al. "RLPR: Extrapolating RLVR to General Domains without Verifiers." arxiv 2025

---

> ### Author Response · Authors · 2025-11-24
> **Part 4**
>
> **Q5:** Regarding the discussion on why reasoning ability would improve under LaSeR or whether the auxiliary objective would influence the model's reasoning dynamics or solution style.
>
> **A5:** As explained in **A2**, encoding the self-rewarding signal on the final token does not interfere with the optimization of reasoning ability theoretically. Therefore, the model's reasoning dynamics and solution style remain consistent with the baseline. However, incorporating self-rewarding–based advantages introduces more fine-grained reward signals and improves RL performance, and our experimental results in Table 1 demonstrate such effectiveness.
>
> -----
>
> **Q6:** Regarding the presentation in Section 3.
>
> **A6:** Thank you for your helpful suggestion. We will carefully proofread the paper and improve the readability of Section 3 in the revision.
>
> -------
>
> **Q7:** Regarding the question "How is the weighted majority voting implemented"?
>
> **A7:** As described in Lines 316–319, during testing, each sampled solution for the same query receives a self-rewarding score $ r_{s}=\beta_{v} \log \pi_{\boldsymbol{\theta}}(z_{c}|\boldsymbol{x},\boldsymbol{y}) - \beta_{v} c_{ref}$. We then clip the self-rewarding scores into the interval [0, 1]. Notably, we point out that this clipping is optional: after optimization, the self-rewarding scores naturally fall within [0, 1] (see the table below), further validating both the boundedness and effectiveness of our self-scoring mechanism. Finally, we select the final answer with the greatest total sum of self-rewarding scores to achieve weighted majority voting (RM@K).
>
> Table 5. The the range between the minimum and maximum self-rewarding scores over all solutions in each test set.
>
>
> |Model|MATH500|AMC23|AIME24|AIME25|OlympiadBench|
> |:---|:------:|:------:|:------:|:------:|:------:|
> |OctoThinker-3B-Short-LaSeR| [0.000, 1.010]|[-0.031, 1.001] |[-0.023, 0.858] | [-0.061, 0.722] | [-0.036, 1.013]|
> |Qwen2.5-7B-LaSeR | [-0.021, 1.031]| [0.013, 1.020] | [-0.052, 1.010] | [-0.041, 0.986] |  [-0.056, 1.029] |
> |Open-Reasoner-Zero-7B-LaSeR | [-0.005, 1.013]|  [-0.085, 1.005] |  [-0.035, 0.998]|[-0.031, 0.963] | [-0.036, 1.016]|
>
>
> ----
>
> **Q8:** Regarding the selection of the pre-specified special token for optimizing the self-rewarding capability.
>
> **A8:** The pre-specified special token $z_c$ can be any unused special token, as we find that for all such tokens, $\pi_{ref}(z_c|x,y) < e^{-20}$ holds true for all $(x,y)$.
>
> ------
>
> **Q9:** Regarding the results and analysis on general reasoning tasks.
>
> **A9:** Refer to the analysis in **A3**.
>
> -----
>
> We hope the above response addresses your questions. We are incorporating the new results and discussions into the revision, and will upload the revision once it is ready.

---

### Official Review · Reviewer_mU64 · 2025-10-31

**Soundness:** 4
**Presentation:** 4
**Contribution:** 2
**Rating:** 4
**Confidence:** 3

**Summary:**

The paper proposes LaSeR, a novel policy optimization method that enables a model to self-verify its own reasoning outputs efficiently by leveraging the next-token probability of a pre-specified special token as a self-rewarding score, thereby improving test-time scaling. The authors report improved reasoning accuracy, self-verification F1, and test-time scaling efficiency across multiple reasoning benchmarks.

**Strengths:**

1. The paper is well-written and easy to follow, with clear theoretical motivation and solid empirical validation.

2. Extensive experiments across multiple reasoning benchmarks and ablation studies demonstrate robustness

3. The proposed idea of last-token self-rewarding is conceptually simple but empirically effective.

**Weaknesses:**

1. The proposed idea overlaps with prior work such as ReVISE [1], which also learns to self-verify using a special-token confidence and applies it for weighted voting at test time.

2. The comparison with external verifiers (in Appendix F) may not be fair without training the verifier on the same dataset and backbone.

3. Although several components are introduced (re-weighting, warm-ups, advantage integration), ablation studies are insufficient to isolate their effects.

4. Evaluation is limited to math reasoning; generalization to other domains (code [2,3], general-reasoning [4]) is unclear.
---
[1] Lee et al., Revise: Learning to refine at test-time via intrinsic self-verification, ICML 2025\
[2] Austin et al., MBPP: Program Synthesis with Large Language Models\
[3] HumanEval:Evaluating Large Language Models Trained on Code\
[4] AlpacaEval: An Automatic Evaluator of Instruction-following Models

**Questions:**

1. How does LaSeR perform under Best-of-N [5] evaluation without weighting?

2. What is the actual inference-time gain (in token count or latency) compared to prior self-verification approaches?
---
[5] Snell et al., Scaling LLM Test-Time Compute Optimally can be More Effective than Scaling Model Parameters

---

> ### Author Response · Authors · 2025-11-24
> **Author Response (Part 1)**
>
> We sincerely thank you for your thoughtful comments and constructive questions. We are glad that you think our paper has clear theoretical motivation and solid empirical validation. We are pleased to see that you believe our idea conceptually simple yet empirically effective. To address your remaining concerns, we make the following detailed response.
>
> ----
>
> **Q1:** Regarding the difference between our work and the previous work ReVISE [1].
>
> **A1:** We appreciate your pointing out this relevant work. The core idea of ReVISE is to let the model choose between two tokens, ```[eos]``` and ```[refine]```, at the end of a response: generating ```[eos]``` indicates that the model believes the solution is correct and terminates generation, whereas generating ```[refine]``` signals that the model considers the solution incorrect and should continue producing a refinement. From the viewpoint of self-verification training, **this formulation is equivalent to optimizing the SFT/BCE loss defined in Eq. (16)**:
>
> $$L_{SFT}= -E_{\boldsymbol{x}\sim D, \boldsymbol{y} \sim \pi_{g}(\cdot | x)} \left[ r_{v}(\boldsymbol{x},\boldsymbol{y}) \cdot \log \pi_{\boldsymbol{\theta}} (z_{c}|\boldsymbol{x},\boldsymbol{y})  + (1 - r_{v} (\boldsymbol{x},\boldsymbol{y})) \cdot \log \pi_{\boldsymbol{\theta}} (z_{i}|\boldsymbol{x},\boldsymbol{y}) \right],$$
> where $z_c$ and $z_i$ are ```[eos]``` and ```[refine]``` respectively.
>
>
> **Our framework is fundamentally different from the above SFT loss, both in formulation and in learning dynamics.** We have discussed the difference between our method and using an SFT loss in the second paragraph of Section 3.4. Here, we would like to provide further explanations and highlight that **the solution to the above SFT loss is in fact a special case of all available solutions in our framework**.
> In the following, we first make detailed derivations to obtain the general form of our self-rewarding MSE loss. Then, we point out the relationship between SFT loss and our framework.
>
> We note that in Eq. (7), the rewards for correct and incorrect verifications can be any two values, $\hat{r}\_{c}$ and $\hat{r}\_{i}$ (though 0/1 is the common choice)​. Under this general definition, following the same derivation as in Eq.(9), we obtain $Z(\boldsymbol{x},\boldsymbol{y})= exp(\frac{\hat{r}\_i}{\beta_v})$, which indicates $\log Z(\boldsymbol{x},\boldsymbol{y})=\frac{\hat{r}\_i}{\beta_v}$ (still a constant but not necessarily to be 0). Then, the optimal solution to the original RL target can be reduced to a general form
> $$
> \hat{r}(\boldsymbol{x},\boldsymbol{y},\boldsymbol{z}) = \beta_{v} \log [\pi_{\boldsymbol{\theta}}(\boldsymbol{z}|\boldsymbol{x},\boldsymbol{y}) / \pi_{ref}(\boldsymbol{z}|\boldsymbol{x},\boldsymbol{y})] + \hat{r}\_{i}.
> $$
> Now, the true verification reward when the model verifies a solution as correct is
> $$
> \hat{r}(\boldsymbol{x},\boldsymbol{y},z_{c}) = \hat{r}\_{c}I_{r_{v}(\boldsymbol{x},\boldsymbol{y})=1} + \hat{r}\_{i} I_{r_{v}(\boldsymbol{x},\boldsymbol{y})=0} =  \beta_{v} \log [\pi_{\boldsymbol{\theta}}(z_{c}|\boldsymbol{x},\boldsymbol{y}) / \pi_{ref}(z_{c}|\boldsymbol{x},\boldsymbol{y})] + \hat{r}\_{i}.
> $$
> Note that the reasoning rewards $r_{v}(\boldsymbol{x}, \boldsymbol{y})$ defined in Eq. (2) can also be two arbitrary values, but we simplified it to $(1,0)$ and this will not affect our derivation. In this general framework, our self-rewarding MSE loss is
> $$
> L=E_{\boldsymbol{x}\sim D, \boldsymbol{y}\sim \pi_{g}(\cdot | x)}\left( \beta_{v} \log [ \pi_{\boldsymbol{\theta}}(z_{c}|\boldsymbol{x},\boldsymbol{y}) / \pi_{ref}(z_{c}|\boldsymbol{x},\boldsymbol{y})] +\hat{r}\_i - (\hat{r}\_{c}I_{r_{v}(\boldsymbol{x},\boldsymbol{y})=1} + \hat{r}\_{i} I_{r_{v}(\boldsymbol{x},\boldsymbol{y})=0}) \right)^{2}.
> $$
>
>
> ----
>
> [1] Lee, Hyunseok, et al. "Revise: Learning to refine at test-time via intrinsic self-verification." ICML 2025

---

> ### Author Response · Authors · 2025-11-24
> **Part 2**
>
> (following above ...) After optimization (i.e., the optimal solution to the MSE loss), we can see that when $r_v(\boldsymbol{x},\boldsymbol{y})=0$ (i.e., the solution is incorrect based on the deterministic verifier), $\pi_{\boldsymbol{\theta}}(z_c|\boldsymbol{x},\boldsymbol{y}) = \pi_{ref}(z_c|\boldsymbol{x},\boldsymbol{y}) $, which means the $\pi_{\boldsymbol{\theta}}(z_c|\boldsymbol{x},\boldsymbol{y})$ remains unchanged (almost near zero, consistent with the solution to SFT loss when $r_v(\boldsymbol{x},\boldsymbol{y})=0$). When $r_v(\boldsymbol{x},\boldsymbol{y})=1$ (i.e., the solution is correct based on the deterministic verifier), our method drives $\pi_{\boldsymbol{\theta}}(z_c|\boldsymbol{x},\boldsymbol{y})$ to fit $\pi_{ref}(z_c|\boldsymbol{x},\boldsymbol{y}) exp(\frac{\hat{r}\_{c}-\hat{r}\_{i}}{\beta_v}) $. As we can see, when $\beta_v$ continues to decrease, the optimized probability $\pi_{\boldsymbol{\theta}}(z_c|\boldsymbol{x},\boldsymbol{y})$ approaches 1, which is exactly the solution to the SFT loss. However, in our framework, **we can explicitly control the value of $\beta_v$, allowing $\pi_{\boldsymbol{\theta}}(z_c |\boldsymbol{x}, \boldsymbol{y})$ to converge to any desired target**. As shown in Figure 9, this controllability provides greater flexibility compared with directly forcing $\pi_{\boldsymbol{\theta}}(z_c |\boldsymbol{x}, \boldsymbol{y})$ toward 1, while also exerting much less interference on the optimization of the reasoning ability. Thus, **our framework is a fundamentally different but more general method**.
>
> We will incorporate the above derivation and discussion into the revision to strengthen our paper.
>
> ------
>
> **Q2:** Regarding the comparison with external verifiers.
>
> **A2:** First, we conduct an additional comparison of LaSeR's verification performance against a state-of-the-art external ORM, ```Qwen2.5-Math-RM-72B```, to further highlight the remarkable self-rewarding
> capabilities of our models. The full results are shown in the following Table 1.
>
> Then, we follow your suggestion to train an ORM using the same backbone and dataset to enable a fair comparison with LaSeR. Specifically, we take the ```Open-Reasoner-Zero-7B``` as the ORM backbone and ```DeepMath-103K``` as the dataset. Then, we use the policy model ```Open-Reasoner-Zero-7B-LaSeR``` to generate a total of 500K solutions (matching the total number of rollouts produced during its RL training, and
> we also perform re-sampling to keep a balanced ratio between correct and incorrect solutions). We then train ```Open-Reasoner-Zero-7B-RM``` on these labled solutions and evaluate its verification performance on the outputs of ```Open-Reasoner-Zero-7B-LaSeR``` on test sets. The comparison results are presented in Table 1, which further demonstrates the comparable verification performance of LaSeR against ```Open-Reasoner-Zero-7B-RM```. However, it is worth noting that additionally training Open-Reasoner-Zero-7B-RM **requires roughly extra 480 GPU hours and it always requires additional GPUs and inference cost whenever doing verification**, whereas our method jointly optimizes the reasoning and self-rewarding capabilities during RLVR and introduces **no extra computational cost in both training and testing**, demonstrating its high efficiency.
>
> Table 1. Comparison of verification F1 scores between LaSeR (self-rewarding) and external reward models
> (```Qwen2.5-Math-7B-PRM800K```, ```Qwen2.5-Math-PRM-7B```, ```Qwen2.5-Math-RM-72B```, and our trained baseline ```Open-Reasoner-Zero-7B-RM```) on responses generated by the same policy model ```Open-Reasoner-Zero-7B-LaSeR```.
>
> |Method|MATH500|AMC23|AIME24|AIME25|OlympiadBench|Avg.|
> |:---|:------:|:------:|:------:|:------:|:------:|:------:|
> |Qwen2.5-Math-7B-PRM800K (7B RM)  | 56.3 | 42.5 | 51.4 | 50.8 | 38.5|  47.9|
> |Qwen2.5-Math-PRM-7B (7B RM) | 86.0 | 79.6 | 70.8 | 67.3 |  76.0 | 75.9 |
> |Qwen2.5-Math-RM-72B (72B RM)  | 86.8 | 79.4  | 71.0  | 71.4 | 75.5 | 76.8|
> |Open-Reasoner-Zero-7B-RM (7B RM)  | 85.9 |  78.1 | **73.8** | **79.2** | 77.3 | **78.9** |
> |LaSeR (7B Self-Rewarding) |**87.2** |**79.7** | 64.6 | 77.7 | **78.7** | 77.6|

---

> > ### Author Response · Authors · 2025-11-24
> > **Part 3**
> >
> > **Q3:** Regarding the ablation studies on introduced components.
> >
> > **A3:** We would like to clarify that **our submission has included all the necessary ablation studies to demonstrate the effectiveness of each component**. In specific,
> >
> > - The ablation studies on self-rewarding hyper-parameters (coefficient $\beta_v$ and self-rewarding MSE loss weight $\alpha$) are in **Figure 7** in Appendix H.
> >
> > - **Table 2** in Section 5.1 demonstrates the feasibility and efficiency of the simplification of reference log-probability ($\log \pi_{ref}(z_c|x,y)$).
> >
> > - **Figure 8** in Appendix I validates the effectiveness of performing class-level self-rewarding loss re-weighting.
> >
> > - Ablation results in **Table 1** (*-SWA*) shows the effectivess of introducing self-rewarding-based advantage integration.
> >
> > - Regarding the warm-up practices, (1) in our preliminary experiments, we have found that directly applying self-rewarding–based advantage integration without self-rewarding warm-up phase leads to **training collapse** (the training reward drops rapidly to near zero within the first few steps). This occurs because, before optimization, the self-rewarding scores are essentially random values near zero, and utilizing the advantages computed based on them introduces substantial noise to training. (2) As noted in Lines 310–312, the reasoning warm-up stage is not inherently necessary: as long as the base model has certain initial reasoning capabilities, self-rewarding and reasoning optimization can be conducted in parallel. However, a pre-trained model initially produces almost no correct solutions, creating a highly skewed distribution of correct versus incorrect samples within each batch. Consequently, a brief reasoning warm-up is beneficial for stabilizing this ratio and ensuring more reliable training dynamics.
> >
> >
> > ----------
> >
> > **Q4:** Regarding the generalizability of our method to other tasks.
> >
> > **A4:** Thank you for your suggestion. In the original submission, we primarily evaluate the out-of-distribution (OOD) generalization of our method to general reasoning tasks after training solely on math data. Here, we conduct additional experiments by performing RLVR with our method on general-reasoning datasets, thereby assessing the broader generalizability of our approach within the general reasoning domain.
> >
> > In specific, we use a filtered version [2] of WebInstruct-verified dataset [3], and conduct RL
> > experiments on ```Qwen3-4B-Base```. We use the ```general-verifier-1.5B``` model from [3] as the model-based verifier and adopt GRPO as the RL algorithm. For our method, we do not
> > perform the advantage integration strategy here. The reason is that we observe the self-verification F1 score of our method during training is relatively low in the general reasoning setting (only between 65% and 70%, and the self-verification F1 scores in the test sets also reveal this phenomenon).
> > This leads to large noise in the self-rewarding-based advantage estimation, and consequently, the integration of self-rewarding-based advantages results in performance degradation. After training, we conduct evaluations on two general reasoning benchmarks: MMLU-Pro [4] and GPQA-Diamond [5]. We
> > sample 4 solutions per problem on each dataset for each model, and calculate both the average accuracy and the
> > (weighted) majority voting accuracy. The results are shown in the following.
> >
> > Table 2. Results on general reasoning tasks.
> >
> > ||Average Accuracy | |Self-Verification F1 Score| |(Weighted) Majority Voting Accuracy||
> > |--------|:------:|:------:|:------:|:------:|:------:|:------:|
> > ||MMLU-Pro | GPQA-D  | MMLU-Pro | GPQA-D  |  MMLU-Pro | GPQA-D  |
> > |Qwen3-4B-GRPO|62.90 |43.69| 22.92 | 18.50 | 66.54 | 44.44|
> > |Qwen3-4B-LaSeR (Ours)| **62.92** | **43.94** | **59.04** | **44.35** | **67.10**  | **45.96**|
> >
> >
> > -----
> >
> > [2] Yu, Tianyu, et al. "RLPR: Extrapolating RLVR to General Domains without Verifiers." arxiv 2025
> >
> > [3] Ma, Xueguang, et al. "General-reasoner: Advancing llm reasoning across all domains." NeurIPS 2025
> >
> > [4] Wang, Yubo, et al. "Mmlu-pro: A more robust and challenging multi-task language understanding benchmark."  NeurIPS 2024
> >
> > [5] Rein, David, et al. "Gpqa: A graduate-level google-proof q&a benchmark." COLM 2024

---

> ### Author Response · Authors · 2025-11-24
> **Part 4**
>
> (following above...) We have several findings: (1)  First, we observe that **jointly optimizing the self-rewarding capability by our method does not impact the model's general reasoning ability**, allowing the policy model to achieve
> comparable average reasoning accuracy to the baseline. (2) However, as mentioned above, the optimized self-rewarding score on general reasoning tasks does not achieve the high accuracy seen in math reasoning tasks. We make further visualizations to see that the self-rewarding score distributions for correct and incorrect solutions on MMLU-Pro exhibit certain overlap, and the distinction further diminishes on the more challenging benchmark GPQA-Diamond (we will put these visualizations into the revision). We speculate that two factors may contribute to this: (a) The model's general reasoning ability is inherently weaker than its math reasoning ability, which limits the upper bound of its self-rewarding capabilities in the general reasoning domain. (b) The model-based verifier used in the experiment (```general-verifier-1.5B```) has limited verification ability, resulting in high noise in the reasoning rewards, which in turn affects the optimization of the self-rewarding capability. (3) Though not perfect, **the optimized self-rewarding scores can still provide useful signals during inference time, leading to better weighted majority voting results**. To examine this effect, we analyze the distribution of self-rewarding scores across solutions generated for the same query. We calculate and report the average *AUROC* score, which is the probability that a correct solution sampled from the same query group (solutions generated for the same query) receives a higher self-rewarding score than an incorrect one. The results are reported in the following Table 3. We find that the AUROC scores are around 0.6, indicating that within the same query group, the self-rewarding scores assigned to correct solutions are generally higher than those assigned to incorrect ones.
> This reveals that, though not perfect, **the self-rewarding signal can still effectively support solution re-ranking, thereby improving the performance of weighted majority voting.**
> A promising direction for future work is to further explore and unlock the full potential of our method in the general reasoning domain.
>
> Table 3. The average self-rewarding AUROC score of ```Qwen3-4B-LaSeR``` to assess how well the self-rewarding score separates correct from incorrect solutions. In calculation, we discard query groups in which all solutions are either correct or incorrect, as such cases provide no meaningful signal for assessing the effectiveness of the self-rewarding scores. To ensure reliability, we sample 32 times for each query. For computational efficiency, we evaluate on 1000 samples [2] from MMLU-Pro.
>
> ||MMLU-Pro-1000 |GPQA-D |
> |----|:------:|:------:|
> |AUROC|0.61| 0.58 |
>
> ---------
>
> **Q5:** Regarding the performance of Best-of-N against weighted majority voting.
>
> **A5:** First, we would like to point out that **the paper you refer to [6] actually adopts a "Best-of-N Weighted" technique rather than the standard Best-of-N**, as stated in the paper: "we apply 'best-of-N weighted' selection rather than standard best-of-N. Best-of-N weighted selection marginalizes the verifier's correctness scores across all solutions with the same final answer, selecting the final answer with the greatest total sum." **This is exactly the Weighted Majority Voting strategy employed in our experiments.**
>
> Then, we follow you question to compare the performance of standard Best-of-N strategy (selecting the answer with highest reward score) against Weighted Majority Voting (Best-of-N Weighted). The results are shown in the following Table 4. As shown, the performance of standard Best-of-N saturates when $N$ becomes large and underperforms Weighted Majority Voting, aligning with previously reported results and observations in the literature [7].
>
>
> Table 4. Comparison between Majority Voting (Maj@N), standard Best-of-N and Weighted Majority Voting (RM@N) for inference-time scaling performance. The experimental model is ```Open-Reasoner-Zero-7B-LaSeR```.
>
>
> || ||MATH500 | | | | | OlympiadBench| |  |
> |--------|:------:|:------:|:------:|:------:|:----:|:------:|:------:|:----:|:------:|:----:|
> |N|2 |  4 | 8 | 16 | 32 | 2 | 4 | 8 | 16 | 32 |
> |Maj@K| 82.6 | 83.8 |84.6 |85.8 |86.2| 49.2 |51.9 |54.2 |55.0|54.7|
> |Best-of-N| **84.2** | 84.0 | 83.8 | 84.0 | 83.8 |  **51.9** |  51.4| 51.1 | 51.9 | 50.5|
> |RM@K|**84.2**  |**85.4** |**86.0**| **86.4** |**88.0** | **51.9** |**53.0**|**54.4**|**55.1**|**55.3**|
>
> ------
>
>
> [6] Snell, Charlie, et al. "Scaling llm test-time compute optimally can be more effective than scaling model parameters." arxiv 2024
>
> [7] Sareen, Kusha, et al. "Putting the Value Back in RL: Better Test-Time Scaling by Unifying LLM Reasoners With Verifiers." arxiv 2025

---

> ### Author Response · Authors · 2025-11-24
> **Part 5**
>
> **Q6:** Regarding the actual inference-time gain compared with prior self-verification approaches.
>
> **A6:** We follow your suggestion to compare the latency of our self-rewarding mechanism with prior self-verification approaches that require separate solution and verification generation. Using the same experimental conditions (vLLM version 0.8.5, identical sampling parameters, etc.), we evaluate the latency of evaluation on MATH500 by measuring the time to generate one solution for each query and the time to execute the corresponding self-verification for each method. The time cost of self-verification for LaSeR is estimated as the total time required to generate solutions and compute the self-rewarding scores immediately after solution generation via one more token inference, minus the time spent solely on solution generation.
> The comparison results in the following Table 5 show that in contrast to the baseline that requires separate generations for the solution and the verification using two distinct prompts, our method produces the self-verification signal immediately after solution generation, thereby **significantly improving verification efficiency by fully leveraging the KV-cache mechanism in vLLM inference**.
>
>
> Table 5. Comparison of inference time between LaSeR and the baseline approach (separate solution and verification generations) on 500 test queries from MATH500. The experimental models are ```Open-Reasoner-Zero-7B-GRPO``` and ```Open-Reasoner-Zero-7B-LaSeR```.
>
>
> |Method|Time Cost of Generating Solutions (s)|Time Cost of Self-Verification (s)|
> |--------|:------:|:------:|
> |Baseline| 104 | 148  |
> |LaSeR| 128  | 1 |
>
> ----
>
>
> We hope the above response addresses your questions. We are incorporating the new results and discussions into the revision, and will upload the revision once it is ready.

---

### Official Review · Reviewer_CW7C · 2025-11-03

**Soundness:** 3
**Presentation:** 3
**Contribution:** 3
**Rating:** 8
**Confidence:** 4

**Summary:**

LaSeR trains a policy to encode a self‑evaluation signal in the final step of its answer: a last‑token score that tracks whether the just‑generated solution is correct. Instead of running a separate verifier sequence, the method adds a tiny auxiliary loss so that this one‑token score matches a standard verifier’s binary reward during RLVR. Inference then needs only the policy’s next‑token distribution at the last position, which also supports inexpensive weighted voting across samples. Experiments on several math benchmarks show small but consistent accuracy gains over GRPO baselines and strong self‑verification F1.

**Strengths:**

1. Elegant and practical idea. This method compresses self‑verification into a single last‑token readout, avoiding a second generation pass and adding almost no training or inference cost.

2. Near‑zero overhead: this method requires no second pass, just a next‑token query; training-time cost is essentially free since token log‑probs are already computed.

3. Empirical Justification: The method improves Pass@1 across multiple backbones and yields high self‑verification F1; the score boosts inference‑time scaling via simple weighted voting.

4. The argument about length bias in cumulative implicit rewards is valid and demonstrated (Appendix B), aligning with new empirical results suggesting shorter chains often outperform longer ones.

**Weaknesses:**

1. Limited breadth: The evaluation centers on math; results on a general reasoning set show only modest separation. More domains (code, QA, open‑ended tasks) would strengthen the case.

2. Calibration analysis of self-rewarding: the reward model here is essentially a binary classification model. F1 is a good metric but not comprehensive enough. It would strengthen the paper if the authors could include reliability diagrams / ECE / AUROC for the last‑token score vs. correctness, and analyze length effects (short vs. long CoT) in line with recent findings that shorter chains can be preferable.

3. This work misses discussions of prior RLHF (not RLVR) papers that incorporates reward signals explicitly in RL training.

**Questions:**

1. Reference‑free test time: How stable is the pre‑computed constant across model updates or decoding settings? Would a quick online calibration (or an ensemble of special tokens) reduce drift?

---

> ### Author Response · Authors · 2025-11-24
> **Author Response (Part 1)**
>
> We sincerely thank you for your positive review and constructive comments. We are glad that you think the idea is elegant and practical. We are encouraged that you think our method is efficient with near-zero overhead and achieves great empirical effectiveness. We make the following response to address your questions.
>
> -----
>
> **Q1:** Regarding the generalizability of our method to other tasks.
>
> **A1:** Thank you for your suggestion. In the original submission, we primarily evaluate the out-of-distribution (OOD) generalization of our method to general reasoning tasks after training solely on math data. Here, we conduct additional experiments by performing RLVR with our method on general-reasoning datasets, thereby assessing the broader generalizability of our approach within the general reasoning domain.
>
> In specific, we use a filtered version [1] of WebInstruct-verified dataset [2], and conduct RL
> experiments on ```Qwen3-4B-Base```. We use the ```general-verifier-1.5B``` model from [2] as the model-based verifier and adopt GRPO as the RL algorithm. For our method, we do not perform the advantage integration strategy here. The reason is that we observe the self-verification F1 score of our method during training is relatively low in the general reasoning setting (only between 65% and 70%, and the self-verification F1 scores in the test sets also reveal this phenomenon).
> This leads to large noise in the self-rewarding-based advantage estimation, and consequently, the integration of self-rewarding-based advantages results in performance degradation. After training, we conduct evaluations on two general reasoning benchmarks: MMLU-Pro [3] and GPQA-Diamond [4]. We
> sample 4 solutions per problem on each dataset for each model, and calculate both the average accuracy and the
> (weighted) majority voting accuracy. The results are shown in the following.
>
> Table 1. Results on general reasoning tasks.
>
> ||Average Accuracy | |Self-Verification F1 Score| |(Weighted) Majority Voting Accuracy||
> |-|:-:|:-:|:-:|:-:|:-:|:-:|
> ||MMLU-Pro |GPQA-D|MMLU-Pro| GPQA-D  |  MMLU-Pro | GPQA-D  |
> |Qwen3-4B-GRPO|62.90 |43.69| 22.92|18.50| 66.54| 44.44|
> |Qwen3-4B-LaSeR (Ours)| **62.92** |**43.94**|**59.04**| **44.35**| **67.10**| **45.96**|
>
> We have several findings: (1)  First, we observe that **jointly optimizing the self-rewarding capability by our method does not impact the model's general reasoning ability**, allowing the policy model to achieve
> comparable average reasoning accuracy to the baseline. (2) However, as mentioned above, the optimized self-rewarding score on general reasoning tasks does not achieve the high accuracy seen in math reasoning tasks. We make further visualizations to see that the self-rewarding score distributions for correct and incorrect solutions on MMLU-Pro exhibit certain overlap, and the distinction further diminishes on the more challenging benchmark GPQA-Diamond (we will put these visualizations into the revision). We speculate that two factors may contribute to this: (a) The model's general reasoning ability is inherently weaker than its math reasoning ability, which limits the upper bound of its self-rewarding capabilities in the general reasoning domain. (b) The model-based verifier used in the experiment (```general-verifier-1.5B```) has limited verification ability, resulting in high noise in the reasoning rewards, which in turn affects the optimization of the self-rewarding capability. (3) Though not perfect, **the optimized self-rewarding scores can still provide useful signals during inference time, leading to better weighted majority voting results**. To examine this effect, we analyze the distribution of self-rewarding scores across solutions generated for the same query. We calculate and report the average **AUROC** score, which is the probability that a correct solution sampled from the same query group (solutions generated for the same query) receives a higher self-rewarding score than an incorrect one. The results are reported in the following Table 2. We find that the AUROC scores are around 0.6, indicating that within the same query group, the self-rewarding scores assigned to correct solutions are generally higher than those assigned to incorrect ones.
> This reveals that, though not perfect, **the self-rewarding signal can still effectively support solution re-ranking, thereby improving the performance of weighted majority voting.**
> A promising direction for future work is to further explore and unlock the full potential of our method in the general reasoning domain.
>
> ---
> [1] Yu, Tianyu, et al. "RLPR: Extrapolating RLVR to General Domains without Verifiers." arxiv 2025
>
> [2] Ma, Xueguang, et al. "General-reasoner: Advancing llm reasoning across all domains." NeurIPS 2025
>
> [3] Wang, Yubo, et al. "Mmlu-pro: A more robust and challenging multi-task language understanding benchmark."  NeurIPS 2024
>
> [4] Rein, David, et al. "Gpqa: A graduate-level google-proof q&a benchmark." COLM 2024

---

> ### Author Response · Authors · 2025-11-24
> **Part 2**
>
> Table 2. The average self-rewarding AUROC score of ```Qwen3-4B-LaSeR``` to assess how well the self-rewarding score separates correct from incorrect solutions. In calculation, we discard query groups in which all solutions are either correct or incorrect, as such cases provide no meaningful signal for assessing the effectiveness of the self-rewarding scores. To ensure reliability, we sample 32 times for each query. For computational efficiency, we evaluate on 1000 samples [1] from MMLU-Pro.
>
> ||MMLU-Pro-1000 |GPQA-D |
> |----|:------:|:------:|
> |AUROC|0.61| 0.58 |
>
>
> ----
>
> **Q2:** Regarding the calibration analysis of self-rewarding scores.
>
> **A2:** Following your suggestion, we first analyze the **AUROC** score of the self-rewarding scores with respect to correctness. The AUROC is defined as above: it measures the probability that, within the same query group, a correct solution receives a higher self-rewarding score than an incorrect one. Here, we focus specifically on the mathematics benchmarks. The evaluation results on ```Open-Reasoner-Zero-7B-LaSeR``` are shown in the following table. We also report the AUROC of the reward scores from a state-of-the-art external ORM ```Qwen2.5-Math-RM-72B``` for comparison. The results demonstrate **the high discriminative power of the self-rewarding score between correct and incorrect solutions**.
>
>
> Table 3. The average AUROC of reward scores on each benchmark. In calculation, we discard query groups in which all solutions are either correct or incorrect, as such cases provide no meaningful signal for assessing the effectiveness of the self-rewarding scores. To ensure reliability, we sample 32 times for each query. The generator is ```Open-Reasoner-Zero-7B-LaSeR```.
>
> |Method|MATH500|AMC23|AIME24|AIME25|OlympiadBench|Avg.|
> |:---|:------:|:------:|:------:|:------:|:------:|:------:|
> |Qwen2.5-Math-RM-72B (72B RM)|**0.77**| **0.73**|0.71|0.60| 0.65| 0.69|
> |LaSeR (7B Self-Rewarding)|0.72|0.68 |**0.77**|**0.74**|**0.67**|**0.72**|
>
> Then, we also calculate and report the **Expected Calibration Error (ECE)** of all reward scores on each benchmark in the following table for your reference. ECE quantifies the discrepancy between a self-rewarding score and its empirical accuracy. The number of score bins is 10 in our calculation. As we can see, **our method achieves better confidence calibration**.
>
> Table 4. ECE of reward scores on each benchmark (lower is better). To ensure reliability, we sample 32 times for each query. The generator is ```Open-Reasoner-Zero-7B-LaSeR```.
>
> |Method|MATH500|AMC23|AIME24|AIME25|OlympiadBench|Avg.|
> |:---|:------:|:------:|:------:|:------:|:------:|:------:|
> |Qwen2.5-Math-RM-72B (72B RM)| 0.178| **0.152** | 0.346| 0.384| 0.194| 0.251|
> |LaSeR (7B Self-Rewarding)| **0.077** | 0.162 |  **0.072** | **0.071** | **0.066** | **0.090**|
>
> Finally, we analyze the effect of response lengths. We calculate and report the average **Spearman Rank Correlation** between self-rewarding scores and response lengths over all query groups on each test set. Spearman Rank Correlation measures the direction and strength of the monotonic relationship between two variables by computing the correlation between their ranked values. The results in the following Table 4 reveals that **shorter responses generally tend to receive higher self-rewarding scores, which is consistent with recent findings that shorter CoTs on the same query are generally more preferable**.
>
> Table 5. The Spearman Rank Correlation between self-rewarding scores and response lengths on ```Open-Reasoner-Zero-7B-LaSeR```. To ensure reliability, we sample 32 times for each query.
>
> ||MATH500|AMC23|AIME24|AIME25|OlympiadBench|Avg.|
> |:---|:------:|:------:|:------:|:------:|:------:|:------:|
> |Spearman Rank Corr. |-0.26| -0.34|-0.58|-0.52|-0.40|-0.42|
>
>
> ----
>
> [1] Yu, Tianyu, et al. "RLPR: Extrapolating RLVR to General Domains without Verifiers." arxiv 2025

---

> > ### Author Response · Authors · 2025-11-24
> > **Part 3**
> >
> > **Q3:** Regarding the discussion of prior RLHF (not RLVR) papers that incorporate rewards signals explicitly in RL training.
> >
> > **A3:** Reinforcement Learning with Human Feedback (RLHF) [5] has been a key technique for aligning a model’s capabilities and behaviors with human preferences. The standard pipeline typically consists of three stages: supervised fine-tuning (SFT) on high-quality instructional data, learning a reward model from pairwise human preference data, and optimizing the policy model via reinforcement learning to maximize the learned reward.
> > Prior work in RLHF spans a broad range of directions, including but not limited to: (1) Developing more effective or efficient RLHF algorithms and frameworks [6,7,8]; (2) Constructing diverse, large-scale, and high-quality preference datasets [9,10]; (3) Training stronger or domain-specific reward models to improve RLHF performance [11,12,13]; (4) Exploring multi-objective alignment [14,15].
> >
> > We will include the above discussion into the revision.
> >
> > ------
> > **Q4:** Regarding the stability of the pre-computed constant across model updates.
> >
> > **A4:** Thank you for your insightful question. Following your suggestion, we report the dynamics of both the mean and standard deviation of $-log \pi_{ref}(z_c|\boldsymbol{x},\boldsymbol{y})$ (where $\boldsymbol{y}$ is generated by the policy model) throughout the policy model's training process. Specifically, we calculate the mean and standard deviation of $-log \pi_{ref}(z_c|\boldsymbol{x},\boldsymbol{y})$ based on the 300 generated samples from the policy model at training steps 0, 100, 200, …, and 1000. We conduct evaluations by taking ```Qwen2.5-7B-Base``` as the base model. The dynamics are presented in the following Table 6. As shown, **the mean of $-log \pi_{ref}(z_c|\boldsymbol{x},\boldsymbol{y})$ remains stable throughout the policy model updates**, which supports our motivation and confirms the feasibility of approximating it as a pre-computed constant in our method to improve efficiency. Furthermore, the ablation results in Table 2 in the submission also helps to validate the feasibility of this practice.
> >
> >
> > Table 6. The dynamics of mean and standard deviation of $-log \pi_{ref}(z_c|\boldsymbol{x},\boldsymbol{y})$ during policy model's training.
> >
> > |Checkpoint|0th Step| 100th Step| 200th Step|300th Step| 400th Step| 500th Step| 600th Step|700th Step| 800th Step|900th Step| 1000th Step|
> > |:---:|:---:|:---:|:---:|:---:|:---:|:---:|:---:|:---:|:---:|:---:|:---:|
> > | mean $\pm$ std |23.04 $\pm$ 0.07 | 23.07 $\pm$ 0.05 |  23.09 $\pm$ 0.06  | 23.06  $\pm$  0.05 | 23.10  $\pm$ 0.14 | 23.10  $\pm$ 0.16 | 23.08 $\pm$ 0.05 | 23.09 $\pm$ 0.08 |  23.09 $\pm$  0.09 | 23.09 $\pm$ 0.06 |  23.09 $\pm$ 0.08 |
> >
> > -----
> >
> > We hope the above response addresses your questions. We are incorporating the new results and discussions into the revision, and will upload the revision once it is ready.
> >
> > [5] Ouyang, Long, et al. "Training language models to follow instructions with human feedback."  NeurIPS 2022
> >
> > [6] Schulman, John, et al. "Proximal policy optimization algorithms." arxiv 2017
> >
> > [7] Ahmadian, Arash, et al. "Back to basics: Revisiting reinforce style optimization for learning from human feedback in llms." arxiv 2024
> >
> > [8] Rafailov, Rafael, et al. "Direct preference optimization: Your language model is secretly a reward model." NeurIPS 2023
> >
> > [9] Bai, Yuntao, et al. "Training a helpful and harmless assistant with reinforcement learning from human feedback." arxiv 2022
> >
> > [10] Cui, Ganqu, et al. "Ultrafeedback: Boosting language models with high-quality feedback." ICML 2024
> >
> >
> > [11] Dong, Hanze, et al. "Rlhf workflow: From reward modeling to online rlhf." arxiv 2024
> >
> > [12] Cai, Zheng, et al. "Internlm2 technical report."  arxiv 2024
> >
> > [13] Yuan, Lifan, et al. "Advancing llm reasoning generalists with preference trees." ICLR 2025
> >
> > [14] Guo, Yiju, et al. "Controllable preference optimization: Toward controllable multi-objective alignment." EMNLP 2024
> >
> > [15] Li, Chengao, et al. "Gradient-Adaptive Policy Optimization: Towards Multi-Objective Alignment of Large Language Models." ACL 2025

---

### Author Response · Authors · 2025-11-27
**Revision Summary**

Dear all reviewers,

We sincerely thank your for your great efforts on reviewing our paper, as well as for your constructive comments. **We are pleased that all reviewers find our work elegant and novel, and we are encouraged by the recognition that our method is both effective and efficient.**

We have followed all your suggestions and incorporated the new experiments and extended discussions into the revised version. Below, we briefly summarize the main changes made in the revision:


-  We include the detailed derivations to present the general form of our method and add the discussions on the difference between our self-rewarding MSE loss and  BCE loss in Appendix C. (Reviewer ```mU64```, ```9vmN```, and ```Z5vT```)

- We include the detailed results and analysis on general reasoning tasks in Appendix R to demonstrate the generalizability of our method. (Reviewer ```CW7C```, ```mU64```, and ```xmH7```)

- We include the dynamics of reference log-probability $\log \pi_{ref}( z_c | \boldsymbol{x}, \boldsymbol{y})$ during policy model's training in Appendix E to validate the practice of simplifying reference log-probability to a pre-computed constant. (Reviewer  ```CW7C``` and ```Z5vT```)

- We make further explanations in Section 3.3 to better illustrate how we perform weighted majority voting, and show the results of the value ranges of the optimized self-rewarding scores in Appendix M to validate the boundedness of our self-scoring mechanism. (Reviewer   ```xmH7``` and ```9vmN```)

- We revise Section 3 and fix the minor derivation error in Eq. (9). (Reviewer  ```xmH7``` and ```9vmN```)

- We display the detailed calibration analysis of self-rewarding scores in Section 5.1 and Appendix O. (Reviewer  ```CW7C```)

- We include the discussion on RLHF (not RLVR) papers in Appendix B. (Reviewer  ```CW7C```)

- We put the comparison results of verification F1 scores against several external reward models (including our trained ORM on the same backbone and dataset) in Table 2 and Table 6 to show the remarkable self-rewarding capabilities of our models. (Reviewer ```mU64```)

-  We revise the introduction to make the motivation of  joint reasoning–verification training clearer. (Reviewer ```xmH7```)

- We put the results of applying our method within PPO framework in Appendix Q to demonstrate the generality of our method. (Reviewer ```9vmN```)

- We put ablation results of only using self-rewarding scores for RL in Appendix L. (Reviewer ```Z5vT```)

- We show the results of cross-model verification performance in Appendix P to show that our method not only enables effective self-rewarding, but also achieves high accuracy and F1 when evaluating the CoTs generated by other models, demonstrating strong generalization ability. (Reviewer ```Z5vT```)

The above modifications are all highlighted in blue in the revision.

We thank all reviewers once again, and we truly appreciate the helpful suggestions that have significantly strengthened our paper. We are glad to continue the discussion if there are any further questions.

Best regards,

Authors of Submission 9562

---

### Author Response · Authors · 2025-12-01
**Message to SAC/AC**

Dear SACs and ACs,

We sincerely appreciate the additional time and effort you devote in response to the unexpected information leakage incident. We have made substantial efforts to address the concerns raised by all reviewers and have incorporated new results and discussions into the revised manuscript, which has been uploaded. The purpose of this message is to provide a summary of the reviews, and to outline how we have addressed each of the raised concerns.

---

### Summary of reviews
First, the main contribution of our work is to propose **a novel and highly effective algorithm that jointly optimizes the reasoning and self-rewarding capabilities of LLMs with *nearly zero additional cost*, and the enhanced self-rewarding capability leads to consistent and remarkable improvements in both training and inference-time scaling performance**. We are encouraged by the following recognition and support from the reviewers:

- **Elegant and well-grounded theoretical derivations, novel and practically idea.** (Reviewer ```CW7C```, ```mU64```, ```xmH7```, and ```Z5vT```)

- **Very efficient method with near-zero overhead** to enable LLMs to perform self-rewarding. (Reviewer ```CW7C```, ```xmH7```, ```9vmN```, and ```Z5vT```)

- **Comprehensive experiments demonstrating the effectiveness of the proposed method**, particularly its substantial improvement of inference-time scaling performance through weighted majority voting with optimized self-rewarding scores. (all reviewers)

---


### Summary of rebuttal and revision
Below, we first summarize our responses to several common and important questions raised by the reviewers.

>  **Regarding more explanations of our theoretical framework and the difference between our self-rewarding MSE loss and standard BCE loss.** (Reviewer ```mU64```, ```9vmN```, and ```Z5vT```)

We include the detailed derivations to present the general form of our method and add the discussions on the difference between our self-rewarding MSE loss and BCE loss in Appendix C.

> **Regarding the generalizability to other domains.** (Reviewer ```CW7C```, ```mU64```, and ```xmH7```)

We include the detailed results and analysis on general reasoning tasks in Appendix R to demonstrate the generalizability of our method. (Reviewer CW7C, mU64, and xmH7)

> **Regarding the stability and the validity of the simplification of reference log-probability.** (Reviewer ```CW7C``` and ```Z5vT```)

We follow reviewers' helpful suggestions to show the dynamics of reference log-probability $\log \pi_{ref}( z_c | \boldsymbol{x}, \boldsymbol{y})$ during policy model's training in Appendix E to validate the practice of simplifying reference log-probability to a pre-computed constant.

> **Regarding how to perform weighted majority voting and the boundedness of self-rewarding scores.** (Reviewer ```xmH7``` and ```9vmN```)

We make further explanations in Section 3.3 to better illustrate how we perform weighted majority voting, and show the results of the value ranges of the optimized self-rewarding scores in Appendix M to validate the boundedness of our self-scoring mechanism.

Other responses to the individual comments from each reviewer are:

- We display the detailed calibration analysis of self-rewarding scores in Section 5.1 and Appendix O to show the good properties of our self-rewarding mechanism. (Reviewer  ```CW7C```)

- We include the discussion on RLHF papers in Appendix B. (Reviewer  ```CW7C```)

- We put the comparison results of verification F1 scores against several external reward models (including our trained ORM on the same backbone and dataset) in Table 2 and Table 6 to show the remarkable self-rewarding capabilities of our models. (Reviewer ```mU64```)

- We show the comparison results of Best-of-N. (Reviewer ```mU64```)

- We provide the actual inference-time gain compared with prior self-verification approaches to validate the remarkable efficiency of our method. (Reviewer ```mU64```)

-  We revise the introduction to make the motivation of  joint reasoning–verification training clearer. (Reviewer ```xmH7```)

- We put the results of applying our method within PPO framework in Appendix Q to demonstrate the generality of our method. (Reviewer ```9vmN```)

- We put ablation results of only using self-rewarding scores for RL in Appendix L. (Reviewer ```Z5vT```)

- We show the results of cross-model verification performance in Appendix P to show that our method not only enables effective self-rewarding, but also achieves high accuracy and F1 when evaluating the CoTs generated by other models, demonstrating strong generalization ability. (Reviewer ```Z5vT```)

---

We sincerely thank all the reviewers for their constructive comments, which have greatly strengthened our paper. We also thank you once again for your additional efforts in the review process. We hope the above summary will assist you in reviewing our submission more efficiently.

Best regards,

Authors of Submission 9562

---

### Meta-Review · Area_Chair_VC52 · 2026-01-08

**Summary:**

This paper proposes LaSeR, an RL algorithm that jointly optimizes LLM reasoning and self-rewarding via a simple MSE loss on the last token's log-prob, enabling efficient self-verification with near-zero extra cost. It shows gains in math benchmarks and inference scaling. Reviewers praised the elegant theory and efficiency but noted modest gains, limited scope, and minor derivation issues.

**Reviewer Concerns:**

Rebuttals addressed generalizability with new general reasoning results, calibration via AUROC/ECE, comparisons to external RMs, stability of approximations, and ablations on components. Outstanding: some derivation tweaks needed; cross-domain results show promise but lower self-reward F1 than in math; modest reasoning lifts persist.

**Reviewer Scores:**

CW7C likely holds at 8, as rebuttals add calibration and generalizability. mU64 may rise to 6, with fairer comparisons and broader evals. xmH7 could bump to 6, motivation clarified and gains contextualized. 9vmN probably to 6, derivation fixed and PPO shown. Z5vT stays at 6, with added stability and cross-model data.

---

### Decision · Program_Chairs · 2026-01-26

Accept (Poster)